# DEMYSTIFYING LINEAR MDPS AND NOVEL DYNAMICS AGGREGATION FRAMEWORK

**Joongkyu Lee**
Graduate School of Data Science
Seoul National University
jklee0717@snu.ac.kr

**Min-hwan Oh**
Graduate School of Data Science
Seoul National University
minoh@snu.ac.kr

## ABSTRACT

In this work, we prove that, in linear MDPs, the feature dimension $d$ is lower bounded by $S/U$ in order to aptly represent transition probabilities, where $S$ is the size of the state space and $U$ is the maximum size of directly reachable states. Hence, $d$ can still scale with $S$ depending on the direct reachability of the environment. To address this limitation of linear MDPs, we propose a novel structural aggregation framework based on dynamics, named as the *dynamics aggregation*. For this newly proposed framework, we design a provably efficient hierarchical reinforcement learning algorithm in linear function approximation that leverages aggregated sub-structures. Our proposed algorithm exhibits statistical efficiency, achieving a regret of $\tilde{\mathcal{O}}\big(d_\psi^{3/2} H^{3/2} \sqrt{NT}\big)$, where $d_\psi$ represents the feature dimension of *aggregated subMDPs* and $N$ signifies the number of aggregated subMDPs. We establish that the condition $d_\psi^3 N \ll d^3$ is readily met in most real-world environments with hierarchical structures, enabling a substantial improvement in the regret bound compared to `LSVI-UCB`, which enjoys a regret of $\tilde{\mathcal{O}}(d^{3/2} H^{3/2} \sqrt{T})$ (Jin et al., 2020). To the best of our knowledge, this work presents the first HRL algorithm with linear function approximation that offers provable guarantees.

## 1 INTRODUCTION

Recent theoretical research in reinforcement learning (RL) has seen a surge in studies focusing on function approximation. Such a research direction seeks to address the generalization problem faced in tabular Markov Decision Processes (MDPs) (Jiang et al., 2017; Yang & Wang, 2019; 2020; Jin et al., 2020; Zanette et al., 2020; Modi et al., 2020; Du et al., 2020; Cai et al., 2020; Ayoub et al., 2020; Wang et al., 2020; Weisz et al., 2021; He et al., 2021; Zhou et al., 2021a;b; Ishfaq et al., 2021; Hu et al., 2022). The linear MDP (Bradtke & Barto, 1996; Jin et al., 2020) serves as a foundational model for function approximation, modeling the transition probability as $\mathbb{P}(s' \mid s, a) = \phi(s, a)^\top \mu(s')$ with known features $\phi \in \mathbb{R}^d$ and unknown measures $\{\mu(s')\}_{s' \in \mathcal{S}}$. Numerous prior studies have demonstrated regret bounds that are not dependent on the size of the state space $S$ (or the action space size $A$), but instead on the feature dimension $d$. (Jin et al., 2020; Zanette et al., 2020; Du et al., 2020; Cai et al., 2020; Weisz et al., 2021; He et al., 2021; Zhou et al., 2021a;b; Ishfaq et al., 2021). Consequently, many of these algorithms proposed for linear MDPs are proven to achieve regret bounds independent of the size of the state space, and depend only on the intrinsic complexity measure of the feature space, $d$, once the parameterization is applied. However, whether such replacement of the state space dependence with the dependence on feature dimension $d$ induces learning independently of the state space entirely for all MDPs still requires an investigation. Hence, we pose a critical research question:

**Q1**: *Does the linear MDP invariably yield regrets that are independent of the state space size $S$?*

In this paper, we rigorously investigate the conditions under which linear MDPs induce learning that is independent of the state space and the conditions under which they do not. Our findings, as detailed in Section 4, prove that the feature dimension $d$ is lower bounded by $S/U$ in order to aptly represent the probability space, where $U$ is the maximum size of directly reachable states (see Definition 2). Thus, if the cardinality of directly reachable states does not grow with the entirety of the state space, that is, $U = o(S)$ — a condition that holds true in *most real-world situations* and

becomes more pronounced as $S$ expands — the feature dimension $d$ has to grow proportionally with $S$ to properly encode the probability distribution over next states. Hence, unless the size of reachable states scales with the entire state space, regret bounds under linear MDPs still implicitly have $S$ dependence through the dependence of $d$ on $S$.[1] To the best of our knowledge, our study presents the first comprehensive exposition of the fundamental limitations of the linear MDP, particularly its intrinsic dependence on state space.

Our results on the limitations of linear MDPs suggest that simply because function approximation is employed, it may not necessarily enable efficient learning where the feature dimension $d$ is independent of the state space. However, should additional structures, such as hierarchies, be present within linear MDPs — facilitating the decomposition of the MDP into smaller sub-problems — it paves the way for the development of a refined framework, possibly enabling efficient learning. Ideally, a well-constructed learning algorithm should then leverage such structures for more efficient learning. Yet, to the best of our knowledge, there is no existing model or algorithm for hierarchical reinforcement learning (HRL) with function approximation that provides regret guarantees. Therefore, the following research question arises:

**Q2**: *Can we formulate a new hierarchical framework for linear MDPs that enables provably efficient learning independent of state space?*

To answer this question, we first introduce the framework of *dynamics aggregation*, which clusters similar sub-structures based on their dynamics of MDPs. Notably, this concept not only includes the extensively studied notion of state aggregation (or state abstraction) (Singh et al., 1994; Van Roy, 2006; Li et al., 2006; Abel et al., 2020; Dong et al., 2019) but also integrates the equivalence mapping proposed in Wen et al. (2020). A key benefit of dynamics aggregation lies in its reusability for similar problems. This new notion of aggregation not only allows efficient learning in technical perspectives but also is very natural in practical perspectives. Then, we propose *linear transition models for aggregated subMDPs*, a generalized approach that extends both non-hierarchical linear MDPs (Jiang et al., 2017; Jin et al., 2020) and tabular MDPs with equivalent subMDPs (Wen et al., 2020).

Under this newly proposed model, we design a model-based HRL algorithm that leverages the hierarchical structure of MDPs and employs optimistic planning. This algorithm is provably efficient and, to our knowledge, is the first HRL algorithm that offers provable guarantees with function approximation. In numerical experiments, our proposed method consistently outperforms existing algorithms by significant margins. Our main contributions can be summarized as follows:

- We establish that the feature dimension $d$ is lower bounded by $S/U$, where $U$ represents the maximum size of directly reachable states (Theorem 1). We also provide examples of various environments where $d$ does not scale with $S$. Consequently, in such scenarios, the regret bound can indeed depend on the size of the state space $S$ despite function approximation. To the best of our knowledge, this is the first work to provide a rigorous proof showing how the feature dimension $d$ relates to the state space size $S$ in linear MDPs. We strongly believe that this finding provides significant implications and will be of independent interest to the broader RL community.

- To address this fundamental issue of the vanilla linear MDP framework, we introduce a new comprehensive framework of *dynamics aggregation*, encompassing both state aggregation and equivalence mapping (Wen et al., 2020). One of the key benefits of this framework lies in its inherent ability to be reused for similar sub-problems.

- Under this newly proposed framework, we present a statistically efficient algorithm that exploits the hierarchical structure of the problem, thereby reducing dependency on the size of the entire state space. Then, we establish a regret bound of $\widetilde{\mathcal{O}}\big(d_\psi^{3/2} H^{3/2} \sqrt{NT} + TH\epsilon_p\big)$ (Theorem 2), where $d_\psi$ represents the feature dimension of aggregated subMDPS, $N$ denotes the number of aggregated subMDPs, and $\epsilon_p$ is the aggregation error. If an MDP adheres to the conditions of Corollary 1 (a common circumstance) and exhibits a hierarchical structure, the condition $d_\psi^3 N \ll d^3$ can be readily fulfilled, dramatically reducing the regret upper bound compared to `LSVI-UCB` (Jin et al., 2020), which enjoys a regret of $\widetilde{\mathcal{O}}(d^{3/2} H^{3/2}\sqrt{T})$.

---

[1]It is important to note that our results do not contradict the previously known $S$-independent regret bounds of the algorithms for linear MDPs (Jin et al., 2020). Rather, we focus on the representation ability of linear MDPs and its feature dimension $d$'s potential dependence on $S$.

- We also conduct numerical experiments in environments with suitable hierarchical structures and show that our proposed framework enables our algorithm to leverage the structure and consistently outperform the existing RL algorithms with provable guarantees.

## 2 RELATED WORK

**Reinforcement Learning with Linear Function Approximation.** In recent years, there has been a surge in research on function approximation with provable guarantees (Jiang et al., 2017; Yang & Wang, 2019; 2020; Jin et al., 2020; Zanette et al., 2020; Modi et al., 2020; Du et al., 2020; Cai et al., 2020; Ayoub et al., 2020; Wang et al., 2020; Weisz et al., 2021; He et al., 2021; Zhou et al., 2021a;b; Ishfaq et al., 2021). All of these works assume certain linear structures of underlying MDP and appear to handle large state spaces effectively, as their regret scales only polynomially in $d$ and not $S$. However, it remains unclear how $d$ is related to $S$ in linear MDPs. In Theorem 1, we provide proof showing that $d$ is lower bounded by $S/U$, where $U$ represents the maximum size of directly reachable states. And in Corollary 1, 2, and 3, we establish that $d$ can be proportional to $S$ in the majority of real-world environments.

**State Aggregation** The study of state aggregation (or state abstraction) in RL has a long and rich history, dating back to early works on approximating dynamic programs and the identification of states that exhibit similar behaviors (Fox, 1973; Whitt, 1978; Bean et al., 1987; Dean & Givan, 1997; Bertsekas et al., 1988). In a similar vein, Li et al. (2006) introduced a unified framework for state aggregation in MDPs, examining the conditions under which such aggregations can preserve optimal behavior and affect the existing convergence guarantees of well-known RL algorithms. However, unlike our proposed dynamics aggregation which embraces a hierarchical structure, these past studies did not explicitly leverage this concept.

**Hierarchical Reinforcement Learning (HRL).** Several studies have explored the decomposition of MDP into sub-problems (Dean & Lin, 1995; Singh & Cohn, 1997; Meuleau et al., 1998) and then solved independently under the weakly coupled resource constraints. The concept of HRL, which allows an agent to act and plan at various levels of temporal abstraction, was established bySutton et al. (1999); Barto & Mahadevan (2003). However, there has been limited research quantifying the theoretical benefits of HRL. The work most closely related to ours is by Wen et al. (2020), who introduced a model-based tabular HRL algorithm designed to leverage repeating sub-structures. Nevertheless, their research focused solely on tabular MDPs when utilizing hierarchical structures, leaving the development of an efficacious HRL algorithm for linear MDPs an open question.

## 3 PROBLEM SETTING

### 3.1 NOTATIONS

We denote by $[n]$ the set $\{1, 2, \ldots, n\}$ for a positive integer $n$. For a real-valued matrix $A$, we use $\|A\|_2 := \sup_{x:\|x\|_2=1} \|Ax\|_2$ to denote the maximum singular value of $A$. With a positive definite matrix $\Lambda$, we denote $\|x\|_\Lambda^2 := x^\top \Lambda x$. We denote $|\cdot|$ as the cardinality of a set.

### 3.2 INHOMOGENEOUS, EPISODIC MDPS

We consider inhomogeneous episodic Markov decision processes (MDPs) denoted by $\mathcal{M}(\mathcal{S}, \mathcal{A}, H, \{\mathbb{P}_h\}_{h=1}^H, \{r_h\}_{h=1}^H)$, where $\mathcal{S}$ is a measurable space, potentially with an infinite number of elements, and has a cardinality of $S$, $\mathcal{A}$ is a finite set with cardinality $A$, $H \in \mathbb{Z}_+$ is the length of each episode, $\mathbb{P}_h$ is the collection of transition probability distributions, and $r_h$ is a reward function. We assume that every state is accessible from at least one other state, i.e. $\forall s', \sum_{(s,a) \in \mathcal{S} \times \mathcal{A}} \mathbb{P}_h(s' \mid s, a) \geqslant 0$ [2].

In each episode, an initial state $s_1$ is picked arbitrarily by an adversary. Then, for every $h \in [H]$ in an episode, an agent takes action $a_h \in \mathcal{A}$ for state $s_h \in \mathcal{S}$ and receives reward $r_h(s_h, a_h) \in [0, 1]$. Then, the next state $s_{h+1}$ is is drawn from the transition probability distribution $\mathbb{P}_h(\cdot \mid s_h, a_h)$ and repeats its interactions until the end of the episode.

---

[2]If a specific state is not accessible, we can exclude it without losing generality.

The agent aims to find a policy $\pi : \mathcal{S} \times [H] \to \mathcal{A}$ that maximizes its expected cumulative reward starting from every state $s$. We define the value function of policy $\pi$, $V_h^\pi : \mathcal{S} \to \mathbb{R}$ as the expected sum of rewards under the policy $\pi$ until the end of the episode when starting from $s_h = s$, i.e., $V_h^\pi(s) := \mathbb{E}_\pi \left[ \sum_{h'=h}^H r_{h'}(s_{h'}, \pi(s_{h'}, h')) \,|\, s_h = s \right]$. We also denote the action-value function of policy $\pi$, $Q_h^\pi : \mathcal{S} \times \mathcal{A} \to \mathbb{R}$ as the expected sum of rewards when following $\pi$ starting from step $h$ until the end of the episode after taking action $a$ in state $s$; that is, $Q_h^\pi(s, a) := r_h(s, a) + \mathbb{E}_\pi \left[ \sum_{h'=h+1}^H r_{h'}(s_{h'}, \pi(s_{h'}, h')) \,|\, s_h = s, a_h = a \right]$.

A policy $\pi^*$ is said to be an *optimal* policy if it achieves the maximal possible value at every state-step pair $(s, h) \in \mathcal{S} \times [H]$. Then, we define the optimal value and action-value functions as $V_h^*(s) := V_h^{\pi^*}(s) = \sup_\pi V_h^\pi(s)$ and $Q_h^*(s, a) := Q_h^{\pi^*}(s, a) = \sup_\pi Q_h^\pi(s, a)$. For a simple notation, by denoting $\mathbb{P}_h V_{h+1}(s, a) := \mathbb{E}_{s' \sim \mathbb{P}_h(\cdot|s,a)}[V_{h+1}(s')]$, both $Q^\pi$ and $Q^*$ can be conveniently written as the result of the Bellman equations as $Q_h^\pi(s, a) = (r_h + \mathbb{P}_h V_{h+1}^\pi)(s, a)$ and $Q_h^*(s, a) = (r_h + \mathbb{P}_h V_{h+1}^*)(s, a)$, where, for all $s \in \mathcal{S}$, $V_{H+1}^\pi(s) = V_{H+1}^*(s) = 0$ and $V_h^*(s) = \max_{a \in \mathcal{A}} Q_h^*(s, a)$.

## 4 Limitations of Linear MDPs

There exists a large amount of literature on function approximation in which linear MDPs serve as a foundational model (Yang & Wang, 2019; Jin et al., 2020; Zanette et al., 2020; Hu et al., 2022). Despite the growing body of research, the limitations associated with linear MDPs have not been adequately addressed. In this section, we provide a comprehensive analysis of inherent limitations in linear MDPs. First, linear transition models of linear MDPs are defined as follows:

**Definition 1** (Linear transition model). *Let there exist a known feature map $\phi : \mathcal{S} \times \mathcal{A} \to \mathbb{R}^d$ and unknown $\mu_h : \mathcal{S} \to \mathbb{R}^d$. Then, the transition operator $\mathbb{P}_h : \mathcal{S} \times \mathcal{A} \to \Delta(\mathcal{S})$ is defined as follows: for all $s, s' \in \mathcal{S}, a \in \mathcal{A}$, $\mathbb{P}_h(s' \mid s, a) = \phi(s, a)^\top \mu_h(s')$.*

The linear structure of the transition probabilities offers the advantage of reducing the number of parameters that need to be estimated, subsequently decreasing the statistical and computational complexity of learning and planning algorithms. However, it is crucial to acknowledge that the set of MDPs that can be accurately represented using linear transition models with small $d$ relative to the size of the state space is notably limited.

For a linear MDP, we generally expect that the transition kernel $\mathbb{P}_h(\cdot \mid \cdot, \cdot) \in \mathbb{R}^{SA \times S}$ has a low-dimensional structure, i.e., $d \ll S$. However, the following statements show that the feature dimension is closely related to the size of the state space, highlighting the inherent limitations associated with linear transition models.

**Definition 2** (Directly reachable states). *For each $(s, a) \in \mathcal{S} \times \mathcal{A}$, "directly reachable states" of $(s, a)$ is defined to be the set of all states which can be reached by taking action $a$ in state $s$ within a single transition, $\mathcal{S}_{s,a} := \{s' \in \mathcal{S} : \mathbb{P}_h(s' \mid s, a) > 0\}$. Also, we denote $U := \max_{(s,a) \in \mathcal{S} \times \mathcal{A}} |\mathcal{S}_{s,a}|$ to be the maximum size of directly reachable states.*

**Theorem 1.** *For an MDP $\mathcal{M}$ with a finite state space, the feature dimension $d$ is lower bonded by*

$$d \geq \lfloor S/U \rfloor,$$

*where $U$ is the maximum size of directly reachable states (Defitiontion 2).*

**Corollary 1.** *If the maximum size of directly reachable states $U$ does not scale with the entire state space by a constant factor, i.e., $U = \Theta(S^p) < \infty$, where $0 \leq p < 1$, then $d \geq \Omega(S^{1-p})$.*

Theorem 1 and Corollary 1 imply that unless the size of directly reachable states (one-step transitable states) scales with the entire state space $S$ — a scenario rarely true in most real-world cases — the feature dimension $d$ would eventually scale with the size of the entire state space polynomially. Consequently, the learning efficiency (e.g., regret) would still depend on $S$ even when function approximation is employed. Furthermore, Theorem 1 can be generalized to an infinite (or even continuous) state space.

**Corollary 2** (Infinite $S$ & finite $U$). *For an MDP $\mathcal{M}$ with a state space that is either countably infinite or normed, compact, and uncountably infinite, and with a finite $U$, $d$ is infinite.*

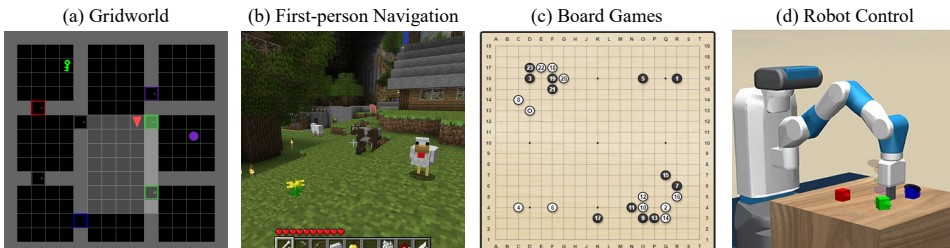

|(a) Gridworld|(b) First-person Navigation|(c) Board Games|(d) Robot Control|

Figure 1: Various environments where the feature dimension scales with the size of the state space.

**Corollary 3** (Euclidean continuous state space). *Consider an MDP $\mathcal{M}$ with state space $\mathcal{S}$ in the $p$-dimensional Euclidean space. Let $\mathrm{Vol}(\cdot)$ represent the volume of a set. Denote the set of directly reachable states with the maximum volume as $\mathcal{U} = \arg\max_{\mathcal{S}_{s,a}} \mathrm{Vol}(\mathcal{S}_{s,a})$ and assume that $\mathrm{Vol}(\mathcal{U}) > 0$. Then, we have $d > 2^p \cdot \mathrm{Vol}(\mathcal{S})/\mathrm{Vol}(\mathcal{U}) - 1$.*

One can observe that most real-world environments, as well as many simulation environments, have a small $U$ compared to the size of the state space. This implies that the statement in Corollary 1 is widely applicable and persuasive. Identifying environments that do not meet the condition of Corollary 1 is rather a challenging endeavor. The following examples, which are widely studied in the RL literature, fulfill the condition:

**Example 1** (Gridworld). *In Figure 1 (a), the agent is allowed to move to neighboring states (left, right, up, down, or stay in the same state), resulting in $U = 5$. Thus, by Theorem 1, $d \geqslant \lfloor S/5 \rfloor$. In a special case where the transitions are deterministic ($U = 1$), we get $d = S$.*

**Example 2** (First-person navigation). *In Figure 1 (b), although the entire state space is extremely large, the agent can only move to the neighboring states, resulting in a constant $U$. Thus, $d \geqslant \Omega(S)$.*

**Example 3** (Board games). *Board games like Go, depicted in Figure 1 (c), have an immense state space, approximately $10^{400}$, but the number of directly reachable states is relatively small, fewer than $19^2$. Hence, $d \gtrsim 2.5 \times 10^{397}$.*

**Example 4** (Control problems). *The state spaces in control problems, as depicted in Figure 1 (d), are continuous (uncountable). The volume of sets of directly reachable states is typically much smaller—especially in cases with minimal stochasticity—than the volume of the full state space. Therefore, $d \geqslant \Omega(\mathrm{Vol}(\mathcal{S}))$.*

To sum up, many existing studies that assume linear MDPs establish regret bounds that depend on the embedding dimension $d$ rather than the size of the state space $S$. However, in most practical environments, $d$ is often proportional to $S$. Consequently, it is crucial to take the state space size into account when employing linear MDPs in real-world applications, as the assumption of linear transition model may (and often does) fail to yield significant improvements in computational or statistical complexity. Motivated by these findings, in the following sections, we study approaches where additional structure may alleviate the limitations of vanilla linear MDPs.

## 5 HIERARCHICAL STRUCTURE

In the context of MDPs, we introduce a notion of modularity (Wen et al., 2020), which divides a large problem into smaller ones. Modularity could be addressed separately and solved independently. Then, sub-problem solutions could be *stitched* together to solve the original problem. This approach can lead to statistically efficient learning if sub-problems are reasonably small and recurring.

**Definition 3** (Sub-problems, Wen et al. 2020). *Assume that the state space $\mathcal{S}$ is divided into $L$ disjoint subgroups $\{\mathcal{S}^i\}_{i=1}^L$. Then, induced subMDPs $\mathcal{M}^i(\mathcal{S}^i \cup \mathcal{E}^i, \mathcal{A}, \{\mathbb{P}_h^i\}_{h=1}^H, \{r_h^i\}_{h=1}^H, \mathcal{E}^i)$ are defined as:*

- *The **internal state set** $\mathcal{S}^i$ is disjoint subset of $\mathcal{S}$ and the action space is still $\mathcal{A}$.*

- *The **exit state set** $\mathcal{E}^i := \{e \in \mathcal{S} \setminus \mathcal{S}^i : \exists (s,a) \in \mathcal{S}^i \times \mathcal{A} \ s.t. \ \mathbb{P}^i(e \mid s,a) > 0\}$.*

- *The state space of $\mathcal{M}^i$ is $\mathcal{S}^i \cup \mathcal{E}^i$.*

- *The supports of $\mathbb{P}_h^i$ and $r_h^i$ are restricted to $\mathcal{S}^i \times \mathcal{A}$.*

- *The subMDP $\mathcal{M}^i$ terminates once the agent reaches an exit state, i.e. $s \in \mathcal{E}^i$.*

Given a partition of $\mathcal{M}$, we examine the collection of induced subMDPs, represented as $\{\mathcal{M}^i\}_{i=1}^L$. If these sub-problems exhibit similarity or identical characteristics, it is possible to solve a single instance and apply the derived solution to other equivalent or analogous cases.

## 5.1 Hierarchical Structure via Dynamics Aggregation

To formalize the hierarchical structure, we employ a concept of state aggregation (or abstraction) method that groups subMDPs exhibiting "behavioral equivalence" (Singh et al., 1994; Li et al., 2006; Wen & Van Roy, 2017; Dong et al., 2019). Employing state aggregation leads to a reduction in state space size or complexity, thereby accelerating the learning process. Inspired by this concept, we propose a new concept called **dynamics aggregation**, which groups sub-MDPs based on the similarity of their dynamics. This approach involves dividing the set of states into $N$ *aggregated subMDPs*,

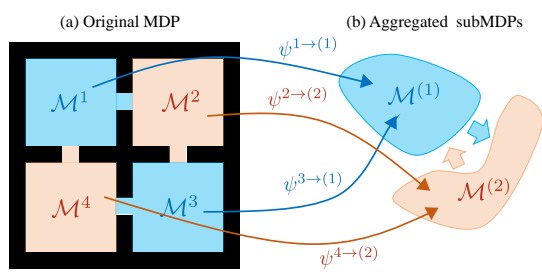

Figure 2: Dynamics aggregation

denoted by $\mathcal{M}^{(n)}(\mathcal{S}^{(n)} \cup \mathcal{E}^{(n)}, \mathcal{A}, \{\mathbb{P}_h^{(n)}\}_{h=1}^H, \{r_h^{(n)}\}_{h=1}^H, \mathcal{E}^{(n)})$ for $n \in [N]$. By employing dynamics aggregation, we can efficiently learn and generalize across different sub-problems with similar dynamics, leading to more effective and faster learning. Formally, we can define an approximate dynamics aggregation as follows:

**Definition 4** (Approximate dynamics aggregation). *For all $h \in [H]$, $i, j \in [L]$, let $\psi_h^{i \to (n)} : \mathcal{S}^i \cup \mathcal{E}^i \to \mathcal{S}^{(n)} \cup \mathcal{E}^{(n)}$ be a mapping that maps the state space of $i$'th subMDP $\mathcal{S}^i$ to its corresponding aggregated state space $\mathcal{S}^{(n)}$, where $n \in [N]$. Let $\psi_h^{i \to (n)}$ and $\psi_h^{j \to (n)}$ exist. Then, for all states $s_1 \in \mathcal{S}^i$, $s_2 \in \mathcal{S}^j$ where $\psi_h^{i \to (n)}(s_1) = \psi_h^{j \to (n)}(s_2)$ and all $a \in \mathcal{A}$, the following conditions hold:*

$$|r_h^i(s_1, a) - r_h^j(s_2, a)| \leqslant \epsilon_r, \quad \left\| \mathbb{P}_h^i \Psi_h^{i \to (n)}(\cdot \mid s_1, a) - \mathbb{P}_h^j \Psi_h^{j \to (n)}(\cdot \mid s_2, a) \right\|_1 \leqslant \epsilon_p,$$

*where $\epsilon_r, \epsilon_p \in \mathbb{R}^+ \cup \{0\}$, and $\Psi_h^{i \to (n)} \in \mathbb{R}^{S \times \bar{S}}$, $S = \sum_{i \in [L]} |\mathcal{S}^i| = |\mathcal{S}|$ and $\bar{S} = \sum_{n \in [N]} |\mathcal{S}^{(n)}|$, is a kernel that satisfying:*

$$\Psi_h^{i \to (n)}(s', \bar{s}') = \mathbb{I}\left(s' \in \mathcal{S}^i \cup \mathcal{E}^i, \bar{s}' \in \mathcal{S}^{(n)} \cup \mathcal{E}^{(n)}, \psi_h^{i \to (n)}(s') = \bar{s}'\right),$$

*where $\mathbb{I}(\cdot)$ is an indicator function that maps to 1 when the condition is true, and 0 otherwise.*

Note that $\mathbb{P}_h^i \Psi_h^{i \to (n)}(\cdot \mid s, a)$ collapses the transition distribution over $\mathcal{S}^i \cup \mathcal{E}^i$ into $\mathcal{S}^{(n)} \cup \mathcal{E}^{(n)}$, and if the dynamics aggregation mapping is exact, then $\epsilon_r = 0$ and $\epsilon_p = 0$.

Dynamics aggregation partitions the original MDPs into subMDPs and projects these subMDPs into aggregated subMDPs, while explicitly considering repeating structures (see Figure 2). This concept encompasses both state aggregation (refer Definition 3 in Li et al. (2006)) and equivalence mapping introduced by Wen et al. (2020). It not only aggregates similar (usually neighboring) states like state aggregation but also aggregates subMDPs that have similar dynamics, akin to equivalence mapping This methodology enables a significant simplification of the representation compared to the other two frameworks. For a more in-depth comparison with other existing frameworks, please refer to Section D in the Appendix.

Intuitively, if all subMDPs are unique, i.e., there are no duplicate substructures, then $N = L$. And if there are subMDPs have the similar dynamics to each other, i.e., some sub-structures are repeated, $N < L$. Thus, we can expect dramatic improvements over the standard algorithms when MDP $\mathcal{M}$ has a hierarchical structure such that $\mathbf{M} \cdot \mathbf{N} \ll \mathbf{S}$, where $M = \max_n |\mathcal{S}^{(n)} \cup \mathcal{E}^{(n)}|$. If $M$ is small, the sizes of all aggregated subMDPs are small, making each subMDP relatively easy to solve. If $N$ is small, a solution to one aggregated subMDP can be reused in other aggregated subMDPs.

## 5.2 Linear Transition Model under the Hierarchical Structure

We assume that the transition probabilities of aggregated subMDP $\mathcal{M}^{(n)}$ are linear.

**Assumption 1** (Linear transition models for aggregated subMDPs). *Denote $d_\psi$ as the feature dimension of aggregated subMDPs. For each $(\bar{s}, a) \in \mathcal{S}^{(n)} \times \mathcal{A}$, let **known** feature vector $\phi(\bar{s}, a) \in \mathbb{R}^{d_\psi}$ be given as a prior. Then, for all $n \in [N]$, there exist $\bar{S}$ **unknown** $d$-dimensional measures $\boldsymbol{\mu}_h^{(n)} = (\mu_h(1), \ldots, \mu_h(\bar{S})) \in \mathbb{R}^{d_\psi \times \bar{S}}$, where $\bar{S} = \sum_{n \in [N]} |\mathcal{S}^{(n)}|$, such that*

$$\mathbb{P}_h^{(n)}(\cdot \mid \bar{s}, a) = \phi(\bar{s}, a)^\top \boldsymbol{\mu}_h^{(n)}(\cdot), \quad \forall \bar{s} \in \mathcal{S}^{(n)},$$

*where each columns of $\boldsymbol{\mu}_h^{(n)}$ corresponds to a unknown vector $\mu_h^{(n)}(\bar{s}') \in \mathbb{R}^{d_\psi}$ for $\forall \bar{s}' \in \mathcal{S}^{(n)} \cup \mathcal{E}^{(n)}$ and $\mathbf{0} \in \mathbb{R}^{d_\psi}$ for $\forall \bar{s}' \notin \mathcal{S}^{(n)} \cup \mathcal{E}^{(n)}$.*

We make the following bounded assumptions, similar to existing literature (Yang & Wang, 2019; Jin et al., 2020): For all $h \in [H]$ (i) $\sup_{s,a} \|\phi(\psi_h^{i \to (n)}(s), a)\|_2 \leqslant C_\phi$, and (ii) $\|\boldsymbol{\mu}_h^{(n)} v\|_2 \leqslant C_{\boldsymbol{\mu}} \cdot \sqrt{d_\psi}$ for any vector $v \in \mathbb{R}^S$ such that $\|v\|_\infty \leqslant 1$. We further assume that the reward function $r$ is known for simplicity[3]. Since we consider low-rank linear subMDPs, the dimension of the feature space $d_\psi$ is upper-bounded by the cardinality of the image of the (linear) transition mapping, i.e., $d_\psi \leqslant \max_n |\mathcal{S}^{(n)} \cup \mathcal{E}^{(n)}| = M$.

When the aggregated state space is just a subset of the original state space with $N = 1$, implying the aggregated state space is just the original state space $\mathcal{S}$, this model reduces to classical non-hierarchical linear MDPs Jiang et al. (2017); Jin et al. (2020). Furthermore, when the feature representation is a one-hot encoding, i.e., $d_\psi = \bar{S}A$, this model corresponds to tabular MDPs with equivalence mappings between subMDPs, as indroduced by Wen et al. (2020). Thus, this model generalizes tabular MDPs with the hierarchical structure as well as non-hierarchical linear MDPs.

Thanks to dynamics aggregation, we only need to learn $\{\boldsymbol{\mu}_h^{(n)}\}_{n=1}^N$ and can reuse them to solve similar sub-problems, highlighting the reusability as a key advantage of this approach.

## 6 Algorithm: UC-HRL

The purpose of the algorithm is to learn the transitions for each aggregated subMDP, denoted by $\mathcal{M}^{(n)}(\mathcal{S}^{(n)} \cup \mathcal{E}^{(n)}, \mathcal{A}, \{\mathbb{P}_h^{(n)}\}_{h=1}^H, \{r_h^{(n)}\}_{h=1}^H, \mathcal{E}^{(n)})$. Let $\psi_h^{i \to (n)} : \mathcal{S}^i \cup \mathcal{E}^i \to \mathcal{S}^{(n)} \cup \mathcal{E}^{(n)}$ are **known** dynamics aggregations. To simplify the presentation, we denote $\bar{s} = \psi_h^{i \to (n)}(s)$. The indices $i$ and $n$ can be abbreviated as they are determined by the state $s$. Specifically, $i$ represents the index of the current subMDP to which the state $s$ belongs, while $n$ denotes the index of the aggregated subMDP that the current subMDP ($i$) is mapped to via an aggregation mapping. We can learn each transition $\mathbb{P}_h^{(n)}(\cdot \mid \bar{s}, a) = \phi(\bar{s}, a)^\top \boldsymbol{\mu}_h^{(n)}$ by approximating $\boldsymbol{\mu}_h^{(n)}$ using data that has been collected so far. Denote $\boldsymbol{\delta}(\bar{s}) \in \mathbb{R}^{\bar{S}}$ as a one-hot vector that has zero everywhere except that the entry corresponding to $\bar{s}$ is one. For episode $k \leqslant K$ and horizon $h \leqslant H$, let $\boldsymbol{\epsilon}_{k,h}^{(n)} := \mathbb{P}_h^{(n)}(\cdot \mid \bar{s}_{k,h}, a_{k,h})^\top - \boldsymbol{\delta}(\bar{s}_{k,h+1})$. Then, conditioned on history $\mathcal{H}_{k,h}$, all information from the beginning of the learning process up to and including $(\bar{s}_{k,h}, a_{k,h})$, we have $\mathbb{E}[\boldsymbol{\epsilon}_{k,h}^{(n)} \mid \mathcal{H}_{k,h}] = 0$ for $n \in [N]$. This implies that $\boldsymbol{\delta}(\bar{s}_{k,h+1})$ is an unbiased estimate of $\mathbb{P}_h^{(n)}(\cdot \mid \bar{s}_{k,h}, a_{k,h})^\top$ conditioned on $(\bar{s}_{k,h}, a_{k,h})$. Define a collection of $(\bar{s}, a, \bar{s}')$ triplet interacted with any aggregated subMDP $\mathcal{M}^{(n)}$ until the end of episode $k - 1$ as

$$\mathcal{D}_{k,h}^{(n)} := \left\{ (\bar{s}_{k',h}, a_{k',h}, \bar{s}_{k',h+1}) : s_{k',h} \in \mathcal{S}^i, \bar{s}_{k',h} = \psi_h^{i \to (n)}(s_{k',h}) \right\}_{k'=1}^{k-1} \tag{1}$$

Then, for all $n \in [N]$, it is reasonable to learn $\boldsymbol{\mu}_h^{(n)}$ via the following ridge linear regression:

$$\widehat{\boldsymbol{\mu}}_{k,h}^{(n)} = \underset{\boldsymbol{\mu}}{\arg\min} \sum_{(\bar{s}, a, \bar{s}') \in \mathcal{D}_{k,h}^{(n)}} \left\| \phi(\bar{s}, a)^\top \boldsymbol{\mu} - \boldsymbol{\delta}(\bar{s}')^\top \right\|_2^2 + \lambda \|\boldsymbol{\mu}\|_F^2.$$

---

[3]Note that we do not lose generality since learning $r$ is much easier than learning $P$. This assumption regarding $r$ is typical in the literature on model-based RL (Yang & Wang, 2019; 2020; Ayoub et al., 2020; Zhou et al., 2021a).

---

**Algorithm 1** Upper Confidence Hierarchical RL with Transition-Targeted Regression (UC-HRL)

---

1: **Inputs:** $\mathcal{M}$, $K$, $\phi$, $N$, $\psi_h^{i\to(n)}$, $\beta$, $\lambda$
2: **Initialize:** $\Lambda_{1,h}^{(n)} = \lambda I \in \mathbb{R}^{d_\psi \times d_\psi}$, $\widehat{\boldsymbol{\mu}}_{1,h}^{(n)} = \mathbf{0} \in \mathbb{R}^{d_\psi \times S}$, $\mathcal{D}_{k,h}^{(n)} = \varnothing$.
3: **for** episode $k = 1, 2, \cdots, K$ **do**
4:      Set $\{\widehat{Q}_{k,h}^{\psi(i)}\}_{h=1}^H$ as described in Eq. 2 using $\widehat{\boldsymbol{\mu}}_k^{(n)}$.
5:      **for** horizon $h = 1, 2, \cdots, H$ **do**
6:          $a_{k,h} \leftarrow \arg\max_{a \in \mathcal{A}} \widehat{Q}_{k,h}^{\psi(i)}(\psi_h^{i\to(n)}(s_{k,h}), a)$, where $s_{k,h} \in \mathcal{S}^i$ and $\exists \psi_h^{i\to(n)}(s_{k,h})$.
7:          Play an action $a_{k,h}$ and observe $s_{k,h+1}$.
8:      **end for**
9:      Update $\mathcal{D}_{k+1,h}^{(n)}$ by Eq. 1.
10:      $\Lambda_{k+1,h}^{(n)} \leftarrow \lambda I + \sum_{(\bar{s},a,\bar{s}') \in \mathcal{D}_{k+1,h}^{(n)}} \phi(\bar{s}, a)\phi(\bar{s}, a)^\top$.
11:      $\widehat{\boldsymbol{\mu}}_{k+1,h}^{(n)} \leftarrow (\Lambda_{k+1,h}^{(n)})^{-1} \sum_{(\bar{s},a,\bar{s}') \in \mathcal{D}_{k+1,h}^{(n)}} \phi(\bar{s}, a)\boldsymbol{\delta}(\bar{s}')^\top$.
12: **end for**

---

In convention, in case of $\mathcal{D}_{k,h}^{(n)} = \varnothing$, the summation over $\mathcal{D}_{k,h}^{(n)}$ is zero. The full algorithm is summarized in Algorithm 1. For every episode $k$, we form an UCB bonus term $\beta \|\phi(\bar{s}, a)\|_{(\Lambda_{k,h}^{(n)})^{-1}}$.

With that, for $s \in \mathcal{S}$, $a \in \mathcal{A}$ and $h \in [H]$, we construct the optimistic aggregated Q-value functions.

**Definition 5** (Optimistic aggregated Q-values). *For any $(s, a) \in \mathcal{S} \times \mathcal{A}$ and $h \in [H]$, let $s \in \mathcal{S}^i$, $\exists \psi_h^{i\to(n)}$, and $\bar{s} = \psi_h^{i\to(n)}(s)$. Then, for all $i \in [L]$, optimistic aggregated Q-values are defined as:*

$$\widehat{Q}_{k,h}^{\psi(i)}(\bar{s}, a) := \min\left\{ r_h(\bar{s}, a) + \phi(\bar{s}, a)^\top \widehat{\boldsymbol{\mu}}_{k,h}^{(n)} \widehat{V}_{h+1}^{\psi(i)} + \beta \|\phi(\bar{s}, a)\|_{(\Lambda_{k,h}^{(n)})^{-1}}, H \right\}, \tag{2}$$

*where $\widehat{V}_{h+1}^{\psi(i)} \in \mathbb{R}^{\bar{S}}$ such that $\widehat{V}_{h+1}^{\psi(i)}(\psi_h^{i\to(n)}(s'))$ for $s' \in \mathcal{S}^i$, $\widehat{V}_{h+1}^{\psi(j)}(\psi_h^{j\to(n)}(s'))$ for $s' \in \mathcal{E}^i \cap \mathcal{S}^j$, and $0$ for otherwise.*

Note that $\widehat{V}_{k,H+1}^{\psi(i)}(s) := 0$ since the agent obtains no reward after $H$'th step. We also point out that for any states from different subMDPs $s_1 \in \mathcal{S}^i$, $s_2 \in \mathcal{S}^j$ where $\psi_h^{i\to(n)}(s_1) = \psi_h^{j\to(n)}(s_2) = \bar{s}$, the Q-value estimates can have different values, i.e., $\widehat{Q}_{k,h}^{\psi(i)}(\bar{s}) \neq \widehat{Q}_{k,h}^{\psi(j)}(\bar{s})$. Thus, the estimated Q-values in the original state space $\mathcal{S}$ are defined as $\widehat{Q}_{k,h}(s, a) := \widehat{Q}_{k,h}^{\psi(i)}(\psi_h^{i\to(n)}(s), a)$ for $\forall (s, a) \in \mathcal{S}^i \cup \mathcal{E}^i \times \mathcal{A}$. By choosing a proper value for $\beta$, we can prove that, with high probability, the Q-value estimates are always optimistic estimates of the actual Q values. Then, in each $(h, k) \in H \times K$, the agent selects an action that maximized these Q-values $\{\widehat{Q}_{k,h}^{\psi(i)}\}_{h=1}^H$.

## 7 REGRESS ANALYSIS

**Theorem 2** (Regret upper bound). *Let $\pi = \{\pi_k\}_{k=1}^K$ be a collection of policies over $K$ episodes and $s_{k,1}$ be the initial state at episode $k$. Denote $d_\psi$ as the maximum rank of the transition kernels for aggregated subMDPs, and $N$ as the number of aggregated subMDPs. Then, under Assumption 1, there exists an absolute constant $C > 0$ such that, for any fixed $\delta \in (0, 1)$, if we set $\beta = C \cdot d_\psi H \ln(2d_\psi T/\delta)$, then with probability at least $1 - \delta$, the regret of UC-HRL policy $\pi$ is bounded by*

$$\sum_{k=1}^K (V_1^* - V_1^{\pi_k})(s_{k,1}) = \widetilde{\mathcal{O}}\big(d_\psi^{3/2} H^{3/2} \sqrt{NT} + TH\epsilon_p\big).$$

**Discussion of Theorem 2.** Theorem 2 implies that if $\epsilon_p$ is sufficiently small — that is, if the aggregation mapping is precise enough — our algorithm enjoys favorable provable guarantees on regret performances. For example, if $\epsilon_p = \widetilde{\mathcal{O}}(1/\sqrt{T})$, the regret is still bounded by $\widetilde{\mathcal{O}}\big(d_\psi^{3/2} H^{3/2} \sqrt{NT}\big)$. We show that the hierarchical structure can enable statistically more efficient learning compared to preceding algorithms that do not utilize the hierarchical structure. Specifically, if $d_\psi^3 N \ll d^3$,

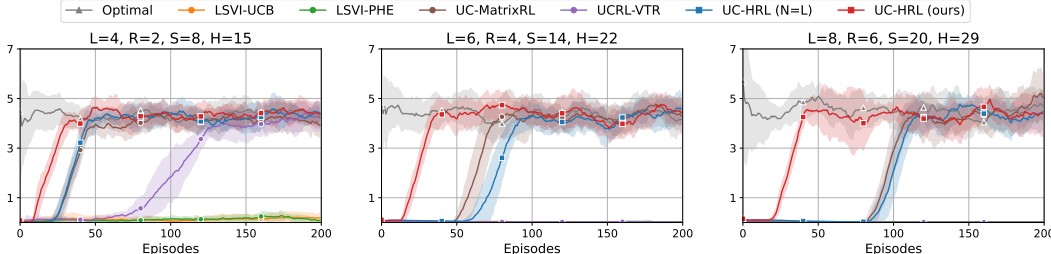

Figure 3: Episodic returns over 10 independent runs under the Block-RiverSwim environment

the regret bound can be significantly improved compared to `LSVI-UCB` (Jin et al., 2020), which has a regret bound of $d^{3/2}H^{3/2}\sqrt{T}$, where $d$ represents the dimension of the feature vector in the original MDP $\mathcal{M}$. Recall Theorem 1 and Corollary 1, which posit that in the majority of real-world environments, particularly those where the maximum number of directly reachable states $U$ is not proportional to the state space size $S$, the dimension of the feature vector $d$ is lower bounded by $S/U$ [4]. It's always the case that $d_\psi \leqslant M$, as we consider low-rank linear subMDPs. Hence, when $MN \ll S$ (indicative of a hierarchical structure), $M^2 \leqslant S^2/U^3$ (signifying a small number of directly reachable states compared to $S$, a common scenario), the inequality $d_\psi^3 N \ll d^3$ can be easily satisfied. We can show this by the following inequality: $d_\psi^3 N \leqslant M^3 N \ll SM^2 \leqslant S^3/U^3 \leqslant d^3$.

# 8 NUMERICAL EXPERIMENTS

We run our numerical experiments on Block-RiverSwim, a variant of RiverSwim (Strehl & Littman, 2008), which repeats the sub-structures called "Block" (refer Appendix H for detailed descriptions). Thus, if the agent can make use of the repeated sub-structures by re-using the learned solution to other blocks, it can learn the optimal policy efficiently.

**Baselines.** We compare our algorithm to other provably efficient RL algorithms with linear function approximation: model-based algorithms such as `UC-MatrixRL` (Yang & Wang, 2020) and `UCRL-VTR` (Ayoub et al., 2020), and model-free algorithms such as `LSVI-UCB` (Jin et al., 2020) and `LSVI-PHE` (Ishfaq et al., 2021). We also included the results of `UC-HRL(N=L)`, which is the variant of `UC-HRL` that naively learns the transition probabilities without the aggregation mappings, in order to directly verify the effect of leveraging hierarchical structure.

**Results.** For a fair comparison, we sweep over the hyper-parameters for each algorithm over certain ranges. Figure 3 depicts learning curves over varying state sizes (and the number of blocks) for `UC-HRL` and other baseline algorithms. When the size of the state space is small and few sub-structures are repeated (e.g., $L = 4, R = 2, S = 8$), our algorithm, as well as other model-based algorithms perform relatively well. However, as the sub-structures repeat more ($R$ increases), our algorithm learns the optimal policy far more quickly than the other algorithms. The results demonstrate that our proposed algorithm is not only provably but also experimentally efficient when the hierarchical structure is presented in the environment.

# 9 CONCLUSION

In this work, we first show that in the majority of real-world environments, the regret can be dependent on the size of the state space $S$ by showing that the dimension of features, $d$, can be proportional to $S$. To mitigate this issue, we formalize a hierarchical decomposition in aggregated state space and propose a `UC-HRL` that can significantly enhance the regret bound if repeated sub-structures are present. However, utilizing a known hierarchical structure is not the sole solution. We leave the exploration of other milder methods as a direction for future research. We anticipate that our research will serve as a pioneering study in rigorously highlighting the limitations of linear models and in enhancing the understanding of provably efficient hierarchical RL with function approximation.

---

[4]For the Euclidean continuous state space, $d \gtrsim \mathrm{Vol}(\mathcal{S})/\mathrm{Vol}(\mathcal{U})$.

ACKNOWLEDGEMENTS

This work was supported by the National Research Foundation of Korea(NRF) grant funded by the Korea government(MSIT) (No. 2022R1C1C1006859 and RS-2023-00222663) and by Creative-Pioneering Researchers Program and AI-Bio Research Grant through Seoul National University.

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

## A    RELATED WORK

In this section, we provide the expanded version of Section 2.

**Reinforcement Learning with Linear Function Approximation.** In recent years, there has been a surge in research on function approximation with provable guarantees (Jiang et al., 2017; Yang & Wang, 2019; 2020; Jin et al., 2020; Zanette et al., 2020; Modi et al., 2020; Du et al., 2020; Cai et al., 2020; Ayoub et al., 2020; Wang et al., 2020; Weisz et al., 2021; He et al., 2021; Zhou et al., 2021a;b; Ishfaq et al., 2021). All these works assume certain linear structures of underlying MDP. Jin et al. (2020) examined linear MDPs and proposed LSVI-UCB, which achieves an $\widetilde{\mathcal{O}}(d^{3/2}H^{3/2}\sqrt{T})$ regret bound. Zanette et al. (2020) proposed an optimistically initialized variant of the randomized least-squares value iteration (RLSVI) algorithm, where exploration is induced by perturbing the estimates of the action-value functions and provided a regret bound $\widetilde{\mathcal{O}}(d^2H^2\sqrt{T})$.

In the realm of model-based algorithms with linear function approximation, Yang & Wang (2020) proposed a model-based algorithm under the assumption that the transition probability kernel is bilinearly parameterized by two feature mappings, enjoying a regret of $\widetilde{\mathcal{O}}(d^{3/2}H^2\sqrt{T})$. Another popular MDP model for RL with linear function approximation is linear mixture model (Modi et al., 2020; Jia et al., 2020; Ayoub et al., 2020), where the transition probability kernel is a linear mixture of some basis kernels. Jia et al. (2020) proposed the UCRL-VTR algorithm under the linear mixture model, where the transition probability kernel is a linear mixture of some basis kernels, with $\widetilde{\mathcal{O}}(dH^{3/2}\sqrt{T})$ regret. Zhou et al. (2021a) proposed a variant of the method proposed by Jia et al. (2020) and proved $\widetilde{\mathcal{O}}(dH\sqrt{T})$ regret bound with a matching lower bound $\Omega(dH\sqrt{T})$.

These algorithms seem to cope with large state space, as their regret scales only polynomially in $d$. However, it remains unclear the relationship between the feature dimension $d$ and the state space size $S$ in linear MDPs. What is known is that in the worst case, $d = SA$, a scenario referred to as the tabular MDP (Example 2.1 in Jin et al. (2020)). Moreover, Du et al. (2020) showed that there exists a family of MDPs where the sample complexity can be lower bounded by $S$ even with good representations under linear function approximation. However, they considered only special instances. Furthermore, Hwang & Oh (2022) showed that UC-MatrixRL (Yang & Wang, 2020) can depend on the size of the state space despite the use of function approximation. However, their analysis is specifically applicable to bilinear models and does not include linear MDPs. In Theorem 1, we rigorously examine the relationship between $d$ and $S$ in linear MDPs, establishing that $d$ can be proportional to $S$ in the majority of real-world environments. To the best of our knowledge, this is the first work to demonstrate the conditions under which $d$ is large or small. In Corollary 1, 2, and 3, we establish that $d$ can be proportional to $S$ in the majority of real-world environments. This implies that the low-rank assumption, as suggested in many existing works (Zanette et al., 2020; Uehara et al., 2021; Modi et al., 2024; Huang et al., 2023), often does not hold in the real world.

On the other hand, a key distinction between our algorithm and LSVI-UCB (Jin et al., 2020) lies in our approach being *model-based*. This distinction primarily offers the benefit of *reusability*. In model-free approaches, Q-values are defined by features and specific parameters, as exemplified by $w_h^{\pi}$ in Jin et al. (2020). Therefore, when two different states are mapped to the same aggregated state, they become indistinguishable in terms of their Q-values because they share the same feature. However, it's important to note that the actual Q-values for these two states may still differ (refer Appendix D.3). Essentially, model-free approaches fail to effectively leverage the hierarchical structure, even with perfectly learned sub-structures and accurate mapping. Conversely, our model-based approach enables the reuse of learned dynamics of subMDPs $\hat{\mu}_{k,h}^{(n)}$ in similar subMDPs. Therefore, it's important to note the fundamental intuition and significance of using a model-based approach to exploit the hierarchical structure.

**State Aggregation** The study of state aggregation (or state abstraction) in RL has a long and rich history, dating back to early work on approximating dynamic programs (Fox, 1973; Whitt, 1978; Bean et al., 1987; Bertsekas et al., 1988). The pioneering work of Whitt (1978) laid a critical foundation for comprehending the impact of state aggregation on value loss in MDPs. Building upon this, Dean & Givan (1997) devised a method for identifying states with similar behaviors through the application of the bisimulation property. In a similar vein, Li et al. (2006) introduced a unified framework for state aggregation in MDPs, examining the conditions under which such aggregations can preserve optimal behavior and affect existing convergence guarantees of well-known RL algorithms. More

recent work has continued to clarify the conditions under which state abstractions preserve value in MDPs (Abel et al., 2020). Previous research has also suggested regret or learning time bounds with aggregated states (Wen & Van Roy, 2017; Dong et al., 2019). However, unlike our work, these past studies did not explicitly leverage the hierarchical structure, a key component that our research embraces.

**Hierarchical Reinforcement Learning.** There are several works that addressed the decomposition of MDP into sub-problems (Dean & Lin, 1995; Singh & Cohn, 1997; Meuleau et al., 1998). Meuleau et al. (1998) decomposed the original MDP into subMDPs and then solved independently under the weakly coupled resource constraints. The concept of hierarchical RL, defined as an agent's ability to act and plan at multiple temporal abstraction levels, was established by Sutton et al. (1999); Barto & Mahadevan (2003). However, few works have quantified the benefits of Hierarchical RL (HRL) from a theoretical perspective. For instance, Fruit et al. (2017) introduced semi-MDP versions of exploration-exploitation algorithms, and Mann et al. (2015) proposed an algorithm that quickly learns policies and models in new circumstances by solving smaller problems. The closest related to our work is Wen et al. (2020), which proposed a model-based Thompson sampling-style HRL algorithm that exploits repeating subMDP structures. They provided a regret bound that shows a significant improvement when the maximum size of subMDPs multiplied by the number of equivalent subMDPs is much smaller than the state space size. However, their work only considered tabular MDPs for hierarchical structure utilization, leaving the question of designing a provably efficient HRL algorithm for linear MDPs with sub-structure still unanswered.

# B    ADDITIONAL EXPLANATION FOR SECTION 4

## B.1    PROOF OF THEOREM 1

*Proof of Theorem 1.* For horizon $h \in [H]$, let $\mathbb{P}_h(\cdot \mid \cdot, \cdot) \in \mathbb{R}^{SA \times S}$ be the transition kernel of an MDP $\mathcal{M}$.

We first show that the feature dimension $d$ is greater than or equal to the rank of the transition kernel $\mathrm{rank}(\mathbb{P}_h)$. In the linear MDPs, the transition kernel can be expressed as $\mathbb{P}_h(\cdot \mid \cdot, \cdot) = \Phi^\top \boldsymbol{\mu}_h$, where $\Phi = [\phi(s_1, a_1), \phi(s_1, a_2), \cdots, \phi(s_S, a_A)] \in \mathbb{R}^{d \times SA}$ and $\boldsymbol{\mu}_h = [\mu_h(s_1), \ldots, \mu_h(s_S)] \in \mathbb{R}^{d \times S}$. Thus, $d$ cannot be smaller than $\mathrm{rank}(\mathbb{P}_h)$.

Now, we select any row, which corresponds to a specific state-action pair $(s, a)$, from the transition kernel $\mathbb{P}_h$ and label it as vector $v_1$. By rearranging the columns, we can make $v_1$ have zero probabilities for $U + 1, \ldots, S$-th entries and have non-zero probabilities for at least one of $1, \ldots, U$-th entries. This is possible due to the condition that the maximum number of directly reachable states is less than equal to $U$, i.e., for any $(s, a)$, there are non-zero transition probabilities for transitioning from $(s, a)$ to a subset of at most $U$ states.

Furthermore, given the condition that every state is accessible from at least one other state (refer problem setting in Section 3), there exists a row vector that has non-zero probabilities for at least one of $U + 1, \ldots, S$-th entries. We denote this vector as $v_2$. By reordering the $U + 1, \ldots, S$-th columns, we can make $v_2$ have zero probabilities for $2U + 1, \ldots, S$-th entries and have non-zero probabilities for at least one of $U + 1, \ldots, 2U$-th entries. Consequently, it is evident that $v_1$ and $v_2$ are linearly independent.

Following a similar logic, we can choose a row vector $v_3$ that has zero probabilities for $3U + 1, \ldots, S$-th columns and have non-zero probabilities for at least one of $2U + 1, \ldots, 3U$-th entries, by reordering the columns properly. It is also clear that $v_1$, $v_2$, and $v_3$ are linearly independent from one another.

By recursively applying this logic, we can choose row vectors $v_1, \ldots, v_{\lfloor S/U \rfloor}$ that are linearly independent from one another. This directly implies that $\mathrm{rank}(\mathbb{P}_h) \geqslant \lfloor S/U \rfloor$. Therefore, we derive that

$$d \geqslant \mathrm{rank}(\mathbb{P}_h) \geqslant \lfloor S/U \rfloor.$$

$\square$

## B.2 EXAMPLE FOR THEOREM 1

For a better explanation, we provide an example for Theorem 1. Consider an MDP where $\mathcal{S} = \{s_1, s_2, \ldots, s_9\}$, $\mathcal{A} = \{a\}$, and the transition kernel is as follows:

$$\mathbb{P}_h = \begin{matrix} & \begin{matrix} s_1 & s_2 & s_3 & s_4 & s_5 & s_6 & s_7 & s_8 & s_9 \end{matrix} & \\ \begin{matrix} (s_1, a) \\ (s_2, a) \\ (s_3, a) \\ (s_4, a) \\ (s_5, a) \\ (s_6, a) \\ (s_7, a) \\ (s_8, a) \\ (s_9, a) \end{matrix} & \begin{pmatrix} 1/3 & 1/3 & 1/3 & 0 & 0 & 0 & 0 & 0 & 0 \\ 0 & 0 & 1/2 & 0 & 1/2 & 0 & 0 & 0 & 0 \\ 1/5 & 1/5 & 3/5 & 0 & 0 & 0 & 0 & 0 & 0 \\ 0 & 0 & 0 & 1/4 & 0 & 1/4 & 1/2 & 0 & 0 \\ 0 & 0 & 0 & 0 & 0 & 0 & 0 & 0 & 1 \\ 0 & 1/2 & 1/2 & 0 & 0 & 0 & 0 & 0 & 0 \\ 0 & 0 & 0 & 0 & 0 & 2/3 & 0 & 1/3 & 0 \\ 0 & 0 & 0 & 1/3 & 1/3 & 1/3 & 0 & 0 & 0 \\ 1/2 & 0 & 1/2 & 0 & 0 & 0 & 0 & 0 & 0 \end{pmatrix} & \begin{matrix} \to v_1 \\ \\ \\ \\ \\ \\ \to v_2 \\ \\ \end{matrix} \end{matrix}$$

Note that $U = 3 \leqslant S$ and every state is accessible from at least one other state. First, choose any row of the transition kernel. Assume that we choose the first row and denote it as $v_1$. The row vector $v_1$ has zero probabilities for $4, \ldots, 9$-th entries and has non-zero probabilities for $1, 2, 3$-th entries.

Second, we can choose the seventh row and denote it as $v_2$. By switching the columns corresponding to $s_4$ and $s_8$, we get

$$\mathbb{P}_h = \begin{matrix} & \begin{matrix} s_1 & s_2 & s_3 & s_8 & s_5 & s_6 & s_7 & s_4 & s_9 \end{matrix} & \\ \begin{matrix} (s_1, a) \\ (s_2, a) \\ (s_3, a) \\ (s_4, a) \\ (s_5, a) \\ (s_6, a) \\ (s_7, a) \\ (s_8, a) \\ (s_9, a) \end{matrix} & \begin{pmatrix} 1/3 & 1/3 & 1/3 & 0 & 0 & 0 & 0 & 0 & 0 \\ 0 & 0 & 1/2 & 0 & 1/2 & 0 & 0 & 0 & 0 \\ 1/5 & 1/5 & 3/5 & 0 & 0 & 0 & 0 & 0 & 0 \\ 0 & 0 & 0 & 0 & 0 & 1/4 & 1/2 & 1/4 & 0 \\ 0 & 0 & 0 & 0 & 0 & 0 & 0 & 0 & 1 \\ 0 & 1/2 & 1/2 & 0 & 0 & 0 & 0 & 0 & 0 \\ 0 & 0 & 0 & 1/3 & 0 & 2/3 & 0 & 0 & 0 \\ 0 & 0 & 0 & 0 & 1/3 & 1/3 & 0 & 1/3 & 0 \\ 1/2 & 0 & 1/2 & 0 & 0 & 0 & 0 & 0 & 0 \end{pmatrix} & \begin{matrix} \to v_1 \\ \\ \\ \\ \\ \\ \to v_2 \\ \to v_3 \\ \end{matrix} \end{matrix}$$

Now, the row vector $v_2$ has zero probabilities for $7, 8, 9$-th columns and has non-zero probabilities for at least one of $4, 5, 6$-th columns.

Finally, we choose the eighth row for $v_3$. Since $v_1$, $v_2$ and $v_3$ are linearly independent of one another, the rank of $\mathbb{P}_h$ is lower bounded by $S/U = 3$.

## B.3 PROOF OF COROLLARY 2

*Proof of Corollary 2.* We provide the proof by examining two cases: 1) when the state space is countably infinite, and 2) when it is normed, compact, and uncountably infinite.

**Case 1. countably infinite state space.**
We prove this by contradiction. Assume that the feature dimension $d$ is finite. Then, the transition kernel possesses a finite number of linearly independent rows and columns. Note that for a countably infinite state space, the transition kernel can be viewed as an infinite matrix with both its rows and columns being infinite.

Now, we follow the same logic from the proof of Theorem 1. We choose any row, which corresponds to a specific state-action pair $(s, a)$, from the transition kernel $\mathbb{P}_h$ and label it as vector $v_1$. Since the maximum number of directly reachable states is less than equal to $U$, by rearranging the columns of $\mathbb{P}_h$ properly, we can make $v_1$ have zero probabilities for $U + 1, \ldots$-th entries and have non-zero probabilities for at least one of $1, \ldots, U$-th entries.

Furthermore, given the condition that every state is accessible from at least one other state (refer Section 3), there exists a row vector that has non-zero probabilities for at least one of $U + 1, \ldots,$-th entries. We denote this vector as $v_2$. By reordering the $U + 1, \ldots$-th columns, we can make $v_2$ have zero probabilities for $2U + 1, \ldots$-th entries and have non-zero probabilities for at least one of $U + 1, \ldots, 2U$-th entries. Consequently, it is evident that $v_1$ and $v_2$ are linearly independent.

By recursively applying this logic, we can identify a set of linearly independent row vectors $\Gamma := \{v_1, v_2 \ldots \}$. Since the cardinality of the state space is infinite, the cardinality of the set $\Gamma$ is also infinite. This contradicts our initial assumption that the transition kernel possesses a finite number of linearly independent rows. Therefore, we conclude that $d$ must be infinite.

**Case 2. normed, compact, and uncountably infinite state space.**
For the normed, uncountably infinite (or continuous), and compact state space, we can utilize the $\varepsilon$-covering. Let $\mathcal{N}(\mathcal{S}, \varepsilon, \rho)$ be the the minimal $\varepsilon$-covering number of $\mathcal{S}$ with respect to a distance metric $\rho$. Further, we define $\widetilde{\mathcal{S}} \subseteq \mathcal{S}$ as the set of all center points of the covering balls. Note that $\widetilde{\mathcal{S}}$ is a finite space since the state space is compact and $|\widetilde{\mathcal{S}}| = \mathcal{N}(\mathcal{S}, \varepsilon, \rho)$. Let $\widetilde{\mathcal{Z}} \subseteq \mathcal{S} \times \mathcal{A}$ be the minimum size subset of the state-action space containing every element in $\widetilde{\mathcal{S}}$, i.e., $\widetilde{\mathcal{Z}} \supseteq \widetilde{\mathcal{S}}$, such that every state in $\widetilde{\mathcal{S}}$ is accessible from at least one state-action pair in $\widetilde{\mathcal{Z}}$, i.e., $\sum_{(s,a) \in \widetilde{\mathcal{Z}}} \mathbb{P}_h(s' \mid s, a) > 0$ for all $s' \in \widetilde{\mathcal{S}}$ and $\widetilde{\mathcal{Z}} \supseteq \widetilde{\mathcal{S}}$. Note that $|\widetilde{\mathcal{S}}| \leqslant |\widetilde{\mathcal{Z}}| \leqslant 2|\widetilde{\mathcal{S}}| < \infty$. This is because if $s' \in \widetilde{\mathcal{S}}$ is not accessible from $\widetilde{\mathcal{S}} \times \mathcal{A}$, we only need to add one state-action pair $(s, a)$ to $\widetilde{\mathcal{Z}}$ that can transition to $s' \in \widetilde{\mathcal{S}}$. Hence, every state is accessible from at least one state-action pair in the reduced form of the transition kernel $\widetilde{\mathbb{P}}_h \in \mathbb{R}^{|\widetilde{\mathcal{Z}}| \times |\widetilde{\mathcal{S}}|}$.

Let $\widetilde{U}$ be the maximum number of directly reachable states for $\widetilde{\mathbb{P}}_h$. Then, by Theorem 1, we can identify linearly independent row vectors every $\widetilde{U}$ columns. Therefore, we have that

$$\mathrm{rank}(\widetilde{\mathbb{P}}_h) \geqslant \lfloor |\widetilde{\mathcal{S}}| / \widetilde{U} \rfloor.$$

On the other hand, we know that $d$ cannot be smaller than $\mathrm{rank}(\widetilde{\mathbb{P}}_h)$; thus, we have $d \geqslant \mathrm{rank}(\widetilde{\mathbb{P}}_h)$. Hence, for any $\varepsilon > 0$, we have that

$$d \geqslant \mathrm{rank}(\widetilde{\mathbb{P}}_h) \geqslant \lfloor |\widetilde{\mathcal{S}}| / \widetilde{U} \rfloor \geqslant \lfloor |\widetilde{\mathcal{S}}| / U \rfloor = \lfloor \mathcal{N}(\mathcal{S}, \varepsilon, \rho) / U \rfloor > \mathcal{N}(\mathcal{S}, \varepsilon, \rho) / U - 1,$$

where the third inequality is by the fact that $\widetilde{U} \leqslant U$. Since $\mathcal{N}(\mathcal{S}, \varepsilon, \rho)$ is non-decreasing as $\varepsilon$ goes to zero, we get

$$\sup_{\varepsilon > 0} \mathcal{N}(\mathcal{S}, \varepsilon, \rho) / U = \lim_{\varepsilon \to 0} \mathcal{N}(\mathcal{S}, \varepsilon, \rho) / U = \infty,$$

where the last inequality holds since $U$ is finite. Therefore, $d$ is infinite. $\qquad \square$

### B.4 PROOF OF COROLLARY 3

*Proof of Corollary 3.* For the $p$-dimensional Euclidean state space $\mathcal{S} \subset \mathbb{R}^p$, we can use the $\varepsilon$-covering method. Let $\mathcal{U}$ represent the set of directly reachable states with the maximum volume, i.e., $\mathcal{U} := \mathcal{S}_{\bar{s}, \bar{a}}$, where $(\bar{s}, \bar{a}) = \arg\max_{(s,a) \in \mathcal{S} \times \mathcal{A}} \mathrm{Vol}(\mathcal{S}_{s,a})$. Then, we denote the minimal $\varepsilon$-covering number of $\mathcal{U}$ with respect to a distance metric $\rho$ as $\mathcal{N}(\mathcal{U}, \varepsilon, \rho)$ and define $\widetilde{\mathcal{U}}(\varepsilon)$ as the sets of all center points of the $\varepsilon$-covering balls for $\mathcal{U}$. We further denote the minimal $\varepsilon$-covering number of $\mathcal{X} := \mathcal{S} \backslash \mathcal{U} = \mathcal{S} \cup \mathcal{U}^c$ with respect to a distance metric $\rho$ as $\mathcal{N}(\mathcal{X}, \varepsilon, \rho)$ and define $\widetilde{\mathcal{X}}(\varepsilon)$ as the sets of all center points of the $\varepsilon$-covering balls for $\mathcal{X}$. Note that $|\widetilde{\mathcal{U}}(\varepsilon)| = \mathcal{N}(\mathcal{U}, \varepsilon, \rho) < \infty$, $|\widetilde{\mathcal{X}}(\varepsilon)| = \mathcal{N}(\mathcal{X}, \varepsilon, \rho) < \infty$, and $\widetilde{\mathcal{U}}(\varepsilon) \cap \widetilde{\mathcal{X}}(\varepsilon) = \varnothing$ since the center points of the $\varepsilon$-covering balls are interior points of the corresponding set.

Now we denote $\widetilde{\mathcal{W}}(\varepsilon) = \widetilde{\mathcal{U}}(\varepsilon) \cup \widetilde{\mathcal{X}}(\varepsilon)$. Let $\widetilde{\mathcal{Z}}(\varepsilon) \subseteq \mathcal{S} \times \mathcal{A}$ be the minimum size subset of the state-action space containing every element in $\widetilde{\mathcal{W}}(\varepsilon)$, i.e., $\widetilde{\mathcal{Z}}(\varepsilon) \supseteq \widetilde{\mathcal{W}}(\varepsilon)$, such that every state in $\widetilde{\mathcal{W}}(\varepsilon)$ is accessible from at least one state-action pair in $\widetilde{\mathcal{Z}}(\varepsilon)$, i.e., $\sum_{(s,a) \in \widetilde{\mathcal{Z}}(\varepsilon)} \mathbb{P}_h(s' \mid s, a) > 0$ for all $s' \in \widetilde{\mathcal{S}}(\varepsilon)$. Note that $|\widetilde{\mathcal{W}}(\varepsilon)| \leqslant |\widetilde{\mathcal{Z}}(\varepsilon)| \leqslant 2|\widetilde{\mathcal{W}}(\varepsilon)| < \infty$. This is because if $s' \in \widetilde{\mathcal{W}}(\varepsilon)$ is not accessible from $\widetilde{\mathcal{W}}(\varepsilon) \times \mathcal{A}$, we only need to add one state-action pair $(s, a)$ to $\widetilde{\mathcal{Z}}(\varepsilon)$ that can transition to $s' \in \widetilde{\mathcal{W}}(\varepsilon)$. Hence, every state is accessible from at least one state-action pair in the reduced form of the transition kernel $\widetilde{\mathbb{P}}_h(\varepsilon) \in \mathbb{R}^{|\widetilde{\mathcal{Z}}(\varepsilon)| \times |\widetilde{\mathcal{W}}(\varepsilon)|}$.

Note that $\widetilde{\mathcal{U}}(\varepsilon)$ is the set of directly reachable states for $\widetilde{\mathbb{P}}_h(\varepsilon)$. By applying Theorem 1, we can identify linearly independent row vectors every $|\widetilde{\mathcal{U}}(\varepsilon)|$ columns. Hence, for all $\varepsilon > 0$, we have

$$\mathrm{rank}(\widetilde{\mathbb{P}}_h(\varepsilon)) \geqslant \left\lfloor \frac{|\widetilde{\mathcal{W}}(\varepsilon)|}{|\widetilde{\mathcal{U}}(\varepsilon)|} \right\rfloor. \tag{B.1}$$

Moreover, by the definition of covering number, we know that

$$\mathrm{Vol}(\mathcal{S}) \leqslant \mathrm{Vol}\,(\varepsilon B) \cdot \mathcal{N}(\mathcal{S}, \varepsilon, \rho) \leqslant \mathrm{Vol}\,(\varepsilon B) \cdot |\widetilde{\mathcal{W}}(\varepsilon)|, \tag{B.2}$$

where the operator $\mathrm{Vol}(\cdot)$ represents a volume of a set, $B$ is the unit ball norm; thus $\varepsilon B$ denotes the norm ball with radius $\varepsilon$, $\mathcal{N}(\mathcal{S}, \varepsilon, \rho)$ is the minimal $\varepsilon$-covering number of $\mathcal{S}$ with respect to a distance metric $\rho$, and the last inequality holds since $\mathcal{N}(\mathcal{S}, \varepsilon, \rho) \leqslant \mathcal{N}(\mathcal{U}, \varepsilon, \rho) + \mathcal{N}(\mathcal{X}, \varepsilon, \rho) = |\widetilde{\mathcal{U}}(\varepsilon)| + |\widetilde{\mathcal{X}}(\varepsilon)| = |\widetilde{\mathcal{W}}(\varepsilon)|$.

Furthermore, we have

$$|\widetilde{\mathcal{U}}(\varepsilon)| = \mathcal{N}(\mathcal{U}, \varepsilon, \rho) \leqslant \mathcal{T}(\mathcal{U}, \varepsilon, \rho) \leqslant \frac{\mathrm{Vol}(\mathcal{U} + \frac{\varepsilon}{2}B)}{\mathrm{Vol}(\frac{\varepsilon}{2}B)}, \tag{B.3}$$

where $+$ is the Minkowski addition, $\mathcal{T}(\mathcal{U}, \varepsilon, \rho)$ is the maximal $\varepsilon$-packing number with respect to a distance metric $\rho$, the first inequality holds by Lemma G.3, and the last inequality holds by the definition of $\varepsilon$-packing; $\varepsilon/2$ packing norm balls are disjoint.

On the other hand, we know that $d$ cannot be smaller than $\mathrm{rank}(\widetilde{\mathbb{P}}_h(\varepsilon))$; thus, we have

$$d \geqslant \mathrm{rank}(\widetilde{\mathbb{P}}_h(\varepsilon)). \tag{B.4}$$

Combining Eq. B.1, B.2, B.3, and B.4, for all $\varepsilon > 0$, we get

$$d \geqslant \mathrm{rank}(\widetilde{\mathbb{P}}_h(\varepsilon)) \geqslant \left\lfloor \frac{|\widetilde{\mathcal{W}}(\varepsilon)|}{|\widetilde{\mathcal{U}}(\varepsilon)|} \right\rfloor \geqslant \left\lfloor \frac{\mathrm{Vol}(\mathcal{S})/\mathrm{Vol}(\varepsilon B)}{|\widetilde{\mathcal{U}}(\varepsilon)|} \right\rfloor = \left\lfloor \frac{\mathrm{Vol}(\mathcal{S})/\mathrm{Vol}(B(\varepsilon))}{\mathrm{Vol}(\mathcal{U} + \frac{\varepsilon}{2}B)/\mathrm{Vol}(\frac{\varepsilon}{2}B)} \right\rfloor$$
$$= \left\lfloor 2^p \cdot \frac{\mathrm{Vol}(\mathcal{S})}{\mathrm{Vol}(\mathcal{U} + \frac{\varepsilon}{2}B)} \right\rfloor > 2^p \cdot \frac{\mathrm{Vol}(\mathcal{S})}{\mathrm{Vol}(\mathcal{U} + \frac{\varepsilon}{2}B)} - 1.$$

Hence, by taking supremums over $\varepsilon > 0$, we derive that

$$d > 2^p \cdot \sup_{\varepsilon > 0} \frac{\mathrm{Vol}(\mathcal{S})}{\mathrm{Vol}(\mathcal{U} + \frac{\varepsilon}{2}B)} - 1 = 2^p \cdot \lim_{\varepsilon \to 0} \frac{\mathrm{Vol}(\mathcal{S})}{\mathrm{Vol}(\mathcal{U} + \frac{\varepsilon}{2}B)} - 1$$
$$= 2^p \cdot \frac{\mathrm{Vol}(\mathcal{S})}{\mathrm{Vol}(\mathcal{U})} - 1,$$

where the first equality is by the fact that $\mathrm{Vol}(\mathcal{U} + \frac{\varepsilon}{2}B)$ is non-increasing as $\varepsilon$ decreases. This concludes the proof. $\qquad\square$

## C  EXAMPLE ENVIRONMENT: RIVERSWIM

In this section, we provide a toy example to offer an in-depth explanation of the main point of our paper. First, we validate Theorem 1 in this environment. Subsequently, we demonstrate the benefits of using a hierarchical structure by showing that $d_\psi^3 N \ll d^3$.

The RiverSwim, as illustrated in Figure C.1, is comprised of $S$ states arranged sequentially, with each edge value representing the transition probability. From the starting point at the far left state, noted as $s_1$, the agent has the option to navigate to the left — an action represented by the dashed lines — thereby earning a small reward, i.e., $r(s_1, \texttt{left}) = 0.005$. Alternatively, the agent can opt to move to the right — an action depicted by the *solid* lines — in each successive state. The objective for the agent is to maximize its total return by attempting to arrive at the far right state, denoted as $s_S$. Here, the agent can earn a large reward $r(s_S, \texttt{right}) = 1$ by choosing to navigate to the right.

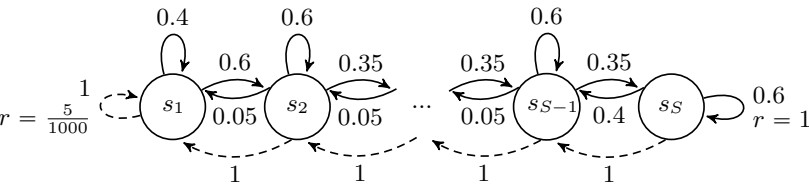

Figure C.1: The "RiverSwim" environment Osband et al. (2013)

The transition kernel of the environment is as follows:

$$
\mathbb{P}_h = \begin{pmatrix}
\mathbb{P}_h(s_1 \mid s_1, \texttt{left}) & \mathbb{P}_h(s_2 \mid s_1, \texttt{left}) & \mathbb{P}_h(s_3 \mid s_1, \texttt{left}) & \mathbb{P}_h(s_4 \mid s_1, \texttt{left}) & \ldots \\
\mathbb{P}_h(s_1 \mid s_1, \texttt{right}) & \mathbb{P}_h(s_2 \mid s_1, \texttt{right}) & \mathbb{P}_h(s_3 \mid s_1, \texttt{right}) & \mathbb{P}_h(s_4 \mid s_1, \texttt{right}) & \ldots \\
\mathbb{P}_h(s_1 \mid s_2, \texttt{left}) & \mathbb{P}_h(s_2 \mid s_2, \texttt{left}) & \mathbb{P}_h(s_3 \mid s_2, \texttt{left}) & \mathbb{P}_h(s_4 \mid s_2, \texttt{left}) & \ldots \\
\mathbb{P}_h(s_1 \mid s_2, \texttt{right}) & \mathbb{P}_h(s_2 \mid s_2, \texttt{right}) & \mathbb{P}_h(s_3 \mid s_2, \texttt{right}) & \mathbb{P}_h(s_4 \mid s_2, \texttt{right}) & \ldots \\
\mathbb{P}_h(s_1 \mid s_3, \texttt{left}) & \mathbb{P}_h(s_2 \mid s_3, \texttt{left}) & \mathbb{P}_h(s_3 \mid s_3, \texttt{left}) & \mathbb{P}_h(s_4 \mid s_3, \texttt{left}) & \ldots \\
\mathbb{P}_h(s_1 \mid s_3, \texttt{right}) & \mathbb{P}_h(s_2 \mid s_3, \texttt{right}) & \mathbb{P}_h(s_3 \mid s_3, \texttt{right}) & \mathbb{P}_h(s_4 \mid s_3, \texttt{right}) & \ldots \\
& \ldots & \ldots & \ldots & \ldots
\end{pmatrix}
$$

$$
= \begin{pmatrix}
1 & 0 & 0 & 0 & \ldots \\
0.4 & 0.6 & 0 & 0 & \ldots \\
1 & 0 & 0 & 0 & \ldots \\
0.05 & 0.6 & 0.35 & 0 & \ldots \\
0 & 1 & 0 & 0 & \ldots \\
0 & 0.05 & 0.6 & 0.35 & \ldots \\
\ldots & \ldots & \ldots & \ldots & \ldots
\end{pmatrix} \tag{C.1}
$$

The agent is allowed to move left, right, or stay in the same state. This means the maximum number of directly reachable states is 3, i.e., $U = 3$. Regardless of an increase in the total number of states, $U$ stays fixed at 3. Consequently, even in this simple example, we can't sidestep a feature dimension that depends on $S$, which means the learning complexity will proportionally increase with $S$, even when using linear function approximation.

We will now demonstrate, using the toy example, how our main claim presented in Section 7 – specifically that $d_\psi^3 N \ll d^3$ – is valid. Let the number of state is 100, i.e., $S = 100$. In this environment, each of the 100 states can be considered as a separate subMDP, and consequently, the number of subMDPs is equal to the size of the state space, i.e., $L = S = 100$. Then, we have

- The maximum size of aggregated subMDPs, $M = 3$: One internal state and its neighboring states.
- The number of aggregated subMDPs, $N = 3$: All subMDPs, excluding $s_1$ and $s_{100}$, which means $s_2, s_3, \ldots, s_{99}$, are grouped into a single aggregated subMDP. Meanwhile, $s_1$ and $s_{100}$ each form distinct aggregated subMDPs.
- $MN = 3^2 \ll S$.
- The maximum size of directly reachable states, $U = 3$.
- The feature dimension of aggregated subMDPs, $d_\psi \leqslant M = 3$: This always holds true.
- The feature dimension of the original MDP, $d = S = 100$: The transition kernel has full rank.

Hence, we get

- $d_\psi^3 N \leqslant 3^4$.
- $M^3 N = 3^4$.

- $SM^2 = 100 \cdot 3^2$.
- $S^3 / U^3 = 100^3 / 3$.

Therefore, we derive that $d_\psi^3 N \leqslant M^3 N \ll SM^2 < S^3 / U^3 < d^3$.

# D    DYNAMICS AGGREGATION

## D.1    COMPARISON TO EXISTING FRAMEWORKS

In this section, we provide a comprehensive description of our proposed framework described in Section 5.

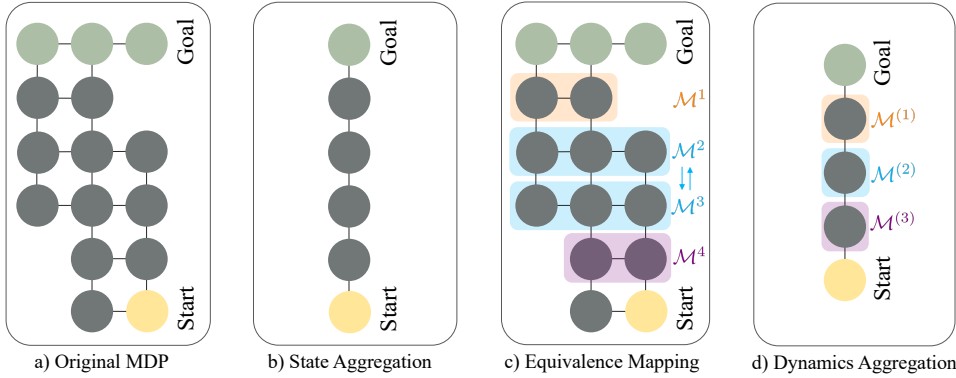

Figure D.1: (a) A simple Gridworld MDP, (b) the aggregated MDP induced by the state aggregation, (c) the partitioned subMDPs with equivalence mappings, and (d) the aggregated MDP induced by the dynamics aggregation

As a motivating example, an agent is placed in a wide hallway, and tasked with reaching an exit located at the far end of the hall (see Figure D.1). The traditional problem setup might use a Cartesian grid, where the agent's location is determined by $x$ and $y$ coordinates, and it navigates using up, down, left, and right actions. Please note that the reduced number of states (circles) in Figure D.1 does not imply a change in the actual MDP structure. Instead, it suggests a simplification in terms of learning complexity.

A careful analysis of the problem reveals that the agent's $x$ coordinate is irrelevant for achieving optimal behavior. Therefore, a state aggregation in this context would be a function that projects the original state to an aggregated state that only tracks the $y$ coordinate. This way, all original states with the same $x$ coordinate belong to the same abstract state, significantly reducing the size of the state space. To intuitively grasp this concept, consider the scenario where you are walking down a hallway without obstacles: if your objective is truly to make forward progress toward the exit of the hall, then there is little need to pay attention to horizontal movement.

In contrast to state aggregation, equivalence mapping (Wen et al., 2020) does not project states into a latent space. Instead, it partitions the original MDPs into disjoint subMDPs and groups them based on identical or similar structures. In the example, the states in the third row (subMDP $\mathcal{M}^2$) and the fourth row (subMDP $\mathcal{M}^3$) have the same transitions and reward functions, and therefore, an equivalence mapping can be established between them. As these subMDPs are indistinguishable in terms of their structures, an agent employing a hierarchical structure can treat them as a single subMDP. This also significantly reduces the size of the state space. However, equivalence mapping necessitates that every aspect of the states is identical, even when certain details may not be crucial for discovering the optimal policy. This requirement can potentially constrain the range of scenarios where this framework can be effectively applied.

Our framework, dynamics aggregation, combines the benefits of both state aggregation and equivalence mapping. It operates by partitioning the original MDPs into subMDPs. These subMDPs are then projected into aggregated subMDPs that focus solely on the $y$ coordinate: the subMDP $\mathcal{M}^1$ into $\mathcal{M}^{(1)}$, the subMDPs $\mathcal{M}^2, \mathcal{M}^3$ into $\mathcal{M}^{(2)}$ (their dynamics are the same), and the subMDP $\mathcal{M}^4$ into $\mathcal{M}^{(3)}$. Notably, our framework is explicitly designed to account for repeating structures in the data, enhancing its effectiveness. This approach results in a considerably simplified representation compared to the other two frameworks. Moreover, dynamics aggregation only demands that the

aggregated states are identical, not minute details. This broadens its applicability compared to equivalence mapping, making it a more versatile framework for a wide range of scenarios.

### D.2 Intuition behind the Dynamics Aggregation

Dynamics aggregation has practical implications for human learning. For instance, a driver who can navigate New York is likely not only to adapt quickly to driving in San Francisco but also to have a high probability of being able to drive in Paris. Similarly, a driver who has practiced exclusively within Manhattan and possesses a driver's license should have no trouble driving to JFK in Queens. This adaptability largely stems from the existence of *repeating structures* and the *mappings* between similar structures. Often, such mappings are readily available, and we humans excel at leveraging them. These mappings need not be perfect; approximations are sufficient, as discussed in this paper. We incorporate these insights into our proposed framework.

### D.3 Comparison to Misspecified Linear MDP

At first glance, one might think it feasible to learn directly from an aggregated MDP (which includes a misspecification error) without employing Dynamics Aggregation, since the misspecified Linear MDP (Jin et al., 2020) also considers the features with a misspecified transition error. However, this approach is *not feasible*. The critical issue is that the Q-value can differ for two different states, even if they are mapped to the same aggregated (or latent) state. This discrepancy renders direct learning from such an aggregated MDP impossible. Let's provide an example:

*Consider an MDP with $\mathcal{S} = \{s_1, s_2\}$, $\mathcal{A} = \{a\}$, and $r(s_1, a) = 1$ and $r(s_2, a) = 0$. Let the horizon length $H$ be 1. And assume that $s_1$ and $s_2$ are exactly mapped into one aggregated state $\bar{s}_1$. Then, it is clear that $Q(s_1, a) = 1$ and $Q(s_2, a) = 0$ since $H = 1$.*

Therefore, if we consider only the aggregated MDP, we cannot differentiate between $Q(s_1, a)$ and $Q(s_2, a)$. Note that in our definition of Q-values in Definition 5, we distinctly define the aggregated Q-values depending on the subMDP to which the state $s$ belongs. Hence, they have the subMDP index $i$.

## E  Notation

We offer a Table E.1 for convenient reference.

We denote $H$ as the episode length, $K$ as the total number of episodes, and $T = HK$ as the time elapsed. We denote $k \in [K]$ as the current episode and $h \in [H]$ as the current horizon step. We use subscript $k, h$ to indicate the quantity at horizon step $h$ in episode $k$. We denote $L$ as the number of subMDPs induced by Definition 3. We denote $N$ as the number of aggregated subMDPs defined by Definition 4 and $M$ as the maximum size of the state space of aggregated subMDPs.

For all $s \in \mathcal{S}$, we can always specify one corresponding internal state set $\mathcal{S}^{(n)}$ of the aggregated subMDPs $\mathcal{M}^{(n)}$, because the entire state space is divided into disjoint subMDPs and each subMDP $i$ has an unique corresponding aggregated subMDPs $\mathcal{M}^{(n)}$. Therefore, for all $s \in \mathcal{S}$, if $s \in \mathcal{S}^i$ and $\exists \psi_h^{i \to (n)}(s)$, we can simply denote $\bar{s} = \psi_h^{i \to (n)}(s)$. We can abbreviate $i$ or $n$ index, because those indexes are determined by the state $s$. In this section, for the sake of simplicity, we will use $d$ to denote the feature dimension of the aggregated state space, i.e., $d = d_\psi$.

## F  Proof of Theorem 2

In this section, we prove the regret bound in Theorem 2. We begin by propose a useful lemma:

**Lemma F.1.** *For any $(s, a) \in \mathcal{S} \times \mathcal{A}$ and $h \in [H]$, let $s \in \mathcal{S}^i$, $\exists \psi_h^{i \to (n)}(s)$, and $\bar{s} = \psi_h^{i \to (n)}(s)$. Given any distributions $p^{(n) \to i}(\cdot \mid \bar{s}) \in \Delta(\mathcal{S}^i)$, define $\mathcal{M}^{(n)}(\mathcal{S}^{(n)} \cup \mathcal{E}^{(n)}, \mathcal{A}, \mathbb{P}_h^{(n)}, r_h^{(n)}, \mathcal{E}^{(n)})$ for $n \in [N]$, where $r_h^{(n)}(\bar{s}, a) = \mathbb{E}_{s \sim p^{(n) \to i}(\cdot \mid \bar{s})}[r_h^i(s, a)]$, and $\mathbb{P}_h^{(n)}(\bar{s}' \mid \bar{s}, a) = \mathbb{E}_{s \sim p^{(n) \to i}(\cdot \mid \bar{s})}[\mathbb{P}_h^i \Psi_h^{i \to (n)}(\bar{s}' \mid s, a)]$. Then, we have*

$$|r_h^{(n)}(\bar{s}, a) - r_h^i(s, a)| \leqslant \epsilon_r, \quad \|\mathbb{P}_h^{(n)}(\cdot \mid \bar{s}, a) - \mathbb{P}_h^i \Psi_h^{i \to (n)}(\cdot \mid s, a)\|_1 \leqslant \epsilon_p.$$

Table E.1: Symbols

| | |
|---|---|
| $s_{k,h}$ | state encountered in horizon $h$ of episode $k$ |
| $a_{k,h}$ | action taken by the algorithm in horizon $h$ of episode $k$ |
| $\psi_h^{i\to(n)}$ | known aggregation mapping from $\mathcal{S}^i \cup \mathcal{E}^i$ to $\mathcal{S}^{(n)} \cup \mathcal{E}^{(n)}$ in horizon $h$ |
| $\Psi_h^{i\to(n)}$ | aggregation mapping kernel satisfying |
| | $\Psi_h^{i\to(n)}(s', \bar{s}') = \mathbb{I}\left(s' \in \mathcal{S}^i \cup \mathcal{E}^i, \bar{s}' \in \mathcal{S}^{(n)} \cup \mathcal{E}^{(n)}, \psi_h^{i\to(n)}(s') = \bar{s}'\right)$ |
| $\mathcal{M}^{(n)}$ | aggregated subMDPs induced by $\psi_h^{i\to(n)}$ |
| $\bar{s}_{k,h}$ | $= \psi_h^{i\to(n)}(s_{k,h})$ where $s_{k,h} \in \mathcal{S}^i$ and $\exists \psi_h^{i\to(n)}(s)$ |
| $\boldsymbol{\delta}(\bar{s})$ | $\bar{S}$-dimensional one-hot vector where the entry corresponding to $\bar{s}$ is one |
| $\boldsymbol{\epsilon}_h^{(n)}(\bar{s}_{k,h}, a_{k,h}, \bar{s}_{k,h+1})$ | $= \mathbb{P}_h^{(n)}\left(\cdot \mid \bar{s}_{k,h}, a_{k,h}\right)^\top - \boldsymbol{\delta}\left(\bar{s}_{k,h+1}\right)$ |
| $\mathcal{D}_{k,h}^{(n)}$ | $= \left\{\left(\bar{s}_{k',h}, a_{k',h}, \bar{s}_{k',h+1}\right) : s_{k',h} \in \mathcal{S}^i, \bar{s}_{k',h} = \psi_h^{i\to(n)}(s_{k',h})\right\}_{k'=1}^{k-1}$ |
| $\lambda$ | regularization parameter |
| $\Lambda_{k,h}^{(n)}$ | $= \lambda I + \sum_{(\bar{s},a,\bar{s}')\in\mathcal{D}_{k,h}^{(n)}} \phi(\bar{s},a)\phi(\bar{s},a)^\top$ |
| $\widehat{\boldsymbol{\mu}}_{k,h}^{(n)}$ | $= \left(\Lambda_{k,h}^{(n)}\right)^{-1} \sum_{(\bar{s},a,\bar{s}')\in\mathcal{D}_{k,h}^{(n)}} \phi(\bar{s},a)\boldsymbol{\delta}(\bar{s}')^\top,$ |
| $\widehat{\mathbb{P}}_{k,h}^{(n)}(\cdot \mid \bar{s},a)$ | $= \phi(\bar{s},a)^\top \widehat{\boldsymbol{\mu}}_{k,h}^{(n)}$ |
| $d$ | $= d_\psi$, the feature dimension of the aggregated state space |
| $\widehat{Q}_{k,h}^{\psi(i)}(\bar{s},a)$ | $= \min\left\{r_h(\bar{s},a) + \phi(\bar{s},a)^\top\widehat{\boldsymbol{\mu}}_{k,h}^{(n)}\widehat{V}_{h+1}^{\psi(i)} + \beta\|\phi(\bar{s},a)\|_{\left(\Lambda_{k,h}^{(n)}\right)^{-1}}, H\right\}$ |

*Proof.* We only prove for the transition part; the reward part follows from a similar (and easier) argument. Consider any fixed $\bar{s}$ and $a$. By Definition 4, we have $\left\|\mathbb{P}_h^i\Psi_h^{i\to(n)}(\cdot \mid s_1, a) - \mathbb{P}_h^i\Psi_h^{i\to(n)}(\cdot \mid s_2, a)\right\|_1 \leqslant \epsilon_p$ for any $\psi_h^{i\to(n)}(s_1) = \psi_h^{i\to(n)}(s_2)$. And let $\Psi_h^{(n)\to i}$ is the inverse image function of $\Psi_h^{i\to(n)}$. Then, we get

$$\|\mathbb{P}_h^{(n)}(\cdot \mid \bar{s}, a) - \mathbb{P}_h^i\Psi_h^{i\to(n)}(\cdot \mid s, a)\|_1$$

$$= \left\|\sum_{\widetilde{s}\in\Psi_h^{(n)\to i}(\bar{s})} p^{(n)\to i}(\widetilde{s} \mid \bar{s})\mathbb{P}_h^i\Psi_h^{i\to(n)}(\cdot \mid \widetilde{s}, a) - \mathbb{P}_h^i\Psi_h^{i\to(n)}(\cdot \mid s, a)\right\|_1$$

$$= \left\|\sum_{\widetilde{s}\in\Psi_h^{(n)\to i}(\bar{s})} p^{(n)\to i}(\widetilde{s} \mid \bar{s})\left(\mathbb{P}_h^i\Psi_h^{i\to(n)}(\cdot \mid \widetilde{s}, a) - \mathbb{P}_h^i\Psi_h^{i\to(n)}(\cdot \mid s, a)\right)\right\|_1$$

$$\leqslant \sum_{\widetilde{s}\in\Psi_h^{(n)\to i}(\bar{s})} p^{(n)\to i}(\widetilde{s} \mid \bar{s})\left\|\left(\mathbb{P}_h^i\Psi_h^{i\to(n)}(\cdot \mid \widetilde{s}, a) - \mathbb{P}_h^i\Psi_h^{i\to(n)}(\cdot \mid s, a)\right)\right\|_1$$

$$\leqslant \sum_{\widetilde{s}\in\Psi_h^{(n)\to i}(\bar{s})} p^{(n)\to i}(\widetilde{s} \mid \bar{s})\epsilon_p = \epsilon_p.$$

$\square$

Now, we provide essential lemmas for the proof of the theorem.

**Lemma F.2** (Difference between $\widehat{\boldsymbol{\mu}}_{k,h}^{(n)}$ and $\boldsymbol{\mu}_h^{(n)}$). *For all $k \in [K]$, $h \in [H]$, and $n \in [N]$, we have:*

$$\widehat{\boldsymbol{\mu}}_{k,h}^{(n)} - \boldsymbol{\mu}_h^{(n)} = -\lambda\left(\Lambda_{k,h}^{(n)}\right)^{-1}\boldsymbol{\mu}_h^{(n)} + \left(\Lambda_{k,h}^{(n)}\right)^{-1}\sum_{(\bar{s},a,\bar{s}')\in\mathcal{D}_{k,h}^{(n)}} \phi(\bar{s},a)\boldsymbol{\epsilon}_h^{(n)}(\bar{s},a,\bar{s}')^\top,$$

where $\boldsymbol{\epsilon}_h^{(n)}(\bar{s}, a, \bar{s}') := \mathbb{P}_h^{(n)}(\cdot \mid \bar{s}, a)^\top - \boldsymbol{\delta}(\bar{s}')$

*Proof.*

$$
\begin{aligned}
\widehat{\boldsymbol{\mu}}_{k,h}^{(n)} &= \left(\Lambda_{k,h}^{(n)}\right)^{-1} \sum_{(\bar{s},a,\bar{s}')\in\mathcal{D}_{k,h}^{(n)}} \phi(\bar{s}, a)\boldsymbol{\delta}(\bar{s}')^\top \\
&= \left(\Lambda_{k,h}^{(n)}\right)^{-1} \sum_{(\bar{s},a,\bar{s}')\in\mathcal{D}_{k,h}^{(n)}} \phi(\bar{s}, a)\left(\mathbb{P}_h^{(n)}(\cdot \mid \bar{s}, a) + \boldsymbol{\epsilon}_h^{(n)}(\bar{s}, a, \bar{s}')^\top\right) \\
&= \left(\Lambda_{k,h}^{(n)}\right)^{-1} \sum_{(\bar{s},a,\bar{s}')\in\mathcal{D}_{k,h}^{(n)}} \phi(\bar{s}, a)\left(\phi(\bar{s}, a)^\top \boldsymbol{\mu}_h^{(n)} + \boldsymbol{\epsilon}_h^{(n)}(\bar{s}, a, \bar{s}')^\top\right) \\
&= \left(\Lambda_{k,h}^{(n)}\right)^{-1} \sum_{(\bar{s},a,\bar{s}')\in\mathcal{D}_{k,h}^{(n)}} \phi(\bar{s}, a)\phi(\bar{s}, a)^\top \boldsymbol{\mu}_h^{(n)} + \left(\Lambda_{k,h}^{(n)}\right)^{-1} \sum_{(\bar{s},a,\bar{s}')\in\mathcal{D}_{k,h}^{(n)}} \phi(\bar{s}, a)\boldsymbol{\epsilon}_h^{(n)}(\bar{s}, a, \bar{s}')^\top \\
&= \left(\Lambda_{k,h}^{(n)}\right)^{-1} \left(\Lambda_{k,h}^{(n)} - \lambda I\right)\boldsymbol{\mu}_h^{(n)} + \left(\Lambda_{k,h}^{(n)}\right)^{-1} \sum_{(\bar{s},a,\bar{s}')\in\mathcal{D}_{k,h}^{(n)}} \phi(\bar{s}, a)\boldsymbol{\epsilon}_h^{(n)}(\bar{s}, a, \bar{s}')^\top \\
&= \boldsymbol{\mu}_h^{(n)} - \lambda\left(\Lambda_{k,h}^{(n)}\right)^{-1} \boldsymbol{\mu}_h^{(n)} + \left(\Lambda_{k,h}^{(n)}\right)^{-1} \sum_{(\bar{s},a,\bar{s}')\in\mathcal{D}_{k,h}^{(n)}} \phi(\bar{s}, a)\boldsymbol{\epsilon}_h^{(n)}(\bar{s}, a, \bar{s}')^\top.
\end{aligned}
$$

We conclude the proof by rearranging terms. $\square$

**Lemma F.3.** *Fix $V : \mathcal{S} \to [0, H]^S$ and $\delta' \in (0, 1)$. For all $k \in [K]$, $h \in [H]$ and $n \in [N]$, with probability at least $1 - \delta'$, we have*

$$
\left\| \sum_{(\bar{s},a,\bar{s}')\in\mathcal{D}_{k,h}^{(n)}} \phi(\bar{s}, a)\left(\boldsymbol{\epsilon}_h^{(n)}(\bar{s}, a, \bar{s}')^\top V\right) \right\|_{\left(\Lambda_{k,h}^{(n)}\right)^{-1}} \leq 3H\left(\ln\left(\frac{1}{\delta'}\right) + \frac{d}{2}\ln\left(\frac{(k-1)C_\phi^2 + \lambda}{\lambda}\right)\right)^{1/2}.
$$

*Proof.* Denote $\mathcal{H}_{k,h}$ by all information from the beginning of the learning process up to and including $(\bar{s}_{k,h}, \bar{a}_{k,h})$. Then, we first check the noise terms $\{\boldsymbol{\epsilon}_h^{(n)}(\bar{s}_{k,h}, a_{k,h}, \bar{s}_{k,h+1})^\top V\}_{(\bar{s}_{k,h}, a_{k,h}, \bar{s}_{k,h+1})\in\mathcal{D}_{k,h}^{(n)}}$. Note that $V$ is independent of the data because it is pre-fixed. Thus, we have

$$
\begin{aligned}
\mathbb{E}[\boldsymbol{\epsilon}_h^{(n)}(\bar{s}_{k,h}, a_{k,h}, \bar{s}_{k,h+1})^\top V \mid \mathcal{H}_{k,h}] &= 0 \\
|\boldsymbol{\epsilon}_h^{(n)}(\bar{s}_{k,h}, a_{k,h}, \bar{s}_{k,h+1})^\top V| &\leq \|\boldsymbol{\epsilon}_h^{(n)}(\bar{s}_{k,h}, a_{k,h}, \bar{s}_{k,h+1})\|_1 \|V\|_\infty \leq 2H.
\end{aligned}
$$

Therefore, this is a martingale difference sequence.

By lemma G.1, all $k \in [K]$ and $n \in [N]$, with probability at least $1 - \delta'$, we have:

$$
\left\| \sum_{(\bar{s},a,\bar{s}')\in\mathcal{D}_{k,h}^{(n)}} \phi(\bar{s}, a)\left(\boldsymbol{\epsilon}_h^{(n)}(\bar{s}, a, \bar{s}')^\top V\right) \right\|_{\left(\Lambda_{k,h}^{(n)}\right)^{-1}} \leq 3H\left(\ln\left(\frac{\det(\Lambda_{k,h}^{(n)})^{1/2}\det(\lambda I)^{-1/2}}{\delta'}\right)\right)^{1/2}
$$

$$
\leq 3H\left(\ln\left(\frac{1}{\delta'}\right) + \frac{d}{2}\ln\left(\frac{(k-1)C_\phi^2 + \lambda}{\lambda}\right)\right)^{1/2},
$$

where the last inequality is by Eq. F.4. $\square$

**Lemma F.4** ($\varepsilon$-covering lemma)**.** *For a quadruplet of $(\boldsymbol{\mu}, V, \beta, \Lambda)$ where $V \in [0, H]^S$, $\boldsymbol{\mu} \in \mathbb{R}^{d \times S}$ and $\|\boldsymbol{\mu}V\|_2 \leq C_\mu H\sqrt{d}$, $\beta \in [0, B]$, and the minimum eigen value of $\Lambda$ satisfying $\lambda_{\min}(\Lambda) \geq \lambda$, we define $O_{\boldsymbol{\mu},V,\beta,\Lambda} : \mathcal{S} \to \mathbb{R}$ as follows:*

$$
O_{\boldsymbol{\mu},V,\beta,\Lambda}(s) = \min\left\{\max_a\left(r(s, a) + \phi(s, a)^\top \boldsymbol{\mu}V + \beta\|\phi(s, a)\|_{\Lambda^{-1}}\right), H\right\}, \forall s \in \mathcal{S}.
$$

We also denote the function class $\mathcal{O}$ as:

$$\mathcal{O} = \{O_{\boldsymbol{\mu},V,\beta,\Lambda} : \|\boldsymbol{\mu}V\|_2 \leqslant C_{\boldsymbol{\mu}} H\sqrt{d}, \beta \in [0, B], \lambda_{\min}(\Lambda) \geqslant \lambda\}.$$

*Suppose $\|\phi(s, a)\|_2 \leqslant C_\phi$ for all $(s, a)$. Denote $\varepsilon$-covering number of $\mathcal{O}$ as $\mathcal{N}_\varepsilon$ with respect to the distance $dist(O, O') = \sup_x |O(x) - O'(x)|$. Then, we have*

$$\ln(|\mathcal{N}_\varepsilon|) \leqslant d\ln\left(1 + \frac{4C_\phi C_{\boldsymbol{\mu}} H\sqrt{d}}{\varepsilon}\right) + d^2 \ln\left(1 + \frac{8B^2 C_\phi^2 \sqrt{d}}{\lambda\varepsilon^2}\right).$$

*Proof.* Equivalently, we can reparametrize the function class $\mathcal{O}$ by defining $\mathbf{A} = \beta^2 \Lambda^{-1}$. Then, we get

$$O_{\boldsymbol{\mu},V,\mathbf{A}}(s) = \min\left\{\max_a \left(r(s, a) + \phi(s, a)^\top \boldsymbol{\mu}V + \|\phi(s, a)\|_{\mathbf{A}}\right), H\right\}, \tag{F.1}$$

where $\|\boldsymbol{\mu}V\|_2 \leqslant C_{\boldsymbol{\mu}} H\sqrt{d}$ and $\|\mathbf{A}\|_F \leqslant B^2 \sqrt{d}/\lambda$. Consider any two functions $O_1, O_2 \in \mathcal{O}$ taking the form in Eq. F.1, parameterized by $(\boldsymbol{\mu}_1, V_1, \Lambda_1)$ and $(\boldsymbol{\mu}_2, V_2, \Lambda_2)$, respectively. Since both $\min\{\cdot, H\}$ and $\max_a$ are contraction maps, we have

$$dist(O_1, O_2)$$
$$\leqslant \sup_{s,a} \left|\left(r(s, a) + \phi(s, a)^\top \boldsymbol{\mu}_1 V_1 + \|\phi(s, a)\|_{\mathbf{A}_1}\right) - \left(r(s, a) + \phi(s, a)^\top \boldsymbol{\mu}_2 V_2 + \|\phi(s, a)\|_{\mathbf{A}_2}\right)\right|$$
$$\leqslant \sup_{\phi:\|\phi\|_2 \leqslant C_\phi} \left|\left(\phi^\top \boldsymbol{\mu}_1 V_1 + \|\phi\|_{\mathbf{A}_1}\right) - \left(\phi^\top \boldsymbol{\mu}_2 V_2 + \|\phi\|_{\mathbf{A}_2}\right)\right|$$
$$\leqslant \sup_{\phi:\|\phi\|_2 \leqslant C_\phi} |\phi(\boldsymbol{\mu}_1 V_1 - \boldsymbol{\mu}_2 V_2)| + \sup_{\phi:\|\phi\|_2 \leqslant C_\phi} \sqrt{|\phi^\top(\mathbf{A}_1 - \mathbf{A}_2)\phi|}$$
$$\leqslant C_\phi \|(\boldsymbol{\mu}_1 V_1 - \boldsymbol{\mu}_2 V_2)\|_2 + C_\phi \sqrt{\|\mathbf{A}_1 - \mathbf{A}_2\|_F}, \tag{F.2}$$

where the third inequality holds due to the fact that for any $x, y \geqslant 0$, $|\sqrt{x} - \sqrt{y}| \leqslant \sqrt{|x - y|}$. Now we consider the $\varepsilon/(2C_\phi)$-Net $\mathcal{N}_{\varepsilon/(2C_\phi),\boldsymbol{\mu}V}$ over $\{\boldsymbol{\mu}V \in \mathbb{R}^d : \|\boldsymbol{\mu}V\|_2 \leqslant C_{\boldsymbol{\mu}} H\sqrt{d}\}$ and $\varepsilon^2/(4C_\phi^2)$-Net $\mathcal{N}_{\varepsilon^2/(4C_\phi^2),\mathbf{A}}$ over $\{\mathbf{A} \in \mathbb{R}^{d\times d} : \|\mathbf{A}\|_F \leqslant B^2 \sqrt{d}/\lambda\}$. By applying Lemma G.2, we can bound the size of $\varepsilon$-net $\mathcal{N}_\varepsilon$ for $\mathcal{O}$ as follows:

$$\ln(|\mathcal{N}_\varepsilon|) \leqslant \ln|\mathcal{N}_{\varepsilon/(2C_\phi),\boldsymbol{\mu}V}| + \ln|\mathcal{N}_{\varepsilon^2/(4C_\phi^2),\mathbf{A}}|$$
$$\leqslant d\ln\left(1 + \frac{4C_\phi C_{\boldsymbol{\mu}} H\sqrt{d}}{\varepsilon}\right) + d^2 \ln\left(1 + \frac{8B^2 C_\phi^2 \sqrt{d}}{\lambda\varepsilon^2}\right).$$

$\square$

**Lemma F.5.** *For all $(k, h) \in [K] \times [H]$ and all $n \in [N]$, we have:*

$$\sum_{(k',\bar{s},a)\in Y_{k,h}^{(n)}} \min\left\{1, \|\phi(\bar{s}, a)\|^2_{\left(\Lambda_{k',h}^{(n)}\right)^{-1}}\right\} \leqslant 2\ln\frac{\det(\Lambda_{k+1,h}^{(n)})}{\det(\lambda I)} \leqslant 2d\ln\left(\frac{C_\phi^2 k + \lambda}{\lambda}\right),$$

*where $Y_{k,h}^{(n)} := \left\{\left(k', \bar{s}_{k',h}, a_{k',h}\right) : \bar{s}_{k',h} \in \mathcal{S}^{(n)}\right\}_{k'=1}^k$.*

*Proof.* Since, for any $x \in [0, 1]$, it holds that $x \leqslant 2\ln(1 + x)$, we have

$$\sum_{(k',\bar{s},a)\in Y_{k,h}^{(n)}} \min\left\{1, \|\phi(\bar{s}, a)\|^2_{\left(\Lambda_{k',h}^{(n)}\right)^{-1}}\right\} \leqslant \sum_{(k',\bar{s},a)\in Y_{k,h}^{(n)}} 2\ln\left(1 + \|\phi(\bar{s}, a)\|^2_{\left(\Lambda_{k',h}^{(n)}\right)^{-1}}\right). \tag{F.3}$$

Now, we bound the right hand side of Eq. F.3. Since $\Lambda_{k+1,h}^{(n)} = \Lambda_{k,h}^{(n)} + \mathbb{I}\left(\bar{s}_{k,h} \in \mathcal{S}^{(n)}\right)\phi(\bar{s}_{k,h}, a_{k,h})\phi(\bar{s}_{k,h}, a_{k,h})^\top$, we have

$$\det(\Lambda_{k+1,h}^{(n)})$$

$$= \det(\Lambda_{k,h}^{(n)}) \cdot \det\left(I + \sqrt{\left(\Lambda_{k,h}^{(n)}\right)^{-1}}\,\mathbb{I}\left(\bar{s}_{k,h} \in \mathcal{S}^{(n)}\right)\phi(\bar{s}_{k,h}, a_{k,h})\phi(\bar{s}_{k,h}, a_{k,h})^\top\sqrt{\left(\Lambda_{k,h}^{(n)}\right)^{-1}}\right)$$

$$= \det(\Lambda_{k,h}^{(n)}) \cdot \left(1 + \mathbb{I}\left(\bar{s}_{k,h} \in \mathcal{S}^{(n)}\right)\|\phi(\bar{s}_{k,h}, a_{k,h})\|^2_{\left(\Lambda_{k,h}^{(n)}\right)^{-1}}\right)$$

$$= \ldots$$

$$= \det(\Lambda_{0,h}^{(n)}) \cdot \prod_{k'=1}^{k}\left(1 + \mathbb{I}\left(\bar{s}_{k',h} \in \mathcal{S}^{(n)}\right)\|\phi(\bar{s}_{k',h}, a_{k',h})\|^2_{\left(\Lambda_{k',h}^{(n)}\right)^{-1}}\right)$$

$$= \det(\Lambda_{0,h}^{(n)}) \cdot \prod_{(k',\bar{s},a)\in Y_{k,h}^{(n)}}\left(1 + \|\phi(\bar{s}, a)\|^2_{\left(\Lambda_{k',h}^{(n)}\right)^{-1}}\right)$$

$$= \det(\lambda I) \cdot \prod_{(k',\bar{s},a)\in Y_{k,h}^{(n)}}\left(1 + \|\phi(\bar{s}, a)\|^2_{\left(\Lambda_{k',h}^{(n)}\right)^{-1}}\right),$$

where $Y_{k,h}^{(n)} := \left\{\left(k', \bar{s}_{k',h}, a_{k',h}\right) : \bar{s}_{k',h} \in \mathcal{S}^{(n)}\right\}_{k'=1}^{k}$. Note that empty product, in case of $Y_{k,h}^{(n)} = \varnothing$, is one in convention. Therefore, we have

$$\sum_{(k',\bar{s},a)\in Y_{k,h}^{(n)}} \min\left\{1, \|\phi(\bar{s}, a)\|^2_{\left(\Lambda_{k',h}^{(n)}\right)^{-1}}\right\} \leq \sum_{(k',\bar{s},a)\in Y_{k,h}^{(n)}} 2\ln\left\{1 + \|\phi(\bar{s}, a)\|^2_{\left(\Lambda_{k',h}^{(n)}\right)^{-1}}\right\}$$

$$\leq 2\ln\frac{\det(\Lambda_{k+1,h}^{(n)})}{\det(\lambda I)},$$

which proves the first inequality of the lemma.

Next, we consider the second inequality. For the trace of $\Lambda_{k+1,h}^{(n)}$, we get

$$\mathrm{tr}(\Lambda_{k+1,h}^{(n)}) = \mathrm{tr}\left(\lambda I + \sum_{(k',\bar{s},a)\in Y_{k,h}^{(n)}} \phi(\bar{s}, a)\phi(\bar{s}, a)^\top\right)$$

$$= \lambda d + \sum_{(k',\bar{s},a)\in Y_{k,h}^{(n)}} \|\phi(\bar{s}, a)\|^2_2 \leq C_\phi^2 dk + \lambda d.$$

Then, since $\Lambda_{k+1,h}^{(n)}$ is positive definite, we have

$$\frac{\det(\Lambda_{k+1,h}^{(n)})}{\det(\lambda I)} \leq \left(\frac{\mathrm{tr}(\Lambda_{k+1,h}^{(n)}/d)}{\mathrm{tr}(\lambda I/d)}\right)^d \leq \left(\frac{C_\phi^2 k + \lambda}{\lambda}\right)^d, \tag{F.4}$$

which concludes the proof. $\square$

We now can construct a uniform convergence argument for all $O \in \mathcal{O}$ defined in Lemma F.4.

**Lemma F.6.** *Fix $\delta' \in (0, 1)$. Suppose that $\bar{s}_{k,h} \in \mathcal{S}^{(n)}$. Then, for all $k \in [K]$, $h \in [H]$, and $O \in \mathcal{O}$, there exists an constant $C > 0$ such that, if we denote $E$ as the event that*

$$\left|\left(\left(\widehat{\mathbb{P}}_{k,h}^{(n)} - \mathbb{P}_h^{(n)}\right)(\cdot \mid \bar{s}_{k,h}, a_{k,h})\right) \cdot O\right| \leq C \cdot dH\ln(dT/\delta') \cdot \|\phi(\bar{s}_{k,h}, a_{k,h})\|_{\left(\Lambda_{k,h}^{(n)}\right)^{-1}},$$

*then $P(E) \geq 1 - \delta'$.*

*Proof.*

$$\left| \left( (\widehat{\mathbb{P}}_{k,h}^{(n)} - \mathbb{P}_h^{(n)})(\cdot \mid \bar{s}_{k,h}, a_{k,h}) \right) \cdot O \right|$$

$$= \left| \phi(\bar{s}_{k,h}, a_{k,h})^\top (\widehat{\boldsymbol{\mu}}_{k,h}^{(n)} - \boldsymbol{\mu}_h^{(n)}) \cdot O \right|$$

$$\leqslant \left| \phi(\bar{s}_{k,h}, a_{k,h})^\top \lambda \left( \Lambda_{k,h}^{(n)} \right)^{-1} \boldsymbol{\mu}_h^{(n)} \cdot O \right| + \left| \phi(\bar{s}_{k,h}, a_{k,h})^\top \left( \Lambda_{k,h}^{(n)} \right)^{-1} \sum_{(\bar{s}, a, \bar{s}') \in \mathcal{D}_{k,h}^{(n)}} \phi(\bar{s}, a) \boldsymbol{\epsilon}_h^{(n)}(\bar{s}, a, \bar{s}')^\top \cdot O \right|,$$

$$\tag{F.5}$$

where $\phi(\bar{s}_{k,h}, a_{k,h}) = \phi(\bar{s}_{k,h}, a_{k,h})$ and the last inequality is by Lemma F.2. First, we bound the first term of Eq. F.5.

$$\left| \phi(\bar{s}_{k,h}, a_{k,h})^\top \lambda \left( \Lambda_{k,h}^{(n)} \right)^{-1} \boldsymbol{\mu}_h^{(n)} \cdot O \right| \leqslant \sqrt{\lambda} \|\phi(\bar{s}_{k,h}, a_{k,h})\|_{\left( \Lambda_{k,h}^{(n)} \right)^{-1}} \|\boldsymbol{\mu}_h^{(n)} \cdot O\|_2$$

$$\leqslant C_{\boldsymbol{\mu}} \sqrt{\lambda d} H \|\phi(\bar{s}_{k,h}, a_{k,h})\|_{\left( \Lambda_{k,h}^{(n)} \right)^{-1}}, \tag{F.6}$$

where the first inequality follows the fact that $\left( \Lambda_{k,h}^{(n)} \right)^{-1}$ is at most $1/\lambda$. Now, we bound the second term of Eq. F.5.

$$\left| \phi(\bar{s}_{k,h}, a_{k,h})^\top \left( \Lambda_{k,h}^{(n)} \right)^{-1} \sum_{(\bar{s}, a, \bar{s}') \in \mathcal{D}_{k,h}^{(n)}} \phi(\bar{s}, a) \boldsymbol{\epsilon}_h^{(n)}(\bar{s}, a, \bar{s}')^\top \cdot O \right|$$

$$\leqslant \|\phi(\bar{s}_{k,h}, a_{k,h})\|_{\left( \Lambda_{k,h}^{(n)} \right)^{-1}} \left\| \sum_{(\bar{s}, a, \bar{s}') \in \mathcal{D}_{k,h}^{(n)}} \phi(\bar{s}, a) \boldsymbol{\epsilon}_h^{(n)}(\bar{s}, a, \bar{s}')^\top \cdot O \right\|_{\left( \Lambda_{k,h}^{(n)} \right)^{-1}}.$$

$$\tag{F.7}$$

For arbitrary $O \in \mathcal{O}$, by the definition of $\varepsilon$-cover, there exists a $V \in \mathcal{N}_\varepsilon$, such that $\|O - V\|_\infty \leqslant \varepsilon$. Hence, we get

$$\left\| \sum_{(\bar{s}, a, \bar{s}') \in \mathcal{D}_{k,h}^{(n)}} \phi(\bar{s}, a) \left( \boldsymbol{\epsilon}_h^{(n)}(\bar{s}, a, \bar{s}')^\top \cdot O \right) \right\|_{\left( \Lambda_{k,h}^{(n)} \right)^{-1}}$$

$$\leqslant \left\| \sum_{(\bar{s}, a, \bar{s}') \in \mathcal{D}_{k,h}^{(n)}} \phi(\bar{s}, a) \left( \boldsymbol{\epsilon}_h^{(n)}(\bar{s}, a, \bar{s}')^\top V \right) \right\|_{\left( \Lambda_{k,h}^{(n)} \right)^{-1}} + \left\| \sum_{(\bar{s}, a, \bar{s}') \in \mathcal{D}_{k,h}^{(n)}} \phi(\bar{s}, a) \left( \boldsymbol{\epsilon}_h^{(n)}(\bar{s}, a, \bar{s}')^\top (V - O) \right) \right\|_{\left( \Lambda_{k,h}^{(n)} \right)^{-1}}$$

$$\leqslant \left\| \sum_{(\bar{s}, a, \bar{s}') \in \mathcal{D}_{k,h}^{(n)}} \phi(\bar{s}, a) \left( \boldsymbol{\epsilon}_h^{(n)}(\bar{s}, a, \bar{s}')^\top V \right) \right\|_{\left( \Lambda_{k,h}^{(n)} \right)^{-1}} + \frac{2\varepsilon C_\phi k H}{\sqrt{\lambda}}$$

$$\leqslant 3H \left( \ln \frac{1}{\delta'} + \frac{d}{2} \ln \left( \frac{(k-1)C_\phi^2 + \lambda}{\lambda} \right) + \ln |\mathcal{N}_\varepsilon| \right)^{1/2} + \frac{2\varepsilon C_\phi k H}{\sqrt{\lambda}},$$

where the second inequality is by the fact that

$$
\left\| \sum_{(\bar{s},a,\bar{s}')\in\mathcal{D}_{k,h}^{(n)}} \phi(\bar{s},a)\big(\boldsymbol{\epsilon}_h^{(n)}(\bar{s},a,\bar{s}')^\top (V-O)\big) \right\|_{\left(\Lambda_{k,h}^{(n)}\right)^{-1}} \leqslant 2\varepsilon H \left\| \sum_{(\bar{s},a,\bar{s}')\in\mathcal{D}_{k,h}^{(n)}} \phi(\bar{s},a) \right\|_{\left(\Lambda_{k,h}^{(n)}\right)^{-1}}
$$

$$
\leqslant \frac{2\varepsilon H}{\sqrt{\lambda}} \left\| \sum_{(\bar{s},a,\bar{s}')\in\mathcal{D}_{k,h}^{(n)}} \phi(\bar{s},a) \right\|_2
$$

$$
\leqslant \frac{2\varepsilon C_\phi k H}{\sqrt{\lambda}},
$$

and in the last inequality, we use Lemma F.3 with applying union bound over all functions in $\mathcal{N}_\varepsilon$. Therefore, by applying Lemma F.4, we have

$$
\left\| \sum_{(\bar{s},a,\bar{s}')\in\mathcal{D}_{k,h}^{(n)}} \phi(\bar{s},a)\big(\boldsymbol{\epsilon}_h^{(n)}(\bar{s},a,\bar{s}')^\top \cdot O\big) \right\|_{\left(\Lambda_{k,h}^{(n)}\right)^{-1}}
$$

$$
\leqslant 3H \sqrt{ \ln\frac{1}{\delta'} + \frac{d}{2}\ln\left(\frac{(k-1)C_\phi^2+\lambda}{\lambda}\right) + d\ln\left(1 + \frac{4C_\phi C_{\boldsymbol{\mu}} H\sqrt{d}}{\varepsilon}\right) + d^2\ln\left(1 + \frac{8B^2 C_\phi^2 \sqrt{d}}{\lambda\varepsilon^2}\right) }
$$

$$
+ \frac{2\varepsilon C_\phi k H}{\sqrt{\lambda}}.
$$

Now, we choose the hyperparameter $B = C \cdot dH \ln(dT/\delta')$ where $C$ is an absolute constant. Set $\varepsilon = d/k$, then there exists a absolute constant $C' > 0$ such that

$$
\left\| \sum_{(\bar{s},a,\bar{s}')\in\mathcal{D}_{k,h}^{(n)}} \phi(\bar{s},a)\big(\boldsymbol{\epsilon}_h^{(n)}(\bar{s},a,\bar{s}')^\top \cdot O\big) \right\|_{\left(\Lambda_{k,h}^{(n)}\right)^{-1}} \leqslant C' \cdot dH \ln(dT/\delta'). \tag{F.8}
$$

Combining the results of Eq. F.6, Eq. F.7 and Eq. F.8 together, there exists a absolute constant $C > 0$ such that

$$
\left| \left(\widehat{\mathbb{P}}_{k,h}^{(n)} - \mathbb{P}_h^{(n)}\right)(\cdot \mid \bar{s}_{k,h}, a_{k,h}) \cdot O \right|
$$

$$
\leqslant C_{\boldsymbol{\mu}}\sqrt{\lambda d} H \cdot \|\phi(\bar{s}_{k,h}, a_{k,h})\|_{\left(\Lambda_{k,h}^{(n)}\right)^{-1}} + C' \cdot dH \ln(dT/\delta') \cdot \|\phi(\bar{s}_{k,h}, a_{k,h})\|_{\left(\Lambda_{k,h}^{(n)}\right)^{-1}}
$$

$$
\leqslant C \cdot dH \ln(dT/\delta') \cdot \|\phi(\bar{s}_{k,h}, a_{k,h})\|_{\left(\Lambda_{k,h}^{(n)}\right)^{-1}}.
$$

$\square$

**Lemma F.7** (Optimism). *Suppose that event $E$ defined in Lemma F.6 happens. Then, for all $k \in [K]$ and $h \in [H]$, we have*

$$
\widehat{Q}_{k,h}(s,a) \geqslant Q_h^*(s,a) - H(H+1-h)\epsilon_p, \ \forall s,a.
$$

*Proof.* First, consider a fixed episode $k$. We prove via induction on $h$. For $h = H$, the statement is true since $\widehat{Q}_{k,H}(s,a) = r(s,a) = Q_H^*(s,a)$. Assume the lemma holds for some $1 < h+1 \leqslant H$. Then, for all $s \in \mathcal{S}$, we have

$$
\widehat{V}_{k,h+1}(s) = \max_a \widehat{Q}_{k,h+1}(s,a) \geqslant V_{h+1}^*(s) - H(H-h)\epsilon_p.
$$

For any $(s, a) \in \mathcal{S} \times \mathcal{A}$ and $h \in [H]$, let $s \in \mathcal{S}^i, \exists \psi_h^{i \to (n)}(s)$, and $\bar{s} = \psi_h^{i \to (n)}(s)$. Then, we have

$$r_h(\bar{s}, a) + \widehat{\mathbb{P}}_{k,h}^{(n)}(\cdot \mid \bar{s}, a)\widehat{V}_{k,h+1}^{\psi(i)} + \beta \|\phi(\bar{s}, a)\|_{\left(\Lambda_{k,h}^{(n)}\right)^{-1}} - Q_h^*(s, a)$$

$$= \beta \|\phi(\bar{s}, a)\|_{\left(\Lambda_{k,h}^{(n)}\right)^{-1}} + \widehat{\mathbb{P}}_{k,h}^{(n)}(\cdot \mid \bar{s}, a)\widehat{V}_{k,h+1}^{\psi(i)} - \mathbb{P}_h^i(\cdot \mid s, a)V_{h+1}^*$$

$$\geqslant \beta \|\phi(\bar{s}, a)\|_{\left(\Lambda_{k,h}^{(n)}\right)^{-1}} + \widehat{\mathbb{P}}_{k,h}^{(n)}(\cdot \mid \bar{s}, a)\widehat{V}_{k,h+1}^{\psi(i)} - \mathbb{P}_h^i(\cdot \mid s, a)\widehat{V}_{k,h+1} - H(H-h)\epsilon_p$$

$$= \beta \|\phi(\bar{s}, a)\|_{\left(\Lambda_{k,h}^{(n)}\right)^{-1}} + (\widehat{\mathbb{P}}_{k,h}^{(n)} - \mathbb{P}_h^{(n)})(\cdot \mid \bar{s}, a)\widehat{V}_{k,h+1}^{\psi(i)} + \mathbb{P}_h^{(n)}(\cdot \mid \bar{s}, a)\widehat{V}_{k,h+1}^{\psi(i)} - \mathbb{P}_h^i(\cdot \mid s, a)\widehat{V}_{k,h+1}$$

$$- H(H-h)\epsilon_p$$

$$= \beta \|\phi(\bar{s}, a)\|_{\left(\Lambda_{k,h}^{(n)}\right)^{-1}} + (\widehat{\mathbb{P}}_{k,h}^{(n)} - \mathbb{P}_h^{(n)})(\cdot \mid \bar{s}, a)\widehat{V}_{k,h+1}^{\psi(i)} + \left(\mathbb{P}_h^{(n)}(\cdot \mid \bar{s}, a) - \mathbb{P}_h^i \Psi_h^{i \to (n)}(\cdot \mid s, a)\right) \widehat{V}_{k,h+1}^{\psi(i)}$$

$$- H(H-h)\epsilon_p$$

$$\geqslant \beta \|\phi(\bar{s}, a)\|_{\left(\Lambda_{k,h}^{(n)}\right)^{-1}} + (\widehat{\mathbb{P}}_{k,h}^{(n)} - \mathbb{P}_h^{(n)})(\cdot \mid \bar{s}, a)\widehat{V}_{k,h+1}^{\psi(i)} - H(H+1-h)\epsilon_p \geqslant -H(H+1-h)\epsilon_p,$$

where the first inequality is by the inductive hypothesis that $\widehat{V}_{k,h+1}(s') \geqslant V_{h+1}^*(s') - H(H-h)\epsilon_p$ for all $s' \in \mathcal{S}$, the third equality is by the fact that $\widehat{V}_{h+1}(s') = \widehat{V}_{h+1}^{\psi(i)}(\bar{s}')$ for $s' \in \mathcal{S}^i \cup \mathcal{E}^i$ and $\mathbb{P}_h^i(s' \mid s, a) = 0$ for $s' \notin \mathcal{S}^i \cup \mathcal{E}^i$, the second inequality is by Lemma F.1, and the last inequality is by applying Lemma F.6 with $\widehat{V}_{h+1}^{\psi(i)} \in \mathcal{O}$. This concludes the proof. $\qquad\square$

**Lemma F.8** (Regret decomposition). *On the event $E$ defined in Lemma F.6, we have*

$$\sum_{k=1}^K (V_1^* - V_1^{\pi_k})(s_{k,1}) \leqslant \sum_{n=1}^N \sum_{h=1}^H \sum_{(k, \bar{s}, a) \in Y_{K,h}^{(n)}} 2\beta \sqrt{\min\left\{1, \|\phi(\bar{s}, a)\|_{\left(\Lambda_{k,h}^{(n)}\right)^{-1}}^2\right\}} + \sum_{k=1}^K \sum_{h=1}^H \zeta_{k,h}$$

$$+ 2TH\epsilon_p,$$

*where* $Y_{K,h}^{(n)} := \left\{(k, \bar{s}_{k,h}, a_{k,h}) : \bar{s}_{k,h} \in \mathcal{S}^{(n)})\right\}_{k=1}^K$, $\zeta_{k,h} = \mathbb{E}[(\widehat{V}_{k,h} - V_h^{\pi_k})(s_{k,h}) \mid s_{k,h}, a_{k,h}] - (\widehat{V}_{k,h} - V_h^{\pi_k})(s_{k,h})$ *and* $\beta = C \cdot dH \ln(dT/\delta')$ *for some absolute constant* $C > 0$.

*Proof.* Assume that the event $E$ holds true. For any $k \in [K]$, without loss of generality, suppose $s_{k,1} \in \mathcal{S}^i, s_{k,2} \in \mathcal{S}^{i'}, \cdots$, and $\bar{s}_{k,1} \in \mathcal{S}^{(n)}, \bar{s}_{k,2} \in \mathcal{S}^{(n')}, \cdots$, then we have that

$$(V_1^* - V_1^{\pi_k})(s_{k,1}) \leqslant (\widehat{V}_{k,1} - V_1^{\pi_k})(s_{k,1}) + H^2\epsilon_p$$

$$= (\widehat{Q}_{k,1} - Q_1^{\pi_k})(s_{k,1}, a_{k,1}) + H^2\epsilon_p$$

$$= \beta \|\phi(\bar{s}_{k,1}, a_{k,1})\|_{\left(\Lambda_{k,1}^{(n)}\right)^{-1}} + \widehat{\mathbb{P}}_{k,1}^{(n)}(\cdot \mid \bar{s}_{k,1}, a_{k,1})\widehat{V}_{k,2}^{\psi(i)} - \mathbb{P}_1^i(\cdot \mid s_{k,1}, a_{k,1})V_2^{\pi_k} + H^2\epsilon_p$$

$$= \beta \|\phi(\bar{s}_{k,1}, a_{k,1})\|_{\left(\Lambda_{k,1}^{(n)}\right)^{-1}} + (\widehat{\mathbb{P}}_{k,1}^{(n)} - \mathbb{P}_1^{(n)})(\cdot \mid \bar{s}_{k,1}, a_{k,1})\widehat{V}_{k,2}^{\psi(i)} + \mathbb{P}_1^{(n)}(\cdot \mid \bar{s}_{k,1}, a_{k,1})\widehat{V}_{k,2}^{\psi(i)}$$

$$- \mathbb{P}_1^i(\cdot \mid s_{k,1}, a_{k,1})V_2^{\pi_k} + H^2\epsilon_p$$

$$\leqslant \beta \|\phi(\bar{s}_{k,1}, a_{k,1})\|_{\left(\Lambda_{k,1}^{(n)}\right)^{-1}} + (\widehat{\mathbb{P}}_{k,1}^{(n)} - \mathbb{P}_1^{(n)})(\cdot \mid \bar{s}_{k,1}, a_{k,1})\widehat{V}_{k,2}^{\psi(i)} + \mathbb{P}_1^i \Psi_1^{i \to (n)}(\cdot \mid s_{k,1}, a_{k,1})\widehat{V}_{k,2}^{\psi(i)}$$

$$- \mathbb{P}_1^i(\cdot \mid s_{k,1}, a_{k,1})V_2^{\pi_k} + H(H+1)\epsilon_p$$

$$= \beta \|\phi(\bar{s}_{k,1}, a_{k,1})\|_{\left(\Lambda_{k,1}^{(n)}\right)^{-1}} + (\widehat{\mathbb{P}}_{k,1}^{(n)} - \mathbb{P}_1^{(n)})(\cdot \mid \bar{s}_{k,1}, a_{k,1})\widehat{V}_{k,2}^{\psi(i)} + \mathbb{P}_1^i(\cdot \mid s_{k,1}, a_{k,1})(\widehat{V}_{k,2} - V_2^{\pi_k})$$

$$+ H(H+1)\epsilon_p$$

$$\leqslant 2\beta \|\phi(\bar{s}_{k,1}, a_{k,1})\|_{\left(\Lambda_{k,1}^{(n)}\right)^{-1}} + \mathbb{P}_1^i(\cdot \mid s_{k,1}, a_{k,1})(\widehat{V}_{k,2} - V_2^{\pi_k}) + H(H+1)\epsilon_p$$

$$= 2\beta \|\phi(\bar{s}_{k,1}, a_{k,1})\|_{\left(\Lambda_{k,1}^{(n)}\right)^{-1}} + \mathbb{E}[(\widehat{V}_{k,2} - V_2^{\pi_k})(s_{k,2}) \mid s_{k,1}, a_{k,1}] + H(H+1)\epsilon_p,$$

where the first equality is by Lemma F.7, the second inequality is by Lemma F.1, the fourth equality is by the fact that $\widehat{V}_{k,2}(s') = \Psi_h^{i \to (n)} \widehat{V}_{k,2}^{\psi(i)}(s')$ for $s' \in \mathcal{S}^i \cup \mathcal{E}^i$ and $\mathbb{P}_1^i(s' \mid s_{k,1}, a_{k,1}) = 0$ for $s' \notin \mathcal{S}^i \cup \mathcal{E}^i$, and the third inequality is by Lemma F.6.

Define $\zeta_{k,h}$ as $\mathbb{E}[(\widehat{V}_{k,h} - V_h^{\pi_k})(s_{k,h}) \mid s_{k,h}, a_{k,h}] - (\widehat{V}_{k,h} - V_h^{\pi_k})(s_{k,h})$. Then, we get
$(V_1^* - V_1^{\pi_k})(s_{k,1})$

$$\leqslant 2\beta \, \|\phi(\bar{s}_{k,1}, a_{k,1})\|_{\left(\Lambda_{k,1}^{(n)}\right)^{-1}} + \zeta_{k,h} + (\widehat{V}_{k,2} - V_2^{\pi_k})(s_{k,2}) + H(H+1)\epsilon_p$$

$$\leqslant 2\beta \|\phi(\bar{s}_{k,1}, a_{k,1})\|_{\left(\Lambda_{k,h}^{(n)}\right)^{-1}} + 2\beta\|\phi(\bar{s}_{k,2}, a_{k,2})\|_{\left(\Lambda_k^{(n')}\right)^{-1}} + \zeta_{k,1} + \zeta_{k,2} + (\widehat{V}_{k,3} - V_3^{\pi_k})(s_{k,3}) + H(H+2)\epsilon_p$$

$$\leqslant \ldots$$

$$\leqslant \sum_{n=1}^{N} \sum_{h=1}^{H} \mathbb{I}\left(\bar{s}_{k,h} \in \mathcal{S}^{(n)}\right) 2\beta\|\phi(\bar{s}_{k,h}, a_{k,h})\|_{\left(\Lambda_{k,h}^{(n)}\right)^{-1}} + \sum_{h=1}^{H} \zeta_{k,h} + 2H^2 \epsilon_p.$$

Further, since we immediately have $(V_1^* - V_1^{\pi_k})(s_{k,1}) \leqslant H$, we derive that

$$\sum_{k=1}^{K} (V_1^* - V_1^{\pi_k})(s_{k,1})$$

$$\leqslant \sum_{k=1}^{K} \min\left[\sum_{n=1}^{N} \sum_{h=1}^{H} \mathbb{I}\left(\bar{s}_{k,h} \in \mathcal{S}^{(n)}\right) 2\beta\|\phi(\bar{s}_{k,h}, a_{k,h})\|_{\left(\Lambda_{k,h}^{(n)}\right)^{-1}} + \sum_{h=1}^{H} \zeta_{k,h} + 2H^2\epsilon_p, H\right]$$

$$\leqslant \sum_{k=1}^{K} \min\left[\sum_{n=1}^{N} \sum_{h=1}^{H} \mathbb{I}\left(\bar{s}_{k,h} \in \mathcal{S}^{(n)}\right) 2\beta\|\phi(\bar{s}_{k,h}, a_{k,h})\|_{\left(\Lambda_{k,h}^{(n)}\right)^{-1}}, H\right] + \sum_{k=1}^{K} \sum_{h=1}^{H} \zeta_{k,h} + 2TH\epsilon_p$$

$$\leqslant \sum_{k=1}^{K} \sum_{n=1}^{N} \sum_{h=1}^{H} \mathbb{I}\left(\bar{s}_{k,h} \in \mathcal{S}^{(n)}\right) 2\beta\sqrt{\min\left\{1, \|\phi(\bar{s}_{k,h}, a_{k,h})\|_{\left(\Lambda_{k,h}^{(n)}\right)^{-1}}^2\right\}} + \sum_{k=1}^{K} \sum_{h=1}^{H} \zeta_{k,h} + 2TH\epsilon_p$$

$$= \sum_{n=1}^{N} \sum_{h=1}^{H} \sum_{k=1}^{K} \mathbb{I}\left(\bar{s}_{k,h} \in \mathcal{S}^{(n)}\right) 2\beta\sqrt{\min\left\{1, \|\phi(\bar{s}_{k,h}, a_{k,h})\|_{\left(\Lambda_{k,h}^{(n)}\right)^{-1}}^2\right\}} + \sum_{k=1}^{K} \sum_{h=1}^{H} \zeta_{k,h} + 2TH\epsilon_p$$

$$= \sum_{n=1}^{N} \sum_{h=1}^{H} \sum_{(k,\bar{s},a)\in Y_{K,h}^{(n)}} 2\beta\sqrt{\min\left\{1, \|\phi(\bar{s}, a)\|_{\left(\Lambda_{k,h}^{(n)}\right)^{-1}}^2\right\}} + \sum_{k=1}^{K} \sum_{h=1}^{H} \zeta_{k,h} + 2TH\epsilon_p,$$

where $Y_{K,h}^{(n)} := \left\{\left(k, \bar{s}_{k,h}, a_{k,h}\right) : \bar{s}_{k,h} \in \mathcal{S}^{(n)}\right\}_{k=1}^{K}$ and $T = KH$. This completes the proof. $\qquad\square$

*Proof of Theorem 2.* On the event $E$ defined in Lemma F.6, by Lemma F.8, we have

$$\sum_{k=1}^{K} (V_1^* - V_1^{\pi_k})(s_{k,1}) \leqslant \sum_{n=1}^{N} \sum_{h=1}^{H} \sum_{(k,\bar{s},a)\in Y_{K,h}^{(n)}} 2\beta\sqrt{\min\left\{1, \|\phi(\bar{s}, a)\|_{\left(\Lambda_{k,h}^{(n)}\right)^{-1}}^2\right\}} + \sum_{k=1}^{K} \sum_{h=1}^{H} \zeta_{k,h}$$

$$+ 2TH\epsilon_p. \tag{F.9}$$

First, we bound the first term of Eq. F.9. Then, we have

$$\sum_{n=1}^{N} \sum_{h=1}^{H} \sum_{(k,\bar{s},a)\in Y_{K,h}^{(n)}} 2\beta\sqrt{\min\left\{1, \|\phi(\bar{s}, a)\|_{\left(\Lambda_{k,h}^{(n)}\right)^{-1}}^2\right\}}$$

$$\leqslant 2\beta\sqrt{KH}\sqrt{\sum_{n=1}^{N} \sum_{h=1}^{H} \sum_{(k,\bar{s},a)\in Y_{K,h}^{(n)}} \min\left\{1, \|\phi(\bar{s}, a)\|_{\left(\Lambda_{k,h}^{(n)}\right)^{-1}}^2\right\}}$$

$$\leqslant 2\beta\sqrt{KH}\sqrt{NH \cdot 2d\ln\left(\frac{C_\phi^2 k + \lambda}{\lambda}\right)}, \tag{F.10}$$

where the first inequality is by Cauchy-Schwarz inequality, the last inequality is by Lemma F.5. For the second term of Eq. F.9, since $\mathbb{E}\left[\zeta_{k,h} \mid \mathcal{H}_{k,h-1}\right] = 0$ and $|\zeta_{k,h}| \leqslant 2H$ for all $(k, h)$, $\zeta_{k,h}$ is a bounded martingale difference sequence. Therefore, by the Azuma-Hoeffding inequality, for any $t > 0$, we have

$$P\left(\sum_{k=1}^{K}\sum_{h=1}^{H}\zeta_{k,h} > t\right) \geqslant \exp\left(\frac{-t^2}{2TH^2}\right).$$

Hence, with probability at least $1 - \delta/2$, we have

$$\sum_{k=1}^{K}\sum_{h=1}^{H}\zeta_{k,h} \leqslant \sqrt{2TH^2\ln(2/\delta)} \leqslant 2H\sqrt{T\ln(2/\delta)}. \tag{F.11}$$

Finally, combining Eq. F.9, Eq. F.10, Eq. F.11, and with choice of $\beta = C \cdot dH\ln(2dT/\delta)$ for some absolute constant $C$, with probability at least $1 - \delta$, we have

$$\sum_{k=1}^{K}(V_1^* - V_1^{\pi_k})(s_{k,1})$$

$$\leqslant 2C \cdot dH\ln(2dT/\delta)\sqrt{KH}\sqrt{NH \cdot 2d\ln\left(\frac{C_\phi^2 k + \lambda}{\lambda}\right)} + 2H\sqrt{T\ln(2/\delta)} + 2TH\epsilon_p$$

$$= \widetilde{\mathcal{O}}(\sqrt{d^3 H^3 NT} + TH\epsilon_p)$$

$$= \widetilde{\mathcal{O}}(\sqrt{d_\psi^3 H^3 NT} + TH\epsilon_p),$$

where the last equality is by the fact that $d = d_\psi$. This concludes the proof. $\qquad\square$

## G  TECHNICAL LEMMAS

**Lemma G.1** (Self-normalized process, Abbasi-Yadkori et al. 2011). *Let $\{x_t\}_{t=1}^{\infty}$ be a real-valued stochastic process over the filtration $\{\mathcal{F}_t\}_{t=0}^{\infty}$. Let $x_t$ be conditionally $B$-subgaussian given $\mathcal{F}_{t-1}$. Let $\{\phi_t\}_{t=1}^{\infty}$ with $\phi_t \in \mathcal{F}_{t-1}$ be a stochastic process in $\mathbb{R}^d$. Assume that $\Lambda_0$ is a $d \times d$ positive definite matrix, and let $\Lambda_t = \Lambda_0 + \sum_{s=1}^{t}\phi_s\phi_s^\top$. Then, for any $\delta > 0$, with probability at least $1 - \delta$, we have for all $t \geqslant 0$:*

$$\left\|\sum_{s=1}^{t}\phi_s x_s\right\|_{\Lambda_t^{-1}}^2 \leqslant 2B^2\ln\left(\frac{\det(\Lambda_t)^{1/2}\det(\Lambda_0)^{-1/2}}{\delta}\right).$$

**Lemma G.2** (Covering numbers, Pollard 1990). *The $\varepsilon$-covering number of an euclidean ball of radius $B$ in $\mathbb{R}^d$ is upper bounded by $(1 + 2B/\varepsilon)^d$.*

**Lemma G.3** (Relation between packing and covering numbers, Pollard 1990). *Let $(G, \|\cdot\|)$ be a normed space, and $\Theta \subset G$. Then,*

$$\mathcal{T}(\Theta, 2\varepsilon, \|\cdot\|) \leqslant \mathcal{N}(\Theta, \varepsilon, \|\cdot\|) \leqslant \mathcal{T}(\Theta, \varepsilon, \|\cdot\|),$$

*where $\mathcal{T}(\Theta, \varepsilon, \|\cdot\|)$ is the $\varepsilon$-packing number and $\mathcal{N}(\Theta, \varepsilon, \|\cdot\|)$ is the $\varepsilon$-covering number.*

## H  BLOCK-RIVERSWIM ENVIRONMENT

In this section, we provide a detailed explanation of the structure of the Block-RiverSwim environment. In our experiments, we use the tabular setting as it is easy to construct linear transitions. Note that the tabular setting is a special case of the linear model (Jin et al., 2020).

Block-RiverSwim is constituted of $S$ states, including the initial state $s_1$, $R$ blocks each containing 3 states, and a rewarding end state $s_S$. Hence, the equation $S = 3R + 2$ naturally holds true. Each of these blocks has identical structures: the same transitions and reward functions. Please note that the number of subMDPs $L$ amounts to $R + 2$ (made up of $R$ blocks, $s_1$, and $s_S$) and the total of aggregated subMDPs $N$ equals to 3 (comprising one aggregated block, $s_1$, and $s_S$).

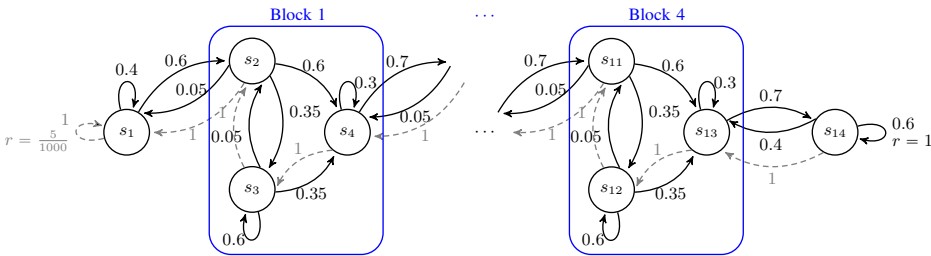

Figure H.1: The Block-RiverSwim environment with $4$ repeating sub-structures ($R$) with size $3$ (Blocks) and $S = 14$ states. State $s_1$ has a small reward of $r(s_1, \texttt{left}) = 5/1000$, while state $s_{14}$ has a large reward of $r(s_{14}, \texttt{right}) = 1$. The dashed arrows indicate deterministic transitions caused by $\texttt{left}$ actions. And line arrows refer to stochastic transitions caused by $\texttt{right}$ actions.

Starting at $s_1$, the agent can select to move left, an action denoted by the gray dashed lines, and as a result, obtain a minor reward, $r(s_1, \texttt{left}) = 0.005$. Alternatively, the agent can choose to navigate to the right, an action represented by the black *solid* lines, in each successive state. The objective for the agent is to maximize its total return by attempting to arrive at the far right state, denoted as $s_S$, and move to the right to earn a large reward $r(s_S, \texttt{right}) = 1$.

Figure H.1 depicts the diagram of Block-RiverSwim with $L = 6$ ($4$ repeating substructures, i.e., $R = 4$, and $2$ unique substructures, thus $N = 3$) and $S = 14$. This environment consists of the initial state $s_1$, the big reward state $s_{14}$, and $4$ blocks, each of which consists of three states. The goal of the agent is to maximize its return by learning the policy that reaches the rightmost state, where the large reward $r(s_{14}, \texttt{right}) = 1$ can be obtained, as fast as possible.

