# OpenReview forum: "Demystifying Linear MDPs and Novel Dynamics Aggregation Framework"
_ICLR.cc/2024/Conference — ICLR 2024 poster_

### Official Review · Reviewer_Taww · 2023-10-20

**Soundness:** 3 good
**Presentation:** 3 good
**Contribution:** 3 good
**Rating:** 6
**Confidence:** 4

**Summary:**

In this paper, the authors prove a lower bound of $d$ for the linear MDP to aptly represent the transition probability. Therefore, they claim that linear MDPs may have regret guarantees dependent of the state space. To address the issue, they propose a novel structural aggregation framework based on dynamics, named as the dynamics aggregation. For this framework, they design a provably efficient hierarchical reinforcement learning algorithm and provide a regret upper bound for this algorithm.

**Strengths:**

1. The problem of proving lower bounds for $d$ and considering hierarchical structure is very interesting and important.

2. The paper is solid, the proof looks correct to me.

3. The lower bound is meaningful, demonstrating the limitations of the linear MDP.

4. The presentation is clear in general. The simulation is interesting.

**Weaknesses:**

My main concern is about technical novelty. The dynamics aggregation is very similar to the misspecified linear MDP considered in [1]. Also, the algorithm is adapted from LSVI-UCB. The result can be expected given [1].

[1] Chi Jin, Zhuoran Yang, Zhaoran Wang, and Michael I Jordan. Provably efficient reinforcement learning with linear function approximation.

**Questions:**

Please refer to the weakness section. Generally speaking, I lean towards acceptance of this paper because of the interesting lower bound and its meaningful insights.

---

> ### Author Response · Authors · 2023-11-13
>
> We sincerely thank you for your time to review our paper and for your positive and valuable feedback. As you pointed out, the findings of Theorem 1 are of great importance. We anticipate that these results will significantly influence the RL community.
>
> Regarding the second part of our paper, we emphasize the **broad impact** of our dynamics aggregation framework, particularly its seamless integration into existing algorithms. This indicates that our framework can be readily applied to a wide range of Linear MDP algorithms, thereby improving their efficiency. Additionally, we believe that investigating methods with higher versatility – those that can easily adapt to existing algorithms – is a crucial research direction.
>
>
> Here are our response to your question:
>
> ### 1. Dynamics aggregation vs misspecified Linear MDP
> If we have understood your question correctly, it appears that you are inquiring about the distinction between employing dynamics aggregation and directly learning from an aggregated MDP that contains a misspecification error. However, we *cannot* directly consider the aggregated MDP since the Q-value may differ for two states that are mapped to the same aggregated (or latent) state. Let's provide an example:
>
> *Consider an MDP with $\mathcal{S} = \{s_1, s_2\}$, $\mathcal{A} = \{a\}$, and $r(s_1, a) =1$ and $r(s_2, a) =0$. Let the horizon length $H$ be 1 for simplicity. And assume that $s_1$ and $s_2$ are mapped into one aggregated state $\bar{s}_1$. Then, it is clear that $Q(s_1, a) = 1$ and $Q(s_2, a) = 0$ since $H=1$.*
>
> Therefore, if we consider only the aggregated MDPs, we cannot differentiate between $Q(s_1, a)$ and $Q(s_2, a)$. Note that in our definition of Q-values (Definition 5), we distinctly define the aggregated Q-values depending on the subMDP to which the state $s$ belongs. Hence, they have the subMDP index $i$.
>
>
> ### 2. Comparison to LSVI-UCB
> The key distinction of our algorithm compared to LSVI-UCB lies in its **model-based** nature. This primarily brings the advantage of **reusability**. In model-free approaches, Q-values are defined by the aggregated feature and specific parameters ($w_h^{\pi}$ in Jin et al., 2020). Thus, it's impossible to distinguish the Q-values for two states mapped to the same aggregated state $\bar{s}$. As mentioned above, the Q-value may differ for two states mapped to the same aggregated state. In essence, model-free approaches fail to effectively leverage the hierarchical structure, even with perfectly learned sub-structures and accurate mapping. On the other hand, our model-based approach enables the reuse of learned dynamics of subMDPs ($\hat{\mu}^{(n)}$) in similar subMDPs. Therefore, it’s important to note the fundamental intuition and significance of using a model-based approach to exploit the hierarchical structure.

---

> > ### Comment · Reviewer_Taww · 2023-11-16
> >
> > Thanks for the detailed explanation. I will keep my score to vote for acceptance.

---

> > > ### Author Response · Authors · 2023-11-19
> > >
> > > We are deeply grateful for your continued support of our paper. If you have any further questions or need more information, please feel free to ask at any time. Thank you.

---

### Official Review · Reviewer_JZay · 2023-10-26

**Soundness:** 3 good
**Presentation:** 3 good
**Contribution:** 2 fair
**Rating:** 6
**Confidence:** 3

**Summary:**

This paper provides a new perspective into the low-dimensional representation structures of MDPs. On the one hand, it casts reasonable doubt on the popular linear representation structure via a simple lower bound on the feature dimension $d$, showing that $d$ may actually scale up with $S$ when the direct reachability $U$ of the environment is limited. On the other hand, it proposes a novel dynamics-based hierarchical aggregation framework that leverages *known* mappings to aggregated sub-MDPs (each equipped with linear representation structure), and shows that it achieves a competitive regret *under certain assumptions*.

**Strengths:**

1. This paper provides a new angle for researchers to understand the fundamental limit of linear MDPs. The result by itself is technically simple and straightforward, but the valuable part of it is the motivation it provides to reflect upon a popular modelling option that is potentially subject to implementation issues.
2. The flow and writing of this paper is good. It provides the reader with adequate background knowledge, illustrates the key points clearly with concrete examples and figures, and accompanies the main results with intuitions and discussions.
3. The mathematical proofs seem correct to the reviewer in the form they are presented in the paper, though results cited from literature are taken for granted.

**Weaknesses:**

1. The authors claim that the dynamics-based aggregation framework proposed in the second part of the paper *addresses the limitation of linear MDPs*. The reviewer is skeptical about the contribution, in that:
    * The aggregation framework seems very artificial. There is not enough motivation why people would have the aggregation mapping $\psi^{i \to (n)}$ in hand *a priori*. Specifically, why don't people directly consider the aggregated MDP when they model real-world scenarios, but rather introduce a large-scale MDP and identifies the similarity between (unnecessarily differentiated) states in the meantime?
    * Apart from the novel idea of substructures, the contribution of the second part seems minimal to the reviewer since it looks like a simple application of LSVI-UCB in sub-MDPs. The proof structure is also similar to that of LSVI-UCB with minor changes.
    * The results will be very interesting if the aggregation structure can be learned (either online or offline) rather than given, but reviewer fails to come up with a quick fix that enables such learning. The reviewer would be more positive about this paper if the authors can, at least, illustrate a potential algorithm design idea to learn the structure.
2. The comparison against LSVI-UCB seems sketchy. The authors claim in the abstract that $d_{\psi}^3 N \ll d^3$ is "readily met in real-world environments", but the only discussion about this is a few conjectured inequalities on page 9 without any real-world data. This seems like too much overclaiming to the reviewer, and thus the authors are urged to provide real-world evidence for their claim.
3. The experiment design can be improved in the following ways:
    * The source code to reproduce the results is not publicly available.
    * The environment is designed to be tabular, which is reducible to linear MDPs, but only in a very inefficient way. The comparison is therefore unfair. It would be more convincing if the algorithms can be evaluated and compared in MDPs with intrinsic low-dimensional structures.
    * The MDPs used in the experiment are very small in size. Experiments are expected to, at least, show adequate scalability of the algorithm.

**Questions:**

1. Why would people have the aggregation mapping $\psi^{i \to (n)}$ in hand *a priori* in modelling?
2. In what kind of real-world environments would the condition $d_{\psi}^3 N \ll d^3$ be met? Please provide concrete examples.
3. In Algorithm 1, $n$ and $i$ seems to be variables that automatically get their values upon observation of states. Should it be written in a clearer way to show that $n$ and $i$ are actually calculated from the state?

---

> ### Author Response · Authors · 2023-11-13
>
> We sincerely appreciate the time and effort you have invested in reviewing our paper. Thank you for your feedback, and we are more than happy to address your comments and questions. Below, we address each point raised in your review with the aim of clarifying and further enriching our research.
>
> ### 1. Motivation for dynamics aggregation
> We *cannot* directly consider the aggregated MDP since the Q-value may differ for two states that are mapped to the same aggregated (or latent) state. Let's provide an example:
>
> *Consider an MDP with $\mathcal{S} = \{s_1, s_2\}$, $\mathcal{A} = \{a\}$, and $r(s_1, a) =1$ and $r(s_2, a) =0$. Let the horizon length $H$ be 1 for simplicity. And assume that $s_1$ and $s_2$ are mapped into one aggregated state $\bar{s}_1$. Then, it is clear that $Q(s_1, a) = 1$ and $Q(s_2, a) = 0$ since $H=1$.*
>
> Therefore, if we consider only the aggregated MDPs, we cannot differentiate between $Q(s_1, a)$ and $Q(s_2, a)$. Note that in our definition of Q-values (Definition 5), we distinctly define the aggregated Q-values depending on the subMDP to which the state $s$ belongs. Hence, they have the subMDP index $i$.
>
> We acknowledge that our initial explanation of the motivation lacked sufficient detail. Thank you for bringing this to our attention. We will ensure to provide a more detailed explanation in the revised version of our paper.
>
> ### 2. Contribution of the second part
>
> The core contribution of our paper extends far beyond the introduction of a new algorithm. Our work provides critical and previously overlooked insights into linear function approximation (the fundamental limitation of Linear MDP that no previous works have addressed), insights that have the potential to fundamentally redirect the trajectory of future research in this area. This novel perspective, presented in the first part of our paper, represents a significant implication on the current theoretical RL research.
> Therefore, we respectfully suggest that the evaluative focus should not solely be on the latter sections of our work. The first part of our paper, which lays the theoretical groundwork for the second part, is deserving of particular attention. It is here that we challenge and expand upon the established norms, offering a fresh lens through which to view and understand the field. We firmly believe in the substantial impact our paper can have on the academic community.
>
> #### 2-1. Versatility of dynamics aggregation
> We believe that the ability to seamlessly integrate our dynamics aggreagtion framework into existing algorithms indeed indicates its extensive impact, rather than suggesting a minimal contribution. This implies that dynamics aggregation can be easily applied to various Linear MDP algorithms, enhancing their efficiency. We also think that exploring methods with greater versatility, ones that can easily adapt to existing algorithms, is an essential avenue for research.
>
>
> #### 2-2. Comparison to LSVI-UCB
> Compared to LSVI-UCB, the most salient feature of our algorithm is that it is **model-based**, which notably offers the advantage of **reusability**. As previously mentioned in 1, re-using aggregated Q-values is not feasible in model-free approaches because the Q-values for two states mapped to the same aggregated state can differ. This limitation means that model-free approaches are unable to effectively utilize the hierarchical structure, even with perfectly learned sub-structures and accurate mapping. In contrast, our model-based approach allows for the reuse of learned dynamics of subMDPs ($\hat{\mu}^{(n)}$) in similar subMDPs. Although the proof structure appears similar, the underlying intuition and significance of the model-based approach in leveraging the hierarchical structure are crucial.
>
> ### 3. Learning dynamics aggregation
> While it is not our focus to deal with learning mapping, we believe that the known dynamics aggregation assumption can be **relaxed** through the use of model selection techniques, as proposed in [4] and [5]. At each episode, the agent selects one of the base mappings to play and receives the rewards associated with the (low level) policy deployed by that base mapping. Then, it updates the (high level) policy  for selecting the base mapping. However, this is currently unclear and beyond the scope of our work. We will leave such research directions for future work.
>
> However, more importantly, we highlight that the assumption of a known hierarchical structure is actually common in hierarchical RL (for example, Wen et al., 2020 [3], also assumed a known equivalence mapping).Furthermore, in real-life scenarios, *humans often utilizes the (approximately) known mapping* between two similar environments, even if they don't know the transitions. Take, for instance, playing different versions of video games such as Super Mario. Despite variations in each game, the core gameplay mechanics stay the same -- a fact that players often grasp intuitively or through prior knowledge.

---

> ### Author Response · Authors · 2023-11-13
>
> Therefore, especially in scenarios typical of human learning, knowledge of dynamics aggregation is quite common. We suggest that the RL community should focus more on HRL frameworks. Doing so could lead to the development of algorithms that approach the efficiency seen in human learning.
>
> Furthermore, it is important to recognize that the presence of an additional structure, or a seemingly stronger assumption, in a research framework should **NOT** lead to its immediate dismissal. On the contrary, when such structures are well-justified and relevant, they can be pivotal in advancing efficient learning methodologies. For example, in the study of LinearMDPs, initial efforts concentrated on cases with *known* features. Only in more recent times have researchers started to develop techniques for learning these features. Similarly, in HRL,  the long-standing assumption has been the presence of *known* options. Very recently, the focus has shifting towards the emerging field of option learning. This principle is at the core of our approach.  Our assumptions regarding dynamics aggregation are not just theoretical ideas; they are based on observable phenomena in actual learning environments (refer Appendix C.2). We are convinced that our discoveries about the fundamental limits of Linear MDPs are vital. Our proposed framework lays the groundwork for future research. It creates opportunities for a balanced approach that combines theoretical thoroughness with practical applicability.
>
>
> [1] Cutkosky, Ashok, Abhimanyu Das, and Manish Purohit. "Upper confidence bounds for combining stochastic bandits." arXiv preprint arXiv:2012.13115 (2020).
>
> [2] Cutkosky, Ashok, et al. "Dynamic balancing for model selection in bandits and rl." International Conference on Machine Learning. PMLR, 2021.
>
> [3] Wen, Zheng, et al. "On efficiency in hierarchical reinforcement learning." Advances in Neural Information Processing Systems 33 (2020): 6708-6718.
>
> ### 4. Additional explanations about $d_\psi^3 N \ll d^3$
> In the abstract, we claimed that the inequality $d_\psi^3 N \ll d^3$ is valid in *most real-world environments* with hierarchical structures.  By *most real-world environments*, we refer to scenarios where the size of directly reachable states **$U$ is much smaller than $S$** (for continuous state space, consider $U$ as $Vol(\mathcal{U})$ and $S$ as $Vol(\mathcal{S})$). The real-world examples where this condition holds are extensively discussed in Section 4. Examples 1, 2, 3, and 4 all meet the condition that $U$ does not scale with $S$. On the other hand, since $M$ represents the maximum cardinality of subMDPs, $M \ll S$ is true if  a hierarchical structure exists (with small sub-structures repeating). Consequently, the inequality  $M^2 \leq S^2/U^3$ is satisfied (as $U$ is negligible), leading to the desired result.
>
>
> ### 5. Experiment design
>
> #### 5.1 Source code
> We have attached the source code as supplementary material.
>
> #### 5.2 Desigining low-dimensional features
> We appreciate your question as it allows us to clarify our main claims and what we aim to emphasize in Theorem 1: it is **impossible** to design low-dimensional Linear MDP structures. The transition kernel of the RiverSwim environment is fully ranked, i.e., $rank(\mathbb{P}) = S$ (refer Appendix B.5). In this scenario, can any method decompose $\mathbb{P}$ into two matrice $\mathbb{P} = \Phi \mu$ without an approximation error, where the column dimension of $\Phi$ and the row dimension of $\mu$ are significantly smaller than $S$? The answer is **NO**. In linear algebra, particularly in methods like SVD, the column dimension of $\Phi$ and the row dimension of $\mu$  are required to be $S$. **This is exactly what we claim in Theorem 1!** Unless directly reachable states (by single-step transition) scale with the entire state space by constant factor, (i.e., $U = \Theta(S)$) the feature dimension $d$ should scale with $S$ in Linear MDP.
>
> #### 5.3 Size of experimental environment
> Since designing low-dimensional Linear MDP structures is impossible, implementation of algorithms in Linear MDPs confines us to a (nearly) tabular setting ($d \approx S$). A major challenge with all existing Linear MDP algorithms is the necessity to compute the inverse of the Gram matrix, incurring a computational cost of $O(d^2) \approx O(S^2)$. This is **intractable** when $S$ is very large. This limitation hinders the expansion of the state space size. It is important to note that all existing Linear MDP algorithm experiments have been conducted in a tabular setting ([4] and [5]). And in this work, we have provided a theoretical proof explaining why they couldn't (Theorem 1).
>
> Please not that, this issue is a common challenge encountered in all Linear MDP algorithms. However, in the case of an MDP with a hierarchical structure, where small sub-structures repeat ($MN \ll S$), our algorithm can be effectively implemented since $d_{\psi}\leq M \ll S$, requiring only a

---

> > ### Author Response · Authors · 2023-11-13
> >
> > computational cost of  $O(d_{\psi}^2)$.
> >
> >
> > [4] Ayoub, Alex, et al. "Model-based reinforcement learning with value-targeted regression." International Conference on Machine Learning. PMLR, 2020.
> > [5] Ishfaq, Haque, et al. "Randomized exploration in reinforcement learning with general value function approximation." International Conference on Machine Learning. PMLR, 2021.
> >
> >
> > ### 6. Calculating $n$ and $i$ from the observed states
> > Thank you for the valuable feedback. We agree that representing the function to allow for the return of values $n$ or $i$ would indeed be more precise. We greatly appreciate this recommendation and will consider integrating it into the revised version of our work.

---

> > > ### Comment · Reviewer_JZay · 2023-11-15
> > > **Thanks for the responses!**
> > >
> > > I appreciate the authors' efforts to settle my questions and doubts via concrete examples and detailed explanations.
> > >
> > > The motivation for, and the necessity of, introducing dynamics aggregation is clearer now. The proposed solution to learn to aggregate seems more or less like a bandit over a few candidate mappings, which is reasonable and does sound promising.
> > >
> > > The justification for the contribution of the second part is acceptable. In the original review I mentioned that I also found the first part more interesting, and the argument of versatility is reasonable. It would be even better if you could come up with a general pipeline to convert popular RL algorithms into aggregation-based algorithms, though that is potentially another independent work.
> > >
> > > I'm still not fully convinced about the claim $d_{\psi}^3 N \ll d^3$. Since you have mentioned the examples listed in Section 4, maybe the most straight-forward is to show a few working aggregation mappings for them (which I think should be doable). It's even better if you can come up with systematic methods to do so.
> > >
> > > I appreciate the points that the authors make about experimental design. It again justifies a common myth that low-dimensional features can be designed in common environments.
> > >
> > > Based on the current responses, I'm happy to raise my rating to 6.

---

> > > > ### Author Response · Authors · 2023-11-19
> > > >
> > > > We are truly delighted to have addressed most of the concerns you raised.  We deeply appreciate your reconsideration and positive reassessment of our paper.
> > > >
> > > > Regarding the claim $d_{\psi}^3N \ll d^3$, we provide concrete examples that may help you better understand.
> > > >
> > > > **1) RiverSwim (refer Figure B.1)**
> > > >
> > > > In RiverSwim, all states are arranged in a line, labeled from $s_1$ to $s_{S}$. At each state, the agent has two possible actions: moving left or moving right. If the agent reaches the left end ($s_1$) of the chain and performs a left action, a deceptive small reward ($0.005$) is given. And if the agent reaches the right end ($s_S$) of the chain and performs a right action, large reward ($1.0$) is given. Let the number of state is $100$ ($S=100$). In this environment, each of the $100$ states can be considered as a separate subMDP, and consequently, the number of subMDPs is equal to the size of the state space, i.e., $L=S=100$. Then, we have
> > > >
> > > > * The maximum size of aggregated subMDPs, $M=3$: One internal state and its neighboring states.
> > > > * The number of aggregated subMDPs, $N=3$: All subMDPs, excluding ${s_1}$ and ${s_{100}}$, which means ${s_2}, {s_3}, \dots, {s_{99}}$, are grouped into a single aggregated subMDP. Meanwhile, ${s_1}$ and ${s_{100}}$ each form  distinct aggregated subMDPs.
> > > > * $MN=3^2 < S$
> > > > * The maximum size of directly reachable states, $U=3$
> > > > * The feature dimension of aggregated subMDPs, $d_{\psi}\leq M =3$: This always holds true.
> > > > * The feature dimension of the original MDP, $d=S=100$: The transition kernel has full rank.
> > > >
> > > > Hence, we get
> > > > * $d_{\psi}^3N \leq 3^4$
> > > > * $M^3N = 3^4$
> > > > * $SM^2 = 100 *3^2$
> > > > * $S^3/U^3 = 100^3/3$
> > > > * $d^3 = 100^3$
> > > >
> > > > Therefore, we derive that $d_{\psi}^3N \leq M^3N < SM^2 < S^3/U^3 < d^3$.
> > > >
> > > > **2) FourRoom (refer Figure 2)**
> > > >
> > > > In FourRoom environment, one of the Gridworld variants (Example 1), the agent must navigate in a maze composed of four rooms interconnected by 4 gaps in the walls. The agent has four possible actions to choose: moving left, right, up, or down. Moreover, all transitions in this environment are deterministic. In this environment, each room can be considered a distinct subMDP, thus $L=4$. As shown in Figure 2, subMDP 1 and subMDP 3 are mapped into aggregated subMDP 1, while subMDP 2 and subMDP 4 are mapped into aggregated subMDP 2. Then, we have
> > > >
> > > > * The maximum size of aggregated subMDPs, $M=5^2+1 = 26$: Each room contains $5^2$ states and a corridor state (the gap in the wall).
> > > > * $S=26*4=104$
> > > > * The number of aggregated subMDPs, $N=2$
> > > > * $MN=26*2=52 < S$
> > > > * The maximum size of directly reachable states, $U=1$: All transitions are deterministic.
> > > > * The feature dimension of aggregated subMDPs, $d_{\psi}\leq M =26$: This always holds true.
> > > > * The feature dimension of the original MDP, $d=S=104$: The transition kernel has full rank.
> > > >
> > > > Hence, we get
> > > > * $d_{\psi}^3N \leq 26^3*2$
> > > > * $M^3N = 26^3*2$
> > > > * $SM^2 = 104 *26^2$
> > > > * $S^3/U^3 = 104^3/1$
> > > > * $d^3 = 104^3$
> > > >
> > > > Thus, we derive that $d_{\psi}^3N \leq M^3N < SM^2 < S^3/U^3 = d^3$.
> > > >
> > > > Environments with larger state spaces are essentially scaled-up versions of those described in the examples above. For example, the garbage collecting robot scenario in Wen et al., 2020 [1] is essentially a large-scale version of the FourRoom Gridworld environment (potentially having more rooms):
> > > >
> > > >
> > > > *“To make this notion of hierarchical structure more concrete, consider a garbage collecting robot navigating in a building. The robot’s goal is to maximize the amount of trash it collects before exiting. The building has $L$ floors, and each floor belongs to one of $N$ floor “types” defined according to some criterion relevant to the robot. A floor of type $i$ has $|\mathcal{E}^i|$ exits to other floors (elevators, stairs, etc.). When exiting a floor, the agent will either get to another floor or leave the building.”*
> > > >
> > > > The floors correspond to rooms in the FourRoom environment. Thus, similar reasoning can be applied to derive $d_{\psi}^3N \ll d^3$.
> > > >
> > > > Similarly, in the case of a continuous state space, by redefining $M$, $U$, and $S$ as the volumes of their respective sets, we can easily derive the desired inequality.
> > > >
> > > > We sincerely hope that these additional examples and explanations have clarified the claim $d_{\psi}^3N \ll d^3$. We deeply appreciate your continued interest in our paper.
> > > >
> > > >
> > > > [1] Wen, Zheng, et al. "On efficiency in hierarchical reinforcement learning." Advances in Neural Information Processing Systems 33 (2020): 6708-6718.

---

### Official Review · Reviewer_M46L · 2023-10-26

**Soundness:** 3 good
**Presentation:** 2 fair
**Contribution:** 3 good
**Rating:** 6
**Confidence:** 2

**Summary:**

This paper provides two interesting contributions:

1) It shows a lower bound on the feature dimension in linear MDPs which depends on the inverse of the maximum reachability of the environment.

2) It provides a novel algorithm with sublinear regret for hierarchical RL where each of the subMDP is a linear MDP.

**Strengths:**

I think that (albeit simple and potentially expected) the lower bound on the feature dimension of Linear MDP is an important result for the RL theory community.

The algorithm (UC-HRL) seems to be an interesting and novel contribution to hierarchical RL.

**Weaknesses:**

The regret bound in Theorem 2 has a linear term which can be made sublinear only if $\epsilon_P$ in Definition 4 is of order $\mathcal{O}(1 / poly(T))$. However it is not very clear from the paper how big $\epsilon_P$ can be for common choices of the approximate feature aggregation mappings $\psi$.

The fact that the aggregating functions $\psi$ are required to be known in advance seems rather strong but somehow common in hierarchical RL.

The discussion after Theorem 2 that justifies that $d^3_{\psi} N \leq d^3$ is unclear in my opinion in particular I do not understand why the regime $MN \leq S$ and $M^2 \leq S^2/U^3$ is a reasonable one. Maybe the author should consider expanding this discussion in their revision.

**Questions:**

1) Can you provide an example of aggregating functions $\psi$ such that the error $\epsilon_P \leq 1/\sqrt{T}$ ?

2) Another related question is: do you expect $\epsilon_P$ to decrease as the number of subMDP $N$ increase ? Is it possible in this way to find the value of $N$ which minimizes the regret bound?

3) Could add an example of Linear MDP where the conditions $MN \leq S$ and $M^2 \leq S^2/U^3$ hold and therefore the hierarchical algorithm has a clear advantage ?

4) Can the assumption of known aggregating functions $\psi$ be relaxed ?

---

> ### Author Response · Authors · 2023-11-13
>
> We sincerely appreciate the time and effort you have invested in reviewing our paper. Thank you for overall positive evaluation of our paper. Below, we address each point raised in your review with the aim of clarifying and further enriching our research.
>
> ### 1. Dynamics aggregation misspecification error
>
> The term $\epsilon_p$ represents the model misspecification error allowing that our modeling assumption may be imperfect. Model misspecification is very **common** in real life and has been often considered in many previous works in Linear MDP. For example, in [1] (Theorem 3.2), [2] (Theorem 1) and [3] (Theorem 1), the authors proposed regret bounds that also involve the term $T \epsilon$, where $\epsilon$ is the model misspecification error. Therefore, the term $\epsilon_p$ represents the effect of model misspecification on regret, rather than implying that regret is directly proportional to $T$. Now, there are many works that **assume** there is no misspecification error, namely assuming that their modeling assumption **perfectly** reflects the environment. We could also do so and not deal with the misspecification error if we desired. Yet, we wanted our framework to be more inclusive when presenting the problem formulation, allowing for possible model misspecification. In essense, this is just a matter of problem setting by our choice. That is, if you wish, we can certainly consider problem setting **without model misspecification**. We strongly believe that this should not be any disadvantage on this part. We will make this much clearer in our revision.
>
> Furthermore, the relationship between $N$ and $\epsilon_p$  is not clear to us. The model misspecification error $\epsilon_p$ is generally not within our control, as it is primarily determined by the specific model (or mapping) we select.
>
> [1] Jin, Chi, et al. "Provably efficient reinforcement learning with linear function approximation." Conference on Learning Theory. PMLR, 2020.
>
> [2] Zanette, Andrea, et al. "Frequentist regret bounds for randomized least-squares value iteration." International Conference on Artificial Intelligence and Statistics. PMLR, 2020.
>
> [3]  Zanette, Andrea, et al. "Learning near optimal policies with low inherent bellman error." ICML, 2020.
>
> ### 2. Additional explanations about $d_\psi^3N \ll d^3$
> #### 2-1. $MN \leq S$
> Note that $M$ denotes the maximum number of states within a subMDP, and $N$ indicates  the number of aggregated subMDPs. Therefore, $MN \leq S$ suggests the presence of a hierarchical structure characterized by repeating small sub-structures. Our experimental setup, Block-RiverSwim, exhibits this hierarchical pattern (see Appendix G and Figure G.1 for details). Figure G.1 illustrates the Block-RiverSwim structure with $S=14$ and $L = 6$ (representing the number of subMDPs). Each substructure, or 'Block', is comprised of 3 states with identical dynamics, resulting in 3 distinct  aggregated subMDPs, thus, $N = 3$:  $\\{s_1\\}, \\{s_{14}\\}$ and Blocks.
>
> #### 2-2. $M^2 \leq S^2/U^3$
> This condition is satisfied in *most real-world environments* with hierarchical structures.  By *most real-world environments*, we refer to scenarios where the size of directly reachable states **$U$ is much smaller than $S$** (for continuous state space, consider $U$ as $Vol(\mathcal{U})$ and $S$ as $Vol(\mathcal{S})$). The real-world examples where this condition holds are extensively discussed in Section 4. Examples 1, 2, 3, and 4 all meet the condition that $U$ does not scale with $S$. Since $M$ represents the maximum cardinality of subMDPs, $M \ll S$ is true if  a hierarchical structure exists (with small sub-structures repeating). Consequently, the inequality  $M^2 \leq S^2/U^3$ is satisfied (as $U$ is negligible), leading to the desired result.
>
> ### 3. Relaxing known mapping assumption
> While it is not our focus to deal with learning mapping, we believe that the known dynamics aggregation assumption can be **relaxed** through the use of model selection techniques, as proposed in [4] and [5]. At each episode, the agent selects one of the base mappings to play and receives the rewards associated with the (low level) policy deployed by that base mapping. Then, it updates the (high level) policy  for selecting the base mapping. However, this is currently unclear and beyond the scope of our work. We will leave such research directions for future work.
>
> However, more importantly, we highlight that the assumption of a known hierarchical structure is actually common in hierarchical RL (for example, Wen et al., 2020 [6], also assumed a known equivalence mapping). Furthermore, in real-life scenarios, *humans often utilizes the (approximately) known mapping* between two similar environments, even if they don't know the transitions. Consider the example of playing various versions of video games like Super Mario. Despite the differences in each game, the underlying gameplay mechanics remain consistent, and this is something we often know.

---

> > ### Author Response · Authors · 2023-11-13
> >
> > Therefore, in common scenarios, particularly in human learning, knowledge of dynamics aggregation is prevalent. We advocate that the RL community should place greater emphasis on these HRL frameworks to develop algorithms that mimic human-level efficiency.
> >
> > Our assumption of known dynamics aggregation in the proposed framework is a **strategic starting point** rather than a limitation. It is logical to first demonstrate efficiency in scenarios with a *known* dynamics aggregation before feasibly extending our approach to cases involving *unknown* dynamics aggregation. This step-by-step progression ensures a solid foundation for future explorations as did numerous previous researches. In the field of Linear MDPs, the research initially focused on scenarios with known features. Only recently has there been a shift towards developing methods to learn these features. Similarly, in hierarchical RL, specifically within the *option* framework [7, 8], the assumption of having known options has been prevalent for a long time. However, the field is now beginning to explore option learning, a relatively new area of study.
> >
> > The models studied in theoretical RL research are mostly still distant from the real world, to start with. Take the widely used Linear MDPs and linear mixture MDPs, for example. Is it reasonable to assume a transition probability is linear in real life? Probably not. However, there's a growing effort within the RL research community to tackle problems that are more applicable to real-world scenarios. Our work is part of this movement. While it's still not perfect, we believe it's a significant step forward. We're particularly hopeful that our approach will make RL with function approximation more efficient, as evidenced by the numerical experiments as well as the theoretical guarantees.
> >
> > [4] Cutkosky, Ashok, Abhimanyu Das, and Manish Purohit. "Upper confidence bounds for combining stochastic bandits." arXiv preprint arXiv:2012.13115 (2020).
> >
> > [5] Cutkosky, Ashok, et al. "Dynamic balancing for model selection in bandits and rl." International Conference on Machine Learning. PMLR, 2021.
> >
> > [6] Wen, Zheng, et al. "On efficiency in hierarchical reinforcement learning." Advances in Neural Information Processing Systems 33 (2020): 6708-6718.
> >
> > [7] Richard S Sutton, Doina Precup, and Satinder Singh. Between mdps and semi-mdps: A framework for temporal abstraction in reinforcement learning. Artificial intelligence, 112(1-2):181–211, 1999.
> >
> > [8] Ronan Fruit, Matteo Pirotta, Alessandro Lazaric, and Emma Brunskill. Regret minimization in mdps with options without prior knowledge. Advances in Neural Information Processing Systems, 30,
> > 2017.

---

> > > ### Comment · Reviewer_M46L · 2023-11-15
> > >
> > > Thanks to the authors for their response.
> > >
> > > I now understand better the role of $\epsilon_P$ I think you could add a comment saying that $\epsilon_P=0$ in case the real MDP is perfectly approximated by the hierarchical linear model.
> > >
> > > Secondly, I think it would be fair to add also an example of MDP where there is no hierarchical  structure to exploit. In this case I think that there would be no hope to improve upon the bound that LSVI-UCB provides for linear MDPs.
> > >
> > > I will keep my positive evaluation of the paper.
> > >
> > > Best,
> > >
> > > Reviewer M46L

---

> > > > ### Author Response · Authors · 2023-11-15
> > > >
> > > > Thank you very much for your consistently positive support and the valuable feedback you have provided. We will ensure that your suggestions are taken into account as we work on the revised version.
> > > >
> > > > Best,
> > > >
> > > > Authors

---

### Official Review · Reviewer_hMDT · 2023-10-30

**Soundness:** 2 fair
**Presentation:** 3 good
**Contribution:** 2 fair
**Rating:** 5
**Confidence:** 3

**Summary:**

The paper focuses on the problem in the linear Markov decision process and its linear representation to the transition probability kernel. It first shows that the current regret results form the literature has the dependency on the state cardinality, which comes from the fact that the dimension of the linear representation for the transition kernel is lower bounded by the rank of the matrix. Then it leverages the technique from the previous works on state aggregation and group mapping to propose a hierarchical version of the linear MDP algorithm, reducing the final regret dependency on the state cardinality to the grouped MDP dimension.

**Strengths:**

1. The paper shows that the dimension of the linear representation for probability transition kernel is lower bounded by |S|/|U|, where |S| is the cardinality of the states and |U| is the maximum size of directly reachable states. If |U| is not the order of |S|, the regret would depend on the state cardinality.
2. The paper develops a hierarchical linear MDP algorithm to reduce the state cardinality dependency in the final regret. It leverages the internal structure of the problem with the state aggregation and mapping from previous works. The final regret and examples show the effect of it.

**Weaknesses:**

1. The paper makes stronger assumption than previous linear MDP algorithms. For the sub-structure that is explored by the paper, it assumes that the dynamic aggregation is known and has the desired boundedness in Definition 4.
2. For the final regret proven by the paper, although the regret seems to be improved in terms of the state cardinality theoretically, it also introduces another T-dependent term characterizing the aggregation gap w.r.t the original probability transition kernel. It's not clear to me whether the newly introduced term would cancel out the improvement from the first term in Theorem 2.
3. The algorithm seems to be a direct extension of the previous works in the tabular case by adding in linear representation and similar analysis.

**Questions:**

The assumption of known dynamic aggregation is strong to me. In reality, how could we extract such information without knowing the transition kernel?

============= After rebuttal  ============
I really appreciate the author/s effort in addressing my concerns and questions. After checking the author/s response and other reviewers' comments, I would like to increase my score from 3 to 5, since I am still not totally onboard with the additional stronger assumptions about the dynamic aggregation and the very similar algorithmic design as previous works.

---

> ### Author Response · Authors · 2023-11-13
>
> We sincerely appreciate the time and effort you have invested in reviewing our paper. Thank you for your feedback, and we are more than happy to address your comments and questions. Below, we address each point raised in your review with the aim of clarifying and further enriching our research.
>
> ### 1. The assumption of known dynamics aggregation stronger?
>
> The reviewer raises a concern about our assumption of known dynamics aggregation being "stronger" than those used in Linear MDPs. We appreciate this perspective, but our analysis suggests a more nuanced understanding is required. In Linear MDPs, for algorithms to achieve an $S$-independent regret bound, a critical yet implicit assumption is that the size of directly reachable states scales with the entire state space. This assumption, as we rigorously show in the paper, is quite stringent and often unrealistic in practical scenarios.
>
> In contrast, our approach, while indeed assuming known dynamics aggregation, does not necessitate such extensive reachability. This difference is pivotal. Our assumption of dynamics aggregation being known, or at least approximable, is not only less restrictive but also more aligned with real-world learning scenarios, as detailed in our paper (refer Appendix C.2). Such dynamics aggregation can often be observed or inferred in various practical contexts, unlike the far-fetched assumption of reachability scaling with the entire state-space, although not explicitly stated (since we are the first to show), required by Linear MDPs.
>
> Therefore, while it might initially appear that our assumptions are stronger, a closer examination reveals that they are, in fact, more grounded in real-world applicability. This distinction is crucial in evaluating the relative "strength" of these assumptions. Hence, it is not straightforward to deem one set of assumptions categorically stronger than the other without considering their practical implications and feasibility.
>
> Moreover, it is important to recognize that the presence of an additional structure, or a seemingly stronger assumption, in a research framework should **NOT** lead to its immediate dismissal. On the contrary, when such structures are well-justified and relevant, they can be pivotal in advancing efficient learning methodologies. This principle is at the core of our approach. The assumptions we make about dynamics aggregation, far from being mere theoretical constructs, are grounded in observable phenomena in real-world learning environments. We believe that our new findings of the fundamental limits of the Linear MDPs is crucial and our proposed framework sets the stage for future research, opening new avenues that blend theoretical rigor with practical relevance.
>
> Let us emphasize that the primary contribution of our paper lies not just in introducing a new algorithm, but in providing a **critical and new insight that all exisitng works in linear function approximation overlooked**  and offering **newer direction for future research**. In this regard, we sincerely ask you to reconsider the **broader impact** that the first part of the paper (on the limitation of Linear MDPs) provides and how it connects to the second part.
>
> #### 1-1. Reemphasize Theorem 1
> To begin with, we want to reiterate the importance of Theorem 1. Let us rephrase the key implications of Theorem 1 and Corollary 1.
>
> *Unless directly reachable states (by single-step transition) scale with the entire state space by constant factor, (i.e., $U = \Theta(S)$) the feature dimension $d$ should scale with $S$ in Linear MDP.*
>
> In the vast majority of practical environments, by a single transition, it is impossible that the entire state space or the constant factor of the entire state space (especially for large state spaces) is reachable, as we discussed in Section 4. Then, our Theorem 1 establishes that the feature dimension $d$ should scale with $S$ to properly express the transition probability in Linear MDP. This result is absolutely crucial, and even serves as a counter-example to the assertion that Linear MDP automatically can induce efficient learning for large state space or even infinite state space (without further assumptions on reachability).
>
> In other words, all existing works in Linear MDPs that claim $S$-independent regret are **only** efficient under a very strong and impractical assumption that $U = \Theta(S)$ (once again, this was an aspect overlooked in all existing papers.). Compared to this, we believe our assumption of known dynamics aggregation is not as strong. As another reviewer mentioned, the assumption of a known hierarchical structure is actually **common in hierarchical RL**: [1] Wen et al., 2020, for instance, also assumed a known equivalence mapping.

---

> ### Author Response · Authors · 2023-11-13
>
> #### 1-2. Intuition behind the dynamics aggregation
> In the second part of our paper (Section 5, 6), we aims to **offer a possible solution to overcome the limitation of Linear MDPs**. While the structural assumption might appear stylistic, it provides meaningful insights. Merely employing function approximation is insufficient; instead, effectively utilizing hierarchical structures can prove beneficial.
>
> Moreover, the dynamics aggregation carries the **practical implications for human learning** (refer Appendix C.2). For example, a driver who can drive in New York will be able to drive immediately in San Francisco and would highly likely be able to drive in Paris as well. This is possible mainly due to the existence of repeating structures and mappings between different structures. Often such mappings are readily available, and we humans are good at exploiting them. Mappings do not have to be perfect, but they could be approximate, as we consider in our work. The decompositions we consider in our work incorporate such intuition into Linear MDPs.
>
> #### 1-3. Known dynamics aggregation
>
> In response to your question, it's important to emphasize that *learning transitions* and *knowing dynamics aggregation* are **entirely separate concepts**. We can learn transitions regardless of our knowledge of dynamics aggregation. Knowledge of dynamics aggregation enables us to leverage the hierarchical structure, which in turn makes the learning process more efficient. However, it's important to note that this knowledge, while beneficial, is not essential for learning transitions.
>
> Furthermore, in real-life scenarios, *humans often utilizes the (approximately) known mapping* between two similar environments, even if they don't know the transitions. Take playing different versions of video games such as Super Mario as an example. Although each game has its unique features, the fundamental gameplay mechanics stay largely the same, and this is something players tend to be aware of. Therefore, in common scenarios, particularly in human learning, knowledge of dynamics aggregation is prevalent. Based on this observation, we think that the RL community should give more attention to HRL frameworks. Doing so could lead to the creation of algorithms that emulate the efficiency of human learning.
>
> Our assumption of known dynamics aggregation in the proposed framework is a **strategic starting point** rather than a limitation. We believe it's sensible to first establish efficiency in situations where dynamics aggregation is *known* before moving on to scenarios with *unknown* dynamics aggregation. This gradual progression lays a solid groundwork for future research, as seen in many prior studies. In Linear MDPs, for instance, the initial focus was on scenarios with known features. The shift towards developing methods to learn these features is a more recent development. Similarly, in HRL, particularly within the 'option' framework [2, 3], relying on known options has been a long-standing practice. Only in recent times has the community begun exploring option learning.  Yet, this area, especially in the context of learning options for regret guarantees, still remains underexplored and presents a fertile ground particularly for functiona approximation.
>
>
> The models studied in theoretical RL research are mostly still distant from the real world, to start with. Take the widely used Linear MDPs and linear mixture MDPs, for example. Is it reasonable to assume a transition probability is linear in real life? Probably not. However, the RL research community is making concerted efforts to tackle settings that are more aligned with practical applications. We are confident that our work contributes to this endeavor, even though it may not be entirely perfect yet. We hope that this work serves as an important step toward solving RL with function approximation more efficiently, as evidenced by the numerical experiments as well as the theoretical guarantees.
>
>
> [1] Wen, Zheng, et al. "On efficiency in hierarchical reinforcement learning." Advances in Neural Information Processing Systems 33 (2020): 6708-6718.
>
> [2] Richard S Sutton, Doina Precup, and Satinder Singh. Between mdps and semi-mdps: A framework for temporal abstraction in reinforcement learning. Artificial intelligence, 112(1-2):181–211, 1999.
>
> [3] Ronan Fruit, Matteo Pirotta, Alessandro Lazaric, and Emma Brunskill. Regret minimization in mdps with options without prior knowledge. Advances in Neural Information Processing Systems, 30,
> 2017.
>
> ### 2. Dynamics aggregation misspecification error
>
> The term $\epsilon_p$ represents the model misspecification error allowing that our modeling assumption may be imperfect. Model misspecification is very **common** in real life and has been often considered in many previous works in Linear MDP. For example, in [4] (Theorem 3.2) and [5] (Theorem 1), the authors proposed regret bounds that also involve the term $T \epsilon$,

---

> ### Author Response · Authors · 2023-11-13
>
> where $\epsilon$ is the model misspecification error. Therefore, the term $\epsilon_p$ represents the effect of model misspecification on regret, rather than implying that regret is directly proportional to $T$. Now, there are many works that **assume** there is no misspecification error, namely assuming that their modeling assumption **perfectly** reflects the environment. We could also do so and not deal with the misspecification error if we desired. Yet, we wanted our framework to be more inclusive when presenting the problem formulation, allowing for possible model misspecification. In essense, this is just a matter of problem setting by our choice. That is, if you wish, we can certainly consider problem setting **without model misspecification**. We strongly believe that this should not be any disadvantage on this part.
>
> [4] Jin, Chi, et al. "Provably efficient reinforcement learning with linear function approximation." Conference on Learning Theory. PMLR, 2020.
>
> [5]  Zanette, Andrea, et al. "Learning near optimal policies with low inherent bellman error." ICML, 2020.
>
> ### 3. Comparison to the tabular case
>
> Dynamics aggregation is a more **general** framework than the equivalence mapping in the tabular case proposed by Wen et al. (2020). The equivalence mapping does not project states into a latent space. Instead, it partitions the original MDPs into disjoint subMDPs and groups them based on identical or similar structures. Therefore, their mapping is defined as a bijection. However, our dynamics aggregation is a surjection: given the aggregated state, we cannot identify the exact states mapped to it. Since all bijective functions are surjective, our framework is more general.
>
> Furthermore, equivalence mapping necessitates that every aspect of the states be identical, even when certain detailed information of the states may not be crucial for discovering the optimal policy. This requirement can potentially constrain the range of scenarios where this framework can be effectively applied.
>
> Our framework, dynamics aggregation, **combines the benefits of both state aggregation and equivalence mapping**. Thus, it results in a considerably simplified representation compared to the other two frameworks. For more details, we provieded a comprehensive description of dynamics aggregation with an example in Appendix C.

---

> ### Author Response · Authors · 2023-11-21
> **To Reviewer hMDT**
>
> Dear Reviewer hMDT,
>
> As we approach the conclusion of the discussion period, we want to ensure that all your questions and comments have been comprehensively addressed. Should there be any outstanding concerns or points of clarification, please let us know. Otherwise, we sincerely and respectfully ask you to consider re-evaluating our work.
>
> We would like to highlight the potential far-reaching impact of our findings on the fundamental limitations of linear MDPs. The first part of our paper, in particular, presents critical insights that we believe are imperative to share with the broader RL research community. This contribution alone, we feel, has significant implications for the field.
>
> Moreover, the second part of our work lays a crucial foundation for addressing RL with a hierarchical structure under function approximation. This represents not only the first formal attempt to provide a rigorous theoretical framework for hierarchical structure in conjunction with function approximation but also includes empirical validation of its effectiveness. Such empirical demonstration and theoretical guarantee, especially within RL theory literature, is relatively uncommon.
>
> Importantly, our approach achieves these results without relying on the impractical assumption of direct reachability to a large portion of the MDP, a necessity in conventional linear MDPs for state-space independent performances as shown in Section 4. We earnestly hope that the value and novelty of our work, as reflected in both theoretical, intuitive, and empirical facets, are recognized. Your support in disseminating these findings is crucial for fostering further research and dialogue within the community.

---

### Official Review · Reviewer_kJCM · 2023-10-31

**Soundness:** 3 good
**Presentation:** 3 good
**Contribution:** 3 good
**Rating:** 6
**Confidence:** 4

**Summary:**

Recent advancements in reinforcement learning (RL) have spotlighted function approximation to address the generalization challenges in tabular Markov Decision Processes (MDPs). Linear MDP, a cornerstone model, has demonstrated that regret bounds are influenced more by feature dimension rather than state space size. However, the authenticity of this claim is examined in this paper. Researchers found that for appropriate representation of the probability space, the feature dimension is inevitably influenced by the size of the state space. A discrepancy was observed in the relationship between the feature dimension and state space size, especially as the latter expands. It's concluded that linear MDPs might not inherently allow learning detached from state space size. To counter this, the paper presents a new hierarchical framework called dynamics aggregation. It encompasses both state aggregation and equivalence mapping, promoting efficiency and adaptability. A hierarchical reinforcement learning (HRL) algorithm is proposed within this structure, which is statistically efficient and offers a comprehensive regret bound. The algorithm is validated against existing methods, showcasing its superior performance.

**Strengths:**

1. The paper questions the widely accepted belief about linear MDPs, delivering a comprehensive critique of its fundamentals.

2. The new framework, which fuses state aggregation and equivalence mapping, holds promise for addressing the limitations of linear MDPs, making it a significant contribution to the field.

3. The proposed HRL algorithm not only introduces an innovative approach to RL but is the first of its kind to provide proven guarantees in function approximation.

4. The inclusion of numerical experiments fortifies the theoretical claims, showcasing the algorithm's efficacy against existing counterparts.

**Weaknesses:**

1. While the new algorithm excels in controlled experiments, its scalability and performance in more complex, real-world scenarios are yet to be determined.

2. While numerical experiments are conducted, this paper mentions several examples in section 4 but does not include experiments and analysis in these examples.

**Questions:**

1.For the proof provided for Theorem 1: suppose that there exists a state-action pair$(s, a)$ for which the transition probabilities are non-zero for more than $U$ states. How would this affect the recursive logic applied in the derivation of $\operatorname{rank}(\mathbb{P}_h) \geqslant\lfloor S / U\rfloor$? Would the derived relationship between $d$, the rank of $\mathbb{P}_h$, and the relationship $\lfloor S / U\rfloor$ still hold?

2. Feature Dimension vs. State Space Size: Given that the research reveals a deeper connection between feature dimension and state space size than previously thought. What is potential future work especially in contexts where the state space is vast?

3. Hierarchical Structures in Real-world Scenarios: With the proposed dynamics aggregation framework depending heavily on the hierarchical structure of problems, how feasible is it to identify or establish such hierarchies in complex, real-world scenarios, where the state dynamics might be more intricate and less structured?

4. This paper mentions several examples in section 4. How does this method work and does it perform well in these examples?

---

> ### Author Response · Authors · 2023-11-13
>
> We sincerely appreciate the time and effort you have invested in reviewing our paper. Thank you for overall positive evaluation of our paper. Below, we address each point raised in your review with the aim of clarifying and further enriching our research.
>
> ### 1. Large-scale experiment
> We appreciate your question as it allows us to clarify our main claims and what we aim to emphasize in Theorem 1: designing large-scale environments in Linear MDPs is **extremely challenging** due to expensive computational costs. As stated in Section 4, in most real-world scale problems, $U$ is minimal compared to $S$, leading to an enormously large $d \approx S$ in Linear MDPs (by Collorary 1). A major challenge with all existing Linear MDP algorithms is the necessity to compute the inverse of the Gram matrix, incurring a computational cost of $O(d^2) \approx O(S^2)$. This becomes  **intractable** when $S$ is very large. Similarly, in response to Q4, this is the reason we could not include experiments for examples $(b)$, $(c)$, and $(d)$ (Riverswim is an example of $(a)$ Gridworld). Please note that, this issue is a common challenge encountered in Linear MDP algorithms. Whereas, in the case of an MDP with a hierarchical structure, where small sub-structures repeat ($MN \ll S$), our algorithm can be effectively implemented since $d_{\psi}\leq M \ll S$. Note that our approach requires only a computational cost of  $O(d_{\psi}^2)$.
>
>
> ### 2. What happens if a state-action pair that can trasit to more than $U$ states exists?
>
> Before addressing your question, we would like to clarify that according to the definition of $U$, a state-action pair $(s,a)$ cannot have non-zero transition probabilities across more than $U$ states. Nonetheless, if your query relates to a situation in which there exists a state-action pair $(s,a)$ has non-zero transition probabilities for almost all states (which means $U = \Theta(S)$), we can provide an explanation. However, please note, as discussed in Section 4, that such a scenario is generally  unrealistic.
>
> The existence of a state-action pair $(s,a)$ with non-zero transition probabilities for almost all states has **little impact** on the result of Theorem 1. Let's reconstruct the transition kernel by excluding the state $s$. And let $U'$ be the maximum number of directly reachable states of this new transition kernel. Now, we can then apply Theorem 1 to the new transition kernel. Consequently, we obtain a similar bound $d \geq \lfloor(S-1)/U' \rfloor$. Hence, our main claim remains valid.
>
> ### 3. Potential future work
> We believe there are two possible directions for future work. The first involves utilizing hierarchical structures, as proposed in this paper. A future direction could be to develop more practical methods for leveraging these hierarchical structures.
>
> The second direction is focused on minimizing or completely removing the dependence on $d$ in the regret bound. Although this may entail some concessions in terms of the $H$ or $T$ terms, achieving a reduction in $d$ could prove to be a significant advancement in our research.
>
> ### 4. Hierarchical structures in real-world scenarios
> Firstly, we would like to highlight that the assumption of a known hierarchical structure is actually **common** in hierarchical RL (for example, Wen et al., 2020 [1], also assumed a known equivalence mapping). Furthermore, in real-life scenarios, humans often know the (approximate) mapping between two similar environments, even if they don't know the transitions. Consider the example of playing various versions of video games like Super Mario. Despite the differences in each game, the underlying gameplay mechanics remain consistent, and this is something we often know. Therefore, in common scenarios, particularly in human learning, knowledge of dynamics aggregation is prevalent. We advocate that the RL community should place greater emphasis on these hierarchical RL frameworks to develop algorithms that mimic human-level efficiency.
>
>
> Our assumption of known dynamics aggregation in the proposed framework is a **strategic starting point** rather than a limitation. It is logical to first demonstrate efficiency in scenarios with a *known* dynamics aggregation before feasibly extending our approach to cases involving *unknown* dynamics aggregation. This step-by-step progression ensures a solid foundation for future explorations as did numerous previous researches. In the field of Linear MDPs, the research initially focused on scenarios with known features. Only recently has there been a shift towards developing methods to learn these features. Similarly, in hierarchical RL, specifically within the *option* framework [2, 3], the assumption of *known* options has been a standard for an extended period. It is only relatively recently that the theory RL community has ventured into the problem of option learning. Yet, this area, especially in the context of learning options for regret guarantees, still remains

---

> > ### Author Response · Authors · 2023-11-13
> >
> > underexplored and presents a fertile ground particularly for functiona approximation.
> >
> >
> > The models studied in theoretical RL research are mostly still distant from the real world, to start with. Take the widely used Linear MDPs and linear mixture MDPs, for example. Is it reasonable to assume a transition probability is linear in real life? Probably not. However, as a research community, we are collectively trying to address more and more practical settings. We strongly believe this work also pushes towards such a direction, albeit still not perfect. We hope that this work serves as an important step toward solving RL with function approximation more efficiently, as evidenced by the numerical experiments as well as the theoretical guarantees.
> >
> >
> > [1] Wen, Zheng, et al. "On efficiency in hierarchical reinforcement learning." Advances in Neural Information Processing Systems 33 (2020): 6708-6718.
> >
> > [2] Richard S Sutton, Doina Precup, and Satinder Singh. Between mdps and semi-mdps: A framework for temporal abstraction in reinforcement learning. Artificial intelligence, 112(1-2):181–211, 1999.
> >
> > [3] Ronan Fruit, Matteo Pirotta, Alessandro Lazaric, and Emma Brunskill. Regret minimization in mdps with options without prior knowledge. Advances in Neural Information Processing Systems, 30,
> > 2017.

---

> > ### Comment · Reviewer_kJCM · 2023-11-21
> >
> > Thanks for the detailed response and I will keep my positive score.

---

> > > ### Author Response · Authors · 2023-11-22
> > >
> > > We greatly value and are thankful for your continuous support of our paper. Thank you.

---

### Author Response · Authors · 2023-11-23
**Final Comments**

We sincerely appreciate the time and effort you devoted to reviewing our work and offering insightful feedback. Your comments during the discussion period were invaluable in enhancing our paper. We found this process both constructive and beneficial. As our discussion period comes to an end, we would like to recap the key points addressed.

### 1. Contribution of the first part
Theorem 1 establishes that **the feature dimension, denoted as $d$, should scale with $S$** to accurately represent the transition probabilities in a linear MDP. This suggests the potential far-reaching impact of our findings on the fundamental limitations of linear MDPs. This invites us to take a fresh look at studies that have previously assumed linear MDPs. We strongly believe that this observation alone has significant implication and is of independent interest to the RL community.

### 2. Contribution of the second part
The second part of our work lays a crucial foundation for addressing RL with a **hierarchical structure under function approximation**. This represents not only the first formal attempt to provide a rigorous theoretical framework for hierarchical structure in conjunction with function approximation but also includes empirical validation of its effectiveness. Such empirical demonstration and theoretical guarantee, especially within RL theory literature, is relatively uncommon.

Importantly, our method attains these outcomes **without depending on the unrealistic assumption** that a large portion of the entire states in the MDP is directly reachable, which is a requirement for $S$-independent performances in traditional linear MDPs, as shown in Section 4. Consequently, though we assume known dynamics aggregation, it is not simple to categorically declare one set of assumptions as inherently stronger than another without taking into account their practical consequences and viability.


### 3. Comparison to LSVI-UCB
The primary distinction of our algorithm compared to LSVI-UCB lies in its **model-based** nature. As shown below (and included in our revised paper), model-free methods are unable to distinguish between the Q-values of two different states (with the same action) that are mapped to the same aggregated state. This limitation arises because, in model-free approaches, Q-values are determined by the aggregated features, which are indistinguishable in the aggregated MDPs. Consequently, model-free approaches cannot effectively leverage the hierarchical structure, even with perfectly learned sub-structures and accurate mapping, whereas our model-based approach can. This highlights the critical intuition and importance of employing a model-based approach to leverage hierarchical structures.

### 4. Impossibility of large-scale experiments
Some reviewers have requested more complex, real-world experiments. However, as shown in Theorem 1, this is **extremely challenging** due to the expensive computational costs. As outlined in Section 4, in most real-world scale problems, $U$ is minimal compared to $S$, resulting in a very large $d \approx S$ in Linear MDPs (by Collorary 1). A significant hurdle with all existing Linear MDP algorithms is their need to compute the inverse of the Gram matrix, which incurs a computational cost of $O(d^2) \approx O(S^2)$. This becomes intractable when $S$ is very large.

> We have also uploaded the revised version of our paper.

### Revision

Following the constructive feedback from reviewers, we've made several updates and have now uploaded a revised version. We sincerely appreciate your insightful comments.

* Add a comment saying that $\epsilon_p=0$ represents the case where the dynamics aggregation mapping is exact: Below Definition 4
* Example for the claim $d_{\psi}^3N \ll d^3$: Appendix C (We have added a new Appendix C, resulting in a change to the numbering of the subsequent appendices: C->D, D->E, E->F, F->G, and G->H.)
* Comparison to LSVI-UCB: Appendix A
* Motivation for dynamics aggregation & Comparison to Misspecified Linear MDP: Appendix D.3

---

### Meta-Review · Area_Chair_1WyR · 2023-12-05

**Metareview:**

In this paper, the authors establish that the feature dimension of a linear MDP is inversely proportional to the maximum number of directly reachable states, challenging the prevalent assumption in theoretical RL, where the dimension of linear MDPs is commonly assumed to be independent of the state space size. Building on this insight, the authors further discover that linear MDPs with hierarchical structures allow for a hierarchical reinforcement learning algorithm, leveraging aggregated sub-structures. This results in regret that depends only on the feature dimension of the subMDPs, rather than the original MDP. The paper also includes numerical comparisons with both linear MDP-based algorithms and hierarchical RL algorithms, demonstrating the efficiency of the proposed algorithm.

The majority of reviews are positive. I concur with the reviewers that the novel perspective on the limits of linear MDPs is intriguing and may partially explain the poor empirical performance of exploration algorithms relying directly on the linear MDP assumption. The proposed dynamics aggregation algorithm also makes a good contribution to the theoretical understanding of hierarchical RL under linear MDP assumptions. However, as pointed out by the reviewers, the authors are encouraged to provide more clarity in their paper regarding the relationship between the two parts. Specifically, while assuming a known hierarchical structure could potentially address the dependence on the original dimension size, it should not be presented as the sole solution, as there may be other milder methods. Additionally, it is unknown whether the hierarchical assumption is uniformly weaker than the assumption on the bounded reachable states. Lastly, the common use of linear MDP in the literature often assumes low-rankness of the transition, automatically leading to a small feature dimension. I recommend discussing this point in your final version, comparing the assumption on directly reachable states with the conventional view of linear MDP. Congratulations on the excellent work, and I look forward to seeing these refinements in your camera-ready version.

**Justification For Why Not Higher Score:**

As some of the reviewers pointed out in the discussion and review, the hierarchical RL part is misleading and the paper needs some revision to improve the presentation and clarity.

**Justification For Why Not Lower Score:**

The contributions on the lower bound of the linear MDP and the hierarchical RL are both novel and interesting, which will be of interest to the RL community.

---

### Decision · Program_Chairs · 2024-01-16

Accept (poster)